# SPATIAL DECONFOUNDER: INTERFERENCE-AWARE DECONFOUNDING FOR SPATIAL CAUSAL INFERENCE

## ABSTRACT

Causal inference in spatial domains faces two intertwined challenges: (1) unmeasured spatial factors, such as weather, air pollution, or mobility, that confound treatment and outcome, and (2) interference from nearby treatments that violate standard no-interference assumptions. While existing methods typically address one by assuming away the other, we show they are deeply connected: *interference reveals structure* in the latent confounder. Leveraging this insight, we propose the **Spatial Deconfounder**, a two-stage method that reconstructs a substitute confounder from local treatment vectors using a conditional variational autoencoder (C-VAE) with a spatial prior, then estimates causal effects via a flexible outcome model. We show that this approach enables nonparametric identification of both direct and spillover effects under weak assumptions—without requiring multiple treatment types or a known model of the latent field. Empirically, we extend `SpaCE`, a benchmark suite for spatial confounding, to include treatment interference, and show that the Spatial Deconfounder consistently improves effect estimation across real-world datasets in environmental health and social science. By turning interference into a multi-cause signal, our framework bridges spatial and deconfounding literatures to advance robust causal inference in structured data.

## 1 INTRODUCTION

Causal inference in spatial settings is critical for science and policy, from estimating the health effects of pollution to evaluating land use, climate interventions, and the spread of infectious disease. Most data in these domains are observational, since large-scale interventions are typically infeasible or unethical, so robust methodology is needed to draw valid conclusions. Yet observational studies in these settings face two fundamental challenges that standard methods rarely address together: (1) *spillover (interference)*, where the treatment at one site affects outcomes at nearby sites, violating the Stable Unit Treatment Value Assumption (SUTVA), and (2) *spatially structured unobserved confounding*, where latent fields such as weather or socioeconomic context jointly drive treatment exposures and outcomes. Both are pervasive, and ignoring either leads to biased conclusions.

Consider air quality and health: respiratory mortality rates depend on local pollution and on neighboring regions' pollution due to transport and mobility, while latent meteorological factors such as temperature and humidity confound both. Any method that neglects interference or hidden confounders risks misleading the actionable decisions policy-makers rely on for regulation and public health.

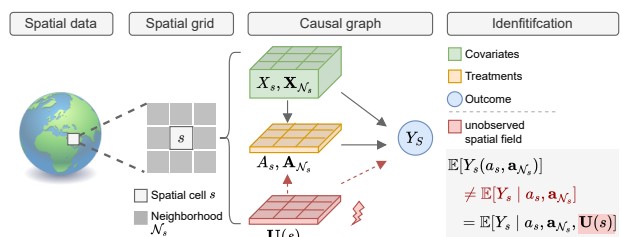

Figure 1: **Schematic of spatial interference/confounding**. Spatial data is represented in geographical cells indexed by site $s$ with neighborhood $\mathcal{N}_s$. The outcome at $s$ (e.g., mortality rate) is affected by the treatments (e.g., air quality) and observed confounders (e.g., demographic informataion) at both $s$ and $\mathcal{N}_s$. However, unobserved latent factors (e.g., humidity) can confound the relationship, rendering causal effects unidentifiable.

Existing approaches for spatial causal inference fall into two camps: (i) Spatial causal methods model spillovers using exposure mappings or autoregressive dependencies but *assume all rele-*

*vant confounders are observed* (Hudgens and Halloran, 2008; Forastiere et al., 2021). If unobserved confounding is present in the data, these methods cannot recover the true effect. (ii) Spatial treatment effect estimation under unobserved confounding is addressed through confounding-adjustment methods—splines, matching, instrumental variables (IVs) (Dupont et al., 2022; Papadogeorgou et al., 2019; Papadogeorgou and Samanta, 2023) by assuming explicit smooth field priors, parametric forms, or exclusion restrictions in the literature. However, these methods treat interference as a nuisance or *neglect the interference structure* completely. Furthermore, they *fail* to correctly model the treatment effect if the parametric assumptions are violated, which is likely in practice.

In an orthogonal literature stream, the *deconfounder* framework (Wang and Blei, 2019) shows that when each unit receives multiple causes, their joint distribution can reveal latent confounders. However, this method is designed for i.i.d. data with simultaneous treatments—not spatial domains with localized interactions. Overall, no method can non-parametrically estimate treatment effects under both interference and unobserved confounding.

We close this gap with the **Spatial Deconfounder**. Our key insight is that interference *creates* the very multi-cause structure that deconfounders require: each unit receives its own treatment together with those of its neighbors, all shaped by the same latent spatial field. Rather than a nuisance, *interference becomes a source of signal for recovering hidden confounders*. Building on this, we develop a non-parametric and model-agnostic two-stage framework that first reconstructs a smooth substitute confounder using a conditional variational autoencoder (C-VAE) with a spatial prior, then estimates direct and spillover effects via any flexible outcome model (e.g., U-Net, GNN). This enables causal identification without requiring multiple treatment types or explicit latent-field models. Of note, our spatial deconfounder framework is completely model-agnostic. In this paper, we present the framework in combination with a C-VAE. However, it can be instantiated with any suitable factor model of choice. Our **contributions** are as follows:

1. We introduce the **Spatial Deconfounder**, a novel *non-parametric and model-agnostic* framework to *jointly* address spatial interference and unmeasured confounding by treating neighborhood treatment exposures as multi-cause signals.

2. We prove *identification* of direct and spillover effects under localized interference and a weak latent-field sufficiency assumption, without requiring a parametric model for the hidden process.

3. We extend the `SpaCE` benchmark to include structured interference and show, across climate-, health-, and social-science datasets, that our method consistently reduces bias relative to spatial autoregressive, matching, and spline-based baselines.

By leveraging interference as a lens into the hidden structure, the Spatial Deconfounder bridges spatial causal inference and multi-cause deconfounding, opening a path to robust causal estimation in complex geographic systems.

## 2    RELATED WORK

We give a brief overview of the related literature (see Appendix A for a comprehensive survey and discussion). Our work sits at the intersection of three main literatures: (i) spatial causal inference under interference and spatially structured confounding, (ii) deconfounding in general average treatment effect (ATE) estimation, and (iii) deep learning for spatial and latent structure modeling.

**Classical spatial causal inference.** Design- and model-based approaches assume exchangeability after conditioning on *observed* covariates (given an exposure mapping) (e.g., Hudgens and Halloran, 2008; Anselin, 1988; Hanks et al., 2015; Forastiere et al., 2021; Tchetgen Tchetgen et al., 2021). They capture spatial dependence (splines/RSR, SAR, GNNs; simulators for domain physics) but do not address *unobserved* spatial confounding.

**Spatial confounding and bias-adjustment methods.** Bias from *unmeasured* spatial structure is mitigated via latent spatial effects, orthogonalization (S2SLS/SPATIAL+), proximity-based matching, IVs, or Bayesian priors (e.g., Hodges and Reich, 2010; Dupont et al., 2022; Papadogeorgou et al., 2019; Angrist et al., 1996). These methods rely on explicit smooth-field models or IV assumptions (or strong priors); none are able to nonparametrically reconstruct the hidden confounder.

**ATE estimation under unobserved confounding.** With unmeasured confounding, point identification typically fails. Sensitivity analyses yield assumption-indexed bounds, trading point identification for robustness (e.g., VanderWeele et al., 2015; Frauen et al., 2023). Another approach is to

reconstruct the unobserved confounder via the *deconfounder* framework, which fits a factor model to multiple causes in order to infer a substitute for the latent confounder, thereby restoring point identification (Wang and Blei, 2019; Bica et al., 2020). However, existing deconfounders require many simultaneous causes and assume no interference. We invert this: interference itself yields multi-cause treatment vectors, enabling latent-field recovery even with a single treatment type.

**Deep learning for spatial modeling.** U-Nets, GNNs, and patch-wise transformers capture multi-scale and long-range spatial structure (e.g., Ronneberger et al., 2015; Kipf, 2016; Liu et al., 2021), yet remain predictive rather than identifying causal effects without added causal structure.

**Deep latent-variable models.** C-VAEs and related deep generative models can recover latent factors from data (Kingma and Welling, 2013; Sohn et al., 2015). We adapt this idea to spatial interference: interference supplies a multi-cause signal to nonparametrically reconstruct a smooth latent confounder, enabling identification of direct and spillover effects without a specified latent field.

**Positioning of our work.** Most spatial–interference methods ignore unmeasured confounders or rely on strong priors, while "deconfounder" methods are not adapted to spatial settings. We close this gap by using interference as a multi-cause signal to nonparametrically reconstruct latent confounders, identifying direct and spillover effects without specifying a latent-field model.

## 3 BACKGROUND AND SETUP

**Notation.** We use uppercase letters (e.g., $X$) for random variables and lowercase letters (e.g., $x$) for their realizations. Bold symbols denote vectors. The distribution of $X$ is written $P_X$, with subscripts omitted when clear from context.

**Data structure: lattice, neighborhoods, and observed variables.** We consider a rectangular lattice $\mathcal{S} = \{(i,j) \mid i \in [N_x],\ j \in [N_y]\}$, where each site $s = (i,j)$ indexes a geographic cell. For a fixed radius $r > 0$, we define the neighborhood of $s$ using the $\ell_\infty$ metric,

$$\mathcal{N}_s = \{s' \in \mathcal{S} : \|s' - s\|_\infty \leq r,\ s' \neq s\}, \quad \text{where } \|s' - s\|_\infty = \max\{|i' - i|, |j' - j|\}. \tag{1}$$

Thus $\mathcal{N}_s$ is the $(2r+1) \times (2r+1)$ square centered at $s$, excluding $s$ itself. We take $r$ to be in *pixels* (multiples of the cell size), though it may also be specified as a physical distance and mapped to the grid resolution. Other shapes (e.g., $\ell_2$ balls) are possible, but we use the square $\ell_\infty$ ball by default for computational convenience.

At each site $s$ we observe covariates $\mathbf{X}_s \in \mathbb{R}^{d_x}$, a binary treatment $A_s \in \{0,1\}$, and an outcome $Y_s \in \mathbb{R}$. For a neighborhood $\mathcal{N}_s$, we write $\mathbf{X}_{\mathcal{N}_s} = \{\mathbf{X}_{s'} : s' \in \mathcal{N}_s\}$, and analogously $A_{\mathcal{N}_s}$ and $Y_{\mathcal{N}_s}$. Realizations are denoted in lowercase, e.g., $\boldsymbol{x}_s$, $a_s$, $y_s$, and $\boldsymbol{x}_{\mathcal{N}_s} = \{\boldsymbol{x}_{s'} : s' \in \mathcal{N}_s\}$. For clarity, we focus on binary treatments, but the framework extends to continuous or multi-valued treatments through standard generalizations of the potential outcomes framework.

**Potential outcomes and interference.** We adopt Rubin's potential outcomes framework (Rubin, 2005). Standard causal inference relies on SUTVA, which rules out interference, i.e., one unit's outcome cannot depend on others' treatments. In spatial settings, this assumption is often violated, since treatment exposures spill over. We assume *localized interference*: the potential outcome at site $s$ depends only on its own treatment and those of its neighbors,

$$Y_s(\mathbf{a}) = Y_s(a_s, \mathbf{a}_{\mathcal{N}_s}), \tag{2}$$

where $\mathbf{a}$ is the full treatment vector, $a_s$ the treatment at $s$, and $\mathbf{a}_{\mathcal{N}_s} = \{a_{s'} : s' \in \mathcal{N}_s\}$. The observed data contain only the realized outcome $Y_s = Y_s(A_s, \mathbf{A}_{\mathcal{N}_s})$ under the assigned intervention.

**Causal estimands.** Let $\mathbf{a}_{\mathcal{N}_s}^{(1)}$ and $\mathbf{a}_{\mathcal{N}_s}^{(0)}$ be two realizations of the neighbor treatments. Our targets are (i) the *average direct effect*, which varies the unit's own treatment while holding neighbors fixed,

$$\tau_{\text{dir}} = \mathbb{E}\big[Y_s(1, \mathbf{a}_{\mathcal{N}_s}) - Y_s(0, \mathbf{a}_{\mathcal{N}_s})\big], \tag{3}$$

and (ii) the *average spillover effect*, which varies neighbors' treatments while holding the unit fixed,

$$\tau_{\text{spill}} = \mathbb{E}\big[Y_s(a, \mathbf{a}_{\mathcal{N}_s}^{(1)}) - Y_s(a, \mathbf{a}_{\mathcal{N}_s}^{(0)})\big], \qquad a \in \{0,1\}, \tag{4}$$

with expectations taken over the observed joint distribution of $(\mathbf{X}_s, A_{\mathcal{N}_s})$.

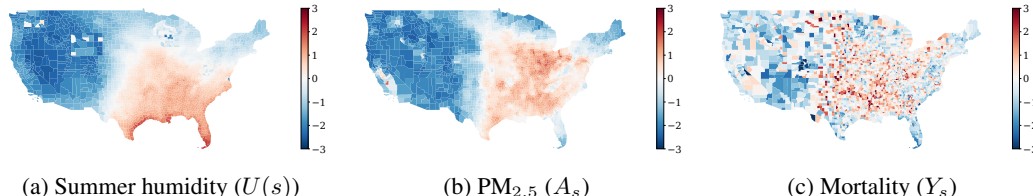

(a) Summer humidity ($U(s)$)    (b) PM$_{2.5}$ ($A_s$)    (c) Mortality ($Y_s$)

Figure 2: Example spatial distribution of (normalized) confounder, treatment, and outcome in real-world dataset. The confounder $U(s)$ (summer humidity) varies smoothly across space, while the treatment $A_s$ (PM$_{2.5}$) shows more local heterogeneity. The outcome $Y_s$ (respiratory and cardiovascular mortality) reflects broader spatial health patterns.

**Unobserved spatial confounding.** To identify the treatment effects in Equations (3) and (4), one typically assumes *ignorability*: potential outcomes $Y_s(a_s, \mathbf{a}_{\mathcal{N}_s})$ are independent of treatment assignment given observed covariates $(\mathbf{X}_s, \mathbf{X}_{\mathcal{N}_s})$. This assumption cannot be tested from the data, and violations lead to biased causal estimates. In practice, many relevant drivers of treatment exposure and outcome remain unobserved. We posit an unobserved spatial field $U : \mathcal{S} \to \mathbb{R}^{d_U}$ that captures latent influences such as topography, wind patterns, or socioeconomic context. Because $U(s)$ may affect both treatment and outcomes, we generally have

$$\mathrm{Cov}(A_s, U(s)) \neq 0 \quad \text{and} \quad \mathrm{Cov}(Y_s(a, \mathbf{a}_{\mathcal{N}_s}), U(s)) \neq 0, \tag{5}$$

where the covariances are understood component-wise when $U(s)$ is vector-valued. Thus, ignorability fails when conditioning only on $\mathbf{X}_s$ and $\mathbf{X}_{\mathcal{N}_s}$. In Section 5, we show that identification can nevertheless be recovered under mild smoothness assumptions on $U$ together with our deconfounding procedure, through reconstructing a substitute latent field from observed treatment patterns.

**Motivating example.** Consider real environmental health data on a $0.25° \times 0.25°$ grid covering the continental United States. At each grid cell $s$, the treatment $A_s$ indicates whether fine particulate matter (PM$_{2.5}$) exceeds the WHO guideline of 10 $\mu$g/m$^3$. Neighbor assignments are defined by a radius of one to two grid cells (roughly 25–50 km). The outcome $Y_s$ is the rate of respiratory and cardiovascular mortality aggregated from hospital records. Latent factors can confound this relationship; for example, a meteorological driver such as humidity varies smoothly across space and may jointly influence both pollution exposures and health outcomes. Figure 2 illustrates treatment, outcome, and such a confounder for this dataset. This example captures the type of smoothly varying, spatially shared latent structure our method targets: large-scale meteorological drivers such as humidity form a latent field $U(s)$ that jointly affects PM$_{2.5}$ exposures and mortality across neighboring counties, while any purely local one-off factors are captured in $(X_s, X_{\mathcal{N}_s})$ or assumed negligible. We formalize this as a latent-field sufficiency assumption in Section 5.

The remainder of the paper shows how the joint vector $(A_s, \mathbf{A}_{\mathcal{N}_s})$—a "multiple-cause" analogue supplied for free by interference—can be harnessed to reconstruct $U(s)$ and obtain unbiased estimates of Equations (3) and (4).

## 4 METHODOLOGY

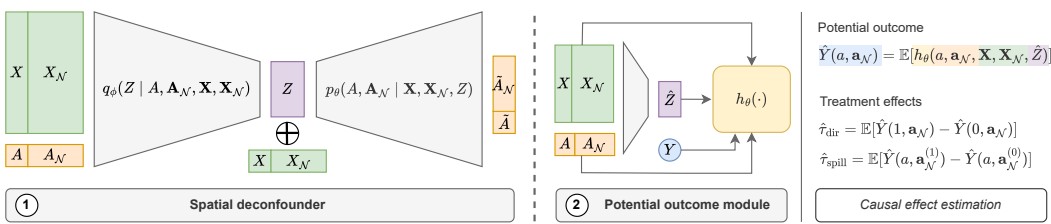

Figure 3: **Architecture of the spatial deconfounder & estimation framework.** Stage ①: The C-VAE takes treatments and observed confounders as input to learn the latent substitute confounder. Stage ②: We employ the reconstructed confounder together with the observed variables (now including the outcome) to train the potential outcome estimation module.

---

**Algorithm 1** Spatial Deconfounder

---

**Input:** Spatial covariates $\{\mathbf{X}_s\}_{s\in\mathcal{S}}$, treatments $\{A_s\}_{s\in\mathcal{S}}$, outcomes $\{Y_s\}_{s\in\mathcal{S}}$, neighborhood radius $r$, grid Laplacian $L$
1: **Stage ①: Confounder reconstruction (C-VAE)**
2: Define encoder $q_\phi(Z_s \,|\, A_s, A_{\mathcal{N}_s}, \mathbf{X}_s, \mathbf{X}_{\mathcal{N}_s}) = \mathcal{N}(\mu_\phi, \mathrm{diag}\,\sigma_\phi^2)$, decoder $p_\psi(A_s \,|\, \mathbf{X}_s, \mathbf{X}_{\mathcal{N}_s}, Z_s)$, and prior $p_\theta(Z) = \mathcal{N}(\mathbf{0}, \tau^{-1}(L + \epsilon I)^{-1})$.
3: Minimize
$$\mathcal{L}_A = \sum_s \mathbb{E}_{q_\phi}\big[-\log p_\psi(A_s \mid \mathbf{X}_s, \mathbf{X}_{\mathcal{N}_s}, Z_s)\big] \;+\; \sum_s D_{\mathrm{KL}}(q_\phi \,\|\, p_\psi),$$
4: Set substitute confounder $\hat{Z}_s \leftarrow \mathbb{E}_{q_\phi}[Z_s]$ for all $s$.
5: **Stage ②: Potential outcome module**
6: Choose a spatial model $h$ (e.g., U-Net) to model the conditional expectation of $Y$ given all observed variables as well as the substitute confounder and fit by minimizing
$$\mathcal{L}_Y = \sum_s \Big(Y_s - h(A_s, A_{\mathcal{N}_s}, \mathbf{X}_s, \mathbf{X}_{\mathcal{N}_s}, \hat{Z}_s)\Big)^2.$$
7: Estimate effects by plug-in contrasts (Eq. 11).

---

As illustrated in Algorithm 1, our approach proceeds in two stages. First, we reconstruct a smooth substitute confounder from the joint distribution of local and neighbor treatments, using a conditional variational autoencoder (C-VAE) that leverages interference as a multi-cause signal. Second, we feed the reconstructed confounder into a flexible potential outcome module for outcome modeling and effect estimation. This separation follows standard practice in deconfounding to prevent mediators from being inadvertently learned into the substitute confounder, which would break the identifiability of the treatment effects.

**Stage ①: Confounder reconstruction.** We model the assignment of treatments $\{A_s\}_{s\in\mathcal{S}}$ using an interference-aware C-VAE. The encoder

$$q_\phi(Z_s \mid A_s, A_{\mathcal{N}_s}, \mathbf{X}_s, \mathbf{X}_{\mathcal{N}_s}) = \mathcal{N}\big(\mu_\phi(\cdot), \mathrm{diag}\,\sigma_\phi^2(\cdot)\big) \tag{6}$$

maps the local treatment and neighborhood treatments, together with local and neighborhood covariates $(\mathbf{X}_s, \mathbf{X}_{\mathcal{N}_s})$, into a latent embedding $Z_s$ of the unobserved spatial field $U(s)$. The decoder

$$p_\psi(A_s \mid \mathbf{X}_s, \mathbf{X}_{\mathcal{N}_s}, Z_s) = \sigma(f_\psi(\mathbf{X}_s, \mathbf{X}_{\mathcal{N}_s}, Z_s)) \tag{7}$$

predicts $A_s$ given covariates and the latent. To encode smoothness, we impose a Gaussian–Markov random-field (GMRF) prior $p_\theta(Z) = \mathcal{N}(\mathbf{0}, \tau^{-1}(L + \epsilon I)^{-1})$ with grid Laplacian $L$, or equivalently a deterministic penalty $\lambda Z^\top L Z$.

Formally, our generative model for the treatment field is

$$p_\theta(Z) = \mathcal{N}\big(\mathbf{0}, \tau^{-1}(L + \epsilon I)^{-1}\big), \qquad p(A \mid X, Z) = \prod_{s\in\mathcal{S}} p_\psi\big(A_s \mid X_s, X_{\mathcal{N}_s}, Z_s\big),$$

with $A_s \mid X_s, X_{\mathcal{N}_s}, Z_s \sim \mathrm{Bernoulli}\big(\sigma(f_\psi(X_s, X_{\mathcal{N}_s}, Z_s))\big)$. Thus, conditional independence of treatments holds across sites given $(Z, X)$, and spatial dependence is encoded entirely via the GMRF prior on $Z$. The "multi-cause" structure of $(A_s, A_{\mathcal{N}_s})$ enters on the inference side through the encoder $q_\phi(Z_s \mid A_s, A_{\mathcal{N}_s}, X_s, X_{\mathcal{N}_s})$, which uses local treatment patterns (plus covariates) to infer a substitute confounder for the local value of the spatial latent field.

This C-VAE is trained by minimizing

$$\mathcal{L}_A(\phi, \psi) = \sum_s \mathbb{E}_{q_\phi}\big[-\log p_\psi(A_s \mid \mathbf{X}_s, \mathbf{X}_{\mathcal{N}_s}, Z_s)\big] + \beta \sum_s D_{\mathrm{KL}}(q_\phi \| p_\psi), \tag{8}$$

with KL warm-up ($\beta \uparrow 1$). After convergence, we set $\hat{Z}_s = \mathbb{E}_{q_\phi}[Z_s]$ as the reconstructed confounder.

Our C-VAE differs from standard C-VAE-type models in two ways tailored to the spatial–interference setting: (i) the encoder explicitly conditions on $(A_s, A_{\mathcal{N}_s}, X_s, X_{\mathcal{N}_s})$, using neighbor treatments as a multi-cause signal, and (ii) the latent field $Z$ is given a GMRF prior with grid Laplacian $L$, enforcing spatial dependence consistent with our latent-field sufficiency assumption (Assumption 4 below).

**Stage ②: Potential outcome module.** Given $\hat{Z}_s$, we estimate outcomes using a flexible function $h$:

$$\hat{Y}_s = \hat{\mathbb{E}}[Y \mid A_s, A_{\mathcal{N}_s}, \mathbf{X}_s, \mathbf{X}_{\mathcal{N}_s}, \hat{Z}_s] = h(A_s, A_{\mathcal{N}_s}, \mathbf{X}_s, \mathbf{X}_{\mathcal{N}_s}, \hat{Z}_s) \tag{9}$$

by minimizing the squared error loss

$$\mathcal{L}_Y = \sum_s \left( Y_s - h(A_s, A_{\mathcal{N}_s}, \mathbf{X}_s, \mathbf{X}_{\mathcal{N}_s}, \hat{Z}_s) \right)^2. \tag{10}$$

This module can be instantiated with any spatial model capable of handling interference and spatial confounding. For example, a U-Net architecture (Ronneberger et al., 2015) captures multiscale spatial dependencies through an encoder–decoder with skip connections. Notably, Oprescu et al. (2025); Ali et al. (2024) use a U-Net to account for interference and spatial confounding in spatiotemporal settings. Other options include graph neural networks, patch-wise transformers, or classical spatial regression models, depending on the data modality.

Effect estimation proceeds by plug-in contrasts: the *direct effect* is

$$\hat{\tau}_{\text{dir}} = \frac{1}{|\mathcal{S}|} \sum_{s \in \mathcal{S}} \left[ h(1, A_{\mathcal{N}_s}, \mathbf{X}_s, \mathbf{X}_{\mathcal{N}_s}, \hat{Z}_s) - h(0, A_{\mathcal{N}_s}, \mathbf{X}_s, \mathbf{X}_{\mathcal{N}_s}, \hat{Z}_s) \right], \tag{11}$$

and analogously for spillover effects by varying $\mathbf{A}_{\mathcal{N}_s}$. By drawing multiple $\hat{Z}_s$ from the full posterior $q_\phi$ instead of the mean, we can obtain uncertainty bands on $\hat{Z}_s$. We can then obtain uncertainty bands (with respect to the substitute confounder) by evaluating Eq. 11 on the different draws of $\hat{Z}_s$.

**Remark 1** (End-to-end variant). *One may train a single network by minimizing $\mathcal{L}_A + \gamma \mathcal{L}_Y$ while blocking gradients from $\mathcal{L}_Y$ into the C-VAE. This preserves mediator avoidance while making the overall implementation and training more straightforward. This separation ensures that the C-VAE is used only to reconstruct a substitute confounder, not to perform outcome estimation end-to-end.*

**Predictive checks.** Following Rubin (1984), we assess whether the substitute confounder adequately explains the treatment assignment through posterior predictive checks. On a held-out validation set, we draw $M$ replicated treatment vectors $\mathbf{a}^{(1)}, \ldots, \mathbf{a}^{(M)}$ from the decoder $p_\psi$ and compare them against the observed assignment $\mathbf{a}$. Specifically, we compute the predictive $p$-value

$$p = \frac{1}{M} \sum_{m=1}^{M} \mathbf{1}\left\{ T(\mathbf{a}^{(m)}) < T(\mathbf{a}) \right\}, \tag{12}$$

where $T(\mathbf{a})$ is a discrepancy statistic measuring model fit. Following Wang and Blei (2019), we use

$$T(\mathbf{a}) = \mathbb{E}_{Z \sim q_\phi}[\log p_\psi(\mathbf{a} \mid \mathbf{X}, Z)], \tag{13}$$

the marginal log-likelihood of the observed assignment under the posterior distribution of $Z$. A value of $p$ close to $0.5$ indicates that the C-VAE reproduces the treatment assignment distribution well, whereas extreme values signal model misspecification. In our experiments, we only consider C-VAE models with $0.25 < p < 0.75$.

## 5 THEORETICAL PROPERTIES OF THE SPATIAL DECONFOUNDER

We now provide conditions under which the Spatial Deconfounder establishes causal identifiability of the direct and spillover effects in Equations (3) and (4). We begin with assumptions on consistency, positivity, and interference structure.

**Assumption 1** (Spatial consistency). *The observed outcome equals the potential outcome under the assigned individual and neighborhood treatments. That is,*

$$Y_s = Y_s(a_s, \mathbf{a}_{\mathcal{N}_s}) \quad \text{if site } s \text{ receives treatment } a_s \text{ and its neighborhood } \mathcal{N}_s \text{ receives } \mathbf{a}_{\mathcal{N}_s}. \tag{14}$$

**Assumption 2** (Spatial positivity). *For any site $s$, covariates $(\mathbf{X}_s, \mathbf{X}_{\mathcal{N}_s})$, and treatment exposures $(a_s, \mathbf{a}_{\mathcal{N}_s})$, the probability of assignment is strictly positive: $0 < \Pr(a_s, \mathbf{a}_{\mathcal{N}_s} \mid \mathbf{X}_s, \mathbf{X}_{\mathcal{N}_s}) < 1$. Furthermore, we require latent positivity conditional on the Z, i.e., $0 < \Pr(a_s, \mathbf{a}_{\mathcal{N}_s} \mid \mathbf{X}_s, \mathbf{X}_{\mathcal{N}_s}, \mathbf{Z}_s) < 1$ if $\Pr(a_s, \mathbf{a}_{\mathcal{N}_s}, \mathbf{X}_s, \mathbf{X}_{\mathcal{N}_s}, \mathbf{Z}_s) > 0$.*

**Assumption 3** (Localized interference). *The potential outcome at site $s$ depends only on its own treatment and those of its neighbors $\mathcal{N}_s$, not on treatments outside $\mathcal{N}_s$.*

Assumptions 1-3 are standard in the causal inference literature (e.g., Chen et al., 2024; Forastiere et al., 2021) and enable identification of the treatment effects. Classical approaches for treatment effect estimation in spatial settings additionally assume ignorability of the joint treatment exposure given observed covariates. We relax this and allow for an unobserved latent field $U : \mathcal{S} \to \mathbb{R}^{d_U}$ spanning the grid. We only require that all confounders affecting purely local variation are observed in $(\mathbf{X}_s, \mathbf{X}_{\mathcal{N}_s})$. This assumption is weaker than full ignorability and is plausible in practice.

**Assumption 4** (Latent field sufficiency). *All confounders that act only on a single site are observed in $(X_s, X_{\mathcal{N}_s})$. Any remaining unobserved confounding is mediated through a shared spatial latent field $U : \mathcal{S} \to \mathbb{R}^{d_U}$ that affects treatment assignments across multiple sites. In particular, there is no additional unobserved confounder $\tilde{U}$ that changes $(A_s, A_{\mathcal{N}_s}, Y_s(a, \mathbf{a}_{\mathcal{N}_s}))$ at some site $s$ without also influencing treatments at other sites $s'$.*

Assumption 4 is the spatial analogue of the "no single-cause confounders" assumption in the deconfounder literature (e.g., Wang and Blei, 2019; Bica et al., 2020): all purely local confounders are observed, and any remaining unobserved confounding arises from a shared latent field $U$ that affects multiple sites. In spatial causal inference, where units are interconnected, causal effects can be identified only if the confounding structure—observed and unobserved—is consistent across the entire lattice. Under this factor-model structure, if the joint treatment distribution admits a representation in terms of a substitute confounder $Z_s$, Proposition 5 of Wang and Blei (2019) implies that the joint assignment $(A_s, A_{\mathcal{N}_s})$ is ignorable given $(X_s, X_{\mathcal{N}_s}, Z_s)$.

Finally, we assume the C-VAE recovers a consistent proxy for the latent field.

**Assumption 5** (Consistency of substitute confounder). *There exists a substitute confounder $Z_s$ that is a deterministic function of the observed causes and covariates,*

$$Z_s = f_\phi(A_s, A_{\mathcal{N}_s}, X_s, X_{\mathcal{N}_s}),$$

*and the encoder $q_\phi(Z_s \mid A_s, A_{\mathcal{N}_s}, X_s, X_{\mathcal{N}_s})$ converges to the corresponding degenerate posterior $\delta_{f_\phi(A_s, A_{\mathcal{N}_s}, X_s, X_{\mathcal{N}_s})}$. Thus $Z_s$ is a deterministic function of $(A_s, A_{\mathcal{N}_s}, X_s, X_{\mathcal{N}_s})$ that, together with $(X_s, X_{\mathcal{N}_s})$, is renders the joint exposure $(A_s, A_{\mathcal{N}_s})$ ignorable as in Definition 1.*

Assumption 5 does not require the learned $Z_s$ to equal the true latent field; it only posits the existence of a deterministic function of $(A_s, A_{\mathcal{N}_s}, X_s, X_{\mathcal{N}_s})$ that restores ignorability when conditioning on $(X_s, X_{\mathcal{N}_s}, Z_s)$, analogous to the "consistency of substitute confounders" condition in Wang and Blei (2019). This is an *identification* assumption rather than a generic claim that CVAEs are identifiable: Theorem 1 only requires that some such $Z_s$ exist, and any estimation procedure that learns a $Z_s$ satisfying ignorability yields a consistent plug-in estimator. In practice, we can encourage such structure by using identifiable objectives such as the IMA-regularized loss of Reizinger et al. (2022).

**Intuition.** Under interference, each site's treatment is observed together with those of its neighbors. Because both $A_s$ and $A_{\mathcal{N}_s}$ are influenced by the same latent field $U(s)$, they provide multiple noisy "views" of the field. By fitting a factor model to the joint distribution of own and neighbor treatments, we reconstruct a substitute confounder $Z_s$ capturing the underlying spatial structure. Conditioning on $Z_s$ (together with observed covariates) restores ignorability, enabling unbiased estimation of direct and spillover effects.

**Theorem 1** (Causal identifiability). *Suppose Assumptions 1–5 hold. Let $Z$ be a piecewise constant function of the assigned causes and covariates $(a, \mathbf{a}_{\mathcal{N}}, \boldsymbol{x}, \boldsymbol{x}_{\mathcal{N}})$ and let the outcome be a separable function of the observed and unobserved variables*

$$\mathbb{E}_Y\big[Y_s(a, \mathbf{a}_{\mathcal{N}}) \mid \mathbf{X}_s = \boldsymbol{x}, \mathbf{X}_{\mathcal{N}_s} = \boldsymbol{x}_{\mathcal{N}}, Z_s = z\big] = f_1(a, \mathbf{a}_{\mathcal{N}}, \boldsymbol{x}, \boldsymbol{x}_{\mathcal{N}}) + f_2(z), \quad (15)$$

$$\mathbb{E}_Y\big[Y_s \mid A_s = a, \mathbf{A}_{\mathcal{N}_s} = \mathbf{a}_{\mathcal{N}}, \mathbf{X}_s = \boldsymbol{x}, \mathbf{X}_{\mathcal{N}_s} = \boldsymbol{x}_{\mathcal{N}}, Z_s = z\big] = f_3(a, \mathbf{a}_{\mathcal{N}}, \boldsymbol{x}, \boldsymbol{x}_{\mathcal{N}}) + f_4(z), \quad (16)$$

*for continuously differentiable functions $f_1, f_2, f_3, f_4$. Consequently, the direct and spillover effects are identifiable as*

$$\tau_{\mathrm{dir}} = \mathbb{E}_{\mathbf{X}_s, \mathbf{X}_{\mathcal{N}_s}, Z}\Big[\mathbb{E}_Y\big[Y_s \mid A_s = 1, \mathbf{A}_{\mathcal{N}_s}, \mathbf{X}_s, \mathbf{X}_{\mathcal{N}_s}, Z_s\big] - \mathbb{E}_Y\big[Y_s \mid A_s = 0, \mathbf{A}_{\mathcal{N}_s}, \mathbf{X}_s, \mathbf{X}_{\mathcal{N}_s}, Z_s\big]\Big], \quad (17)$$

$$\tau_{\mathrm{spill}} = \mathbb{E}_{\mathbf{X}_s, \mathbf{X}_{\mathcal{N}_s}, Z}\Big[\mathbb{E}_Y\big[Y_s \mid a, \mathbf{A}_{\mathcal{N}_s} = \mathbf{a}_{\mathcal{N}_s}^{(1)}, \mathbf{X}_s, \mathbf{X}_{\mathcal{N}_s}, Z_s\big] - \mathbb{E}_Y\big[Y_s \mid a, \mathbf{A}_{\mathcal{N}_s} = \mathbf{a}_{\mathcal{N}_s}^{(0)}, \mathbf{X}_s, \mathbf{X}_{\mathcal{N}_s}, Z_s\big]\Big].$$
$$(18)$$

*Proof.* The proof is provided in Appendix B. $\qquad\square$

**Remark:** The identifiability of our method applies to settings with separable structural equations, a common modeling assumption in the literature (e.g, Wang and Blei, 2019; Papadogeorgou and Samanta, 2023). Many often unobserved or unavailable variables in spatial settings can be assumed to fulfill the equations in practice. In our air quality example, such variables could be persistent differences in baseline respiratory risk driven by unmeasured long-run pollution and chronic disease burden or regional differences in care-seeking and reporting intensity. Additionally, systematic measurement errors in the recorded outcome, i.e., due to the difficulty in detecting, assessing, and correctly identifying respiratory diseases in a unified manner, can represent such latent confounders.

## 6 EXPERIMENTS

We evaluate the Spatial Deconfounder on semi-synthetic datasets from the `SpaCE` benchmark (Tec et al., 2024), modified to incorporate both local interference and spatial confounding on real-world environmental data. To simulate unobserved confounding, we mask key covariates after data generation, i.e., we completely remove them from the dataset. We then compare different instantiations of our method against a range of spatial baselines under both local and spatial confounding scenarios. The section proceeds as follows: we describe the `SpaCE` environment and our data generation process, introduce the baselines and evaluation metrics, and finally interpret the results.

Additional details - including data generation, residual sampling, packages, hyperparameter tuning, and validation procedures - can be found in Appendix C. Replication code is available at `https://anonymous.4open.science/r/Spatial-Deconfounder`.

**Datasets and SpaCE Benchmark.** We build on the `SpaCE` benchmark (Tec et al., 2024), which provides semi-synthetic spatial datasets for causal inference under unobserved confounding. In its original form, `SpaCE` simulates causal effects by masking important covariates in real-world environmental and health data, but it assumes independent treatments and does not account for interference between neighboring units. This makes it inadequate for evaluating methods, such as ours, that explicitly address both unobserved spatial confounding and localized spillovers.

To address this, we extend the `SpaCE` data generation process in two ways. First, we project the raw environmental data onto a uniform $0.25° \times 0.25°$ latitude–longitude grid, allowing convolutional architectures to exploit spatial locality while preserving large-scale patterns. Second, we incorporate *interference* into the potential outcome model by allowing outcomes to depend not only on local treatment $A_s$ but also on neighbor treatments $A_{\mathcal{N}_s}$ within radius $r_d$. Specifically, we generate outcomes under two confounding regimes:

$$\text{(Local confounding)} \quad \hat{Y}_s = f(A_s, A_{\mathcal{N}_s}, X_s) + R_s, \tag{19}$$

$$\text{(Spatial confounding)} \quad \hat{Y}_s = f(A_s, A_{\mathcal{N}_s}, X_s, X_{\mathcal{N}_s}) + R_s, \tag{20}$$

where $f$ is a predictive function learned from the observed data, $X_s$ are observed covariates, and $R_s$ are exogenous residuals. The local setting restricts confounding to site-level variables, while the spatial setting also allows neighborhood covariates to act as confounders.

**Semi-synthetic data generation.** To construct $\hat{Y}_s$, we proceed in four steps: (1) fit $f$ using ensembles of machine learning models to predict observed outcomes $Y_s$, (2) compute residuals $\hat{R}_s = Y_s - f(\cdot)$ and estimate their spatial distribution $P_R$, (3) replace endogenous residuals with exogenous noise $R_s \sim P_R$, and (4) generate counterfactuals by varying local and neighbor treatments while holding confounders and residuals fixed. To simulate hidden confounding, we identify influential covariates by measuring the change in predictive performance when each is removed, then mask the most important ones at training and evaluation time.

**Raw datasets.** From the full `SpaCE` suite, we focus in the main text on two collections:

*Air Pollution and Mortality:* County-level data for the mainland US in 2010, including elderly mortality (CDC), fine particulate matter ($PM_{2.5}$) treatment exposure (Di et al., 2019), behavioral risk factors (BRFSS) (Centers for Disease Control and Prevention, 2010), and Census demographics (U.S. Census Bureau, 2010). We study the effect of $PM_{2.5}$ exposure (treatment) on mortality ($PM_{2.5} \rightarrow m$), with different masked confounders.

*$PM_{2.5}$ Components:* High-resolution ($1 \times 1$ km) gridded data on total $PM_{2.5}$ (Di et al., 2019) and its chemical composition (Amini et al., 2022), using annual averages for 2000. We focus on the

effect of sulfate on overall $PM_{2.5}$ ($SO_4 \rightarrow PM_{2.5}$), with key latent drivers such as *ammonium* ($NH_4$) and *organic carbon* (OC) masked.

These two datasets provide complementary perspectives: the first captures socioeconomic and demographic confounding, while the second reflects atmospheric chemistry. Additional datasets and hidden-confounder variants are described in Appendix D.

Table 1: Performance under *local confounding*. Results averaged over 10 runs with 95% confidence intervals. $r_d$: neighborhood radius in data generation; R: neighborhood radius used by the deconfounder. Lower values indicate less bias. Lower values for ATE and SPILL indicate less bias. $p$ indicates the $p$-value of the predictive check, with values near 0.5 indicating good model fit to 0.5.

| ENVIRONMENT | CONFOUNDER | METHOD | DIR | SPILL | $p$ |
|---|---|---|---|---|---|
| $PM_{2.5} \rightarrow m$ ($r_d = 1$) | $q_{\text{SUMMER}}$ | C-VAE-SPATIAL+ (R=1) | **0.04 ± 0.01** | **0.42 ± 0.08** | 0.37 ± 0.07 |
| | | C-VAE-SPATIAL+ (R=2) | **0.04 ± 0.01** | 0.44 ± 0.09 | 0.36 ± 0.04 |
| | | DAPSM | 0.30 ± 0.03 | N/A | N/A |
| | | GCNN | 0.41 ± 0.03 | N/A | N/A |
| | | S2SLS-LAG1 | 0.20 ± 0.00 | N/A | N/A |
| | | SPATIAL+ | 0.13 ± 0.04 | N/A | N/A |
| | | SPATIAL | 0.10 ± 0.07 | N/A | N/A |
| $PM_{2.5} \rightarrow m$ ($r_d = 2$) | $\rho_{\text{POP}}$ | C-VAE-SPATIAL+ (R=1) | 0.05 ± 0.02 | **0.15 ± 0.05** | 0.34 ± 0.04 |
| | | C-VAE-SPATIAL+ (R=2) | **0.04 ± 0.03** | 0.24 ± 0.06 | 0.35 ± 0.04 |
| | | DAPSM | 0.16 ± 0.01 | N/A | N/A |
| | | GCNN | 0.18 ± 0.03 | N/A | N/A |
| | | S2SLS-LAG1 | 0.07 ± 0.00 | N/A | N/A |
| | | SPATIAL+ | 0.10 ± 0.02 | N/A | N/A |
| | | SPATIAL | 0.17 ± 0.03 | N/A | N/A |
| $SO_4 \rightarrow PM_{2.5}$ ($r_d = 1$) | $NH_4$ | C-VAE-SPATIAL+ (R=1) | **0.07 ± 0.03** | 0.64 ± 0.10 | 0.38 ± 0.04 |
| | | C-VAE-SPATIAL+ (R=2) | **0.07 ± 0.03** | 0.16 ± 0.06 | 0.39 ± 0.06 |
| | | DAPSM | 1.44 ± 0.00 | N/A | N/A |
| | | GCNN | 0.52 ± 0.16 | N/A | N/A |
| | | S2SLS-LAG1 | 0.09 ± 0.00 | N/A | N/A |
| | | SPATIAL+ | 0.11 ± 0.03 | N/A | N/A |
| | | SPATIAL | 0.08 ± 0.02 | N/A | N/A |
| $SO_4 \rightarrow PM_{2.5}$ ($r_d = 2$) | $OC$ | C-VAE-SPATIAL+ (R=1) | **0.06 ± 0.03** | **0.18 ± 0.09** | 0.43 ± 0.03 |
| | | C-VAE-SPATIAL+ (R=2) | 0.12 ± 0.06 | 0.35 ± 0.08 | 0.43 ± 0.04 |
| | | DAPSM | 1.24 ± 0.01 | N/A | N/A |
| | | GCNN | 0.30 ± 0.10 | N/A | N/A |
| | | S2SLS-LAG1 | 0.21 ± 0.00 | N/A | N/A |
| | | SPATIAL+ | 0.13 ± 0.07 | N/A | N/A |
| | | SPATIAL | 0.29 ± 0.01 | N/A | N/A |

**Baselines and model variants.** We benchmark against classical and modern spatial methods: S2SLS (Anselin, 1988) with outcome autoregression; spline-based SPATIAL and residualized SPATIAL+ (Dupont et al., 2022); GCNN (Kipf, 2016) for non-linear neighbor aggregation; DAPSM (Papadogeorgou et al., 2019) for proximity-based matching; and UNET (Ronneberger et al., 2015), which can capture spillovers via neighbor treatments but does not adjust for hidden confounding.

For the *Spatial Deconfounder*, we instantiate the potential outcome module differently by setting the head to SPATIAL+ under local confounding (to ensure fairness) and to UNET under spatial confounding (to flexibly capture multi-scale structure). We also vary the neighborhood radius $r \in \{1, 2\}$ considered by the model and the latent confounder dimension in the C-VAE ($d_Z \in \{1, 2, 4, 8, 16, 32\}$).

**Evaluation metrics.** We assess performance on the direct (DIR) and spillover (SPILL) effects. As standard in causal inference (Hill, 2011; Shi et al., 2019; Cheng et al., 2022), we report standardized absolute bias, $\sigma_y^{-1} |\hat{\tau} - \tau|$, with true effect $\tau$, estimate $\hat{\tau}$, and outcome standard deviation $\sigma_y$.

**Results.** Tables 1 and 2 report performance under local and spatial confounding across different masked confounders (e.g., humidity, population density, ammonium, organic carbon). Across environments, the Spatial Deconfounder (C-VAE) variants consistently achieve lower bias on direct effects than existing spatial baselines. Even with non-smooth unobserved confounders like population density ($\rho_{\text{pop}}$), our framework still achieves lower bias. Importantly, unlike most benchmarks, both C-VAE and UNET can recover spillover effects, with C-VAE generally providing more accurate estimates. Using UNET as the outcome head further strengthens spillover estimation, highlighting the benefit of spatial architectures when paired with deconfounding.

Table 2: Performance under *spatial confounding*. Results averaged over 10 runs with 95% confidence intervals. $r_d$: neighborhood radius in data generation; R: neighborhood radius used by the deconfounder. Lower values indicate less bias. Lower values for ATE and SPILL indicate less bias. $p$ indicates the $p$-value of the predictive check, with values near 0.5 indicating good model fit to 0.5.

| ENVIRONMENT | CONFOUNDER | METHOD | DIR | SPILL | $p$ |
|---|---|---|---|---|---|
| $PM_{2.5} \rightarrow m$ ($r_d = 1$) | $\rho_{POP}$ | C-VAE-UNET (R=1) | $0.05 \pm 0.01$ | $0.22 \pm 0.06$ | $0.34 \pm 0.03$ |
| | | C-VAE-UNET (R=2) | $\mathbf{0.04 \pm 0.02}$ | $\mathbf{0.12 \pm 0.06}$ | $0.36 \pm 0.06$ |
| | | DAPSM | $0.20 \pm 0.01$ | N/A | N/A |
| | | GCNN | $0.17 \pm 0.06$ | N/A | N/A |
| | | S2SLS-LAG1 | $0.05 \pm 0.00$ | N/A | N/A |
| | | SPATIAL+ | $0.27 \pm 0.18$ | N/A | N/A |
| | | SPATIAL | $0.06 \pm 0.06$ | N/A | N/A |
| | | UNET | $0.06 \pm 0.01$ | $0.17 \pm 0.04$ | N/A |
| $SO_4 \rightarrow PM_{2.5}$ ($r_d = 1$) | $OC$ | C-VAE-UNET (R=1) | $0.06 \pm 0.02$ | $0.09 \pm 0.04$ | $0.44 \pm 0.03$ |
| | | C-VAE-UNET (R=2) | $0.06 \pm 0.02$ | $0.18 \pm 0.06$ | $0.45 \pm 0.03$ |
| | | DAPSM | $1.57 \pm 0.00$ | N/A | N/A |
| | | GCNN | $0.42 \pm 0.15$ | N/A | N/A |
| | | S2SLS-LAG1 | $0.13 \pm 0.00$ | N/A | N/A |
| | | SPATIAL+ | $0.06 \pm 0.05$ | N/A | N/A |
| | | SPATIAL | $\mathbf{0.04 \pm 0.01}$ | N/A | N/A |
| | | UNET | $0.07 \pm 0.02$ | $\mathbf{0.05 \pm 0.02}$ | N/A |

Additional experiments in Appendix D confirm these trends across broader settings. In a few cases where classical baselines perform comparably or slightly better, the scenarios involve very weak or extremely smooth confounding — conditions where stronger parametric assumptions may be advantageous. Overall, the results demonstrate that leveraging interference as a multi-cause signal yields substantial improvements in both direct and spillover effect estimation. These findings validate the core premise of the Spatial Deconfounder: interference can be exploited, rather than treated as a nuisance, to improve causal inference under unobserved confounding.

## 7 CONCLUSION

We introduce the **Spatial Deconfounder**, the first framework to jointly address interference and unobserved spatial confounding by treating neighborhood treatments as a multi-cause signal. A C-VAE with a spatial prior reconstructs a substitute confounder, enabling estimation of direct and spillover effects with flexible outcome models. We prove identification of these effects under mild assumptions on the latent spatial field and outcome structure.

Beyond methodological advances, our results highlight a conceptual shift: interference, often treated as a nuisance, can be exploited as a source of information about hidden structure. That said, our goal is not only conceptual but also practical: despite relying on idealized assumptions (latent-field sufficiency, substitute confounders, and a specific C-VAE instantiation), our semi-synthetic experiments on minimally modified environmental-health data show that the Spatial Deconfounder reduces bias relative to strong classical and deep-learning spatial baselines, providing empirical support for leveraging interference-driven multi-cause vectors together with a spatial latent-field representation in practice. This perspective opens the door to more robust causal inference in complex spatial systems, with future extensions to spatiotemporal data, continuous treatments, and large-scale applications. Discussion of broader impacts and the use of LLMs in the preparation of this paper is provided in Appendix F.

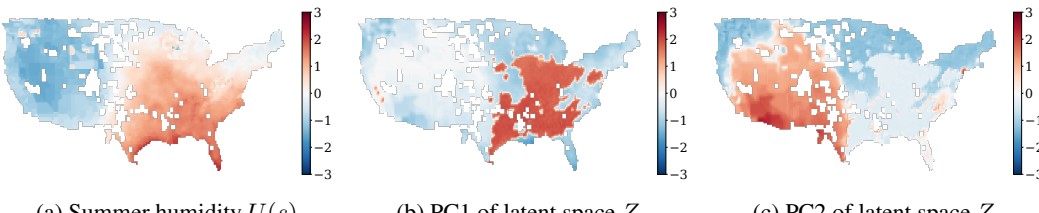

    (a) Summer humidity $U(s)$      (b) PC1 of latent space $Z_s$      (c) PC2 of latent space $Z_s$

Figure 4: Reconstructed latent confounder compared to the true (unobserved) spatial field. The leading principal component of $Z_s$ (PC1) captures the treatment, while the second principal component (PC2) recovers large-scale spatial structure of the true confounder.

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

## A EXTENDED LITERATURE REVIEW

The **Spatial Deconfounder** draws on three strands of prior work: (i) spatial causal inference under interference and spatially structured confounding, (ii) deconfounding methods for ATE estimation with unobserved confounders, and (iii) deep learning for spatial and latent structure modeling. We detail each in the sections that follow.

### A.1 SPATIAL CAUSAL INFERENCE UNDER INTERFERENCE AND SPATIALLY STRUCTURED CONFOUNDING

**Classical spatial causal inference.** Most estimators of direct and spillover effects assume that bias can be removed by conditioning on *observed* covariates (together with a specified exposure mapping or interference structure). Design-based work—grounded in exposure mappings, partial-interference designs, and randomization inference—derives estimators or hypothesis tests under known neighborhood or network structure (e.g., Hudgens and Halloran, 2008; Sobel, 2006; Aronow and Samii, 2017; Forastiere et al., 2021; Tchetgen Tchetgen et al., 2021). Model-based strategies then adjust for that structure while still relying on measured covariates or correct functional form: spatial autoregressive and two-stage least-squares estimators for spatial-lag/lagged-error models (Anselin, 1988), and spline/GAM or restricted spatial regression approaches that treat residual spatial trend as a nuisance to improve precision and approximate balance (e.g., Hanks et al., 2015). Deep graph/convolutional architectures can pool information across nearby units to improve prediction or imputation, but by themselves do not furnish identification without additional causal assumptions (Kipf, 2016). Domain-specific simulators (e.g., wildfire spread or atmospheric transport) encode spatial dependence through process-based physics and are often used as inputs to causal analyses, yet they typically still condition on observed drivers or require design-identifying assumptions (e.g. Larsen et al., 2022; Zigler et al., 2025). All of the above *presume exchangeability given observed covariates (or a valid design)*; if important spatial determinants of treatment and outcome are unmeasured, residual confounding bias can remain.

**Spatial confounding and bias-adjustment methods.** A growing literature tackles *unmeasured* spatial confounding directly. One family augments outcome models with latent spatial random effects (e.g., BYM/ICAR or GMRF priors) to soak up smooth hidden structure; this can reduce bias when the confounder is well captured by the basis, but may leave bias or distort fixed effects under misspecification (Rue and Held, 2005; Hodges and Reich, 2010). Restricted spatial regression and related orthogonalization schemes constrain the latent field away from covariates to mitigate bias (Hanks et al., 2015). Building on this idea, Dupont et al. (2022) (SPATIAL+) explicitly orthogonalizes spatial structure in the covariates from the outcome trend to purge bias from unmeasured *spatial* confounding. Propensity-score strategies that incorporate spatial proximity—such as distance-adjusted propensity score matching—aim to proxy smooth unmeasured confounders via geography (Papadogeorgou et al., 2019). Instrumental-variable designs exploit exogenous spatial shocks (e.g., wind direction, policy boundaries, thermal inversions) to identify causal effects despite hidden confounding, but require strong relevance/exclusion conditions that are difficult to validate under interference (e.g., Angrist et al., 1996; Imbens and Rubin, 2015; Deryugina et al., 2019). Finally, Bayesian frameworks that jointly model interference and latent spatial fields (e.g., Papadogeorgou and Samanta, 2023) achieve identification under specified priors and structural assumptions. In short, existing approaches either (i) assume smoothly varying latent fields or valid instruments or (ii) rely on strong parametric priors. None exploit interference patterns themselves as a *signal* for nonparametrically recovering the hidden confounder, nor do they aim to explicitly reconstruct the unobserved confounding process—a gap our Spatial Deconfounder addresses.

### A.2 DECONFOUNDING METHODS FOR ATE ESTIMATION WITH UNOBSERVED CONFOUNDERS

When confounders are unmeasured, point identification of causal effects generally fails. One approach is to derive bounds through sensitivity analysis (e.g., VanderWeele et al., 2015; Dorn et al., 2025; Oprescu et al., 2023; Frauen et al., 2023), trading identifiability for robustness. Another is the *deconfounder* framework, which fits a factor model to multiple causes in order to infer a substitute for the latent confounder, thereby restoring point identification (Wang and Blei, 2019; Bica et al.,

2020; Hatt and Feuerriegel, 2024). This stream is closest in spirit to our work: like us, it leverages multiplicity of treatments as a proxy for hidden structure. However, existing deconfounder methods require datasets with many simultaneous treatments (e.g., recommender systems, panel data) and assume no interference. Our approach resolves both limitations: interference itself naturally generates multiple-cause treatment vectors, enabling latent field recovery even with a single treatment type.

### A.3 DEEP LEARNING FOR SPATIAL AND LATENT STRUCTURE MODELING

**Deep learning for spatial modeling.** Modern deep architectures capture rich spatial structure but, on their own, remain predictive rather than identifying. U-Nets and encoder–decoder variants model multi-scale patterns on grids (Ronneberger et al., 2015; Oktay et al., 2018); graph neural networks extend to irregular domains (Kipf, 2016; Hamilton et al., 2017; Veličković et al., 2017); and patch-wise transformers model long-range dependencies on images and geospatial rasters (Dosovitskiy et al., 2020; Liu et al., 2021). Spatiotemporal extensions (e.g., ConvLSTM and graph/vision transformers) further capture dynamics (Shi et al., 2015). These tools provide flexible representations but require additional causal structure for identification.

**Deep latent-variable models.** Finally, conditional variational autoencoders (C-VAEs) and related deep generative models are widely used for representation learning with latent factors (Kingma and Welling, 2013; Sohn et al., 2015). Beyond C-VAEs, the broader family of latent-variable models includes variational autoencoders with structured priors (Rezende et al., 2014; Maaløe et al., 2016), disentangled representation learning (Higgins et al., 2017), normalizing flows (Rezende and Mohamed, 2015), and diffusion-based generative models (Ho et al., 2020; Kingma et al., 2021), all of which offer flexible ways to recover hidden structure from high-dimensional data. While these methods are not causal in themselves, they provide natural tools for reconstructing latent processes from observed multi-cause data. In our framework, a C-VAE combined with a spatial prior enables smooth, nonparametric recovery of a substitute confounder from local treatment vectors, which is then used for causal identification. Other architectures (e.g., diffusion models or flow-based methods) could, in principle, be substituted, but the key contribution lies in adapting deep latent-factor reconstruction to the spatial interference setting, where treatments on neighboring units jointly reveal the latent field.

### A.4 CAUSAL GENERATIVE MODELS

Recent work has proposed using expressive generative models as parameterizations of structural causal models. One stream of work uses autoregressive flows to obtain identifiable SCMs given a causal ordering (e.g., Javaloy et al., 2023; Khemakhem et al., 2021). Others combine diffusion- or GAN-based models with structural equations to model complex, high-dimensional counterfactuals (Sanchez and Tsaftaris, 2022; Kocaoglu et al., 2017). However, all of the methods assume unconfoundedness and are thus orthogonal to our Spatial Deconfounder. A different stream of literature combines causal inference and generative modeling under hidden confounding (e.g., Xia et al., 2021; Almodóvar et al., 2025). Similar to our work, the recently proposed DeCaFlow (Almodóvar et al., 2025) extends this line by learning confounded SCMs with causal normalizing flows and variational inference based on the deconfounder framework. However, these works are restricted to specific variables types, e.g., continuous treatments, and do not apply to the spatial setting. Building upon proxy variables, follow-up work on the deconfounder clarifies identifiability conditions in multi-cause settings (Wang and Blei, 2021). Similarly, this work assumes multiple treatments in an independent setting and does not apply to spatial causal inference tasks.

### A.5 DEEP IDENTIFIABLE MODELS AND NETWORK DECONFOUNDING

A complementary line of work focuses on identifiability in deep latent variable models. Sparse deep generative models establish identifiability of VAEs under sparsity constraints Moran et al. (2022), while Intact-VAE (Wu and Fukumizu, 2021) and $\beta$-Intact-VAE (Wu and Fukumizu, 2022) provide identifiable generative models for causal inference under unobserved confounding, IVs, proxies, and networked confounding. Applications to medical data show how identifiable VAEs can recover meaningful latent prognostic factors (Ma et al., 2023). These methods are typically designed for i.i.d. or network-structured observations and often rely on known adjacency structure, e.g.,

using neighbor information to help identify latent confounders in network deconfounding tasks. Our Spatial Deconfounder differs by targeting a specific spatial setting with localized grid-interference. More importantly, we note that our Spatial Deconfounder is not limited to the use of a C-VAE. The framework is model-agnostic and can be combined with other generative factor models. In contrast to these identifiable deep models, our focus is on a spatial–interference design: we show that interference-generated multi-cause vectors $(A_s, A_{\mathcal{N}_s})$, together with a spatial prior on $Z$, are sufficient to identify both direct and spillover effects without specifying a parametric latent-field model.

### A.6 OUR WORK

Our contribution lies at the intersection of spatial causal inference, methods for deconfounding under unobserved confounding, and modern deep latent-variable modeling. Existing approaches to spatial interference either assume that all relevant confounders are observed, or else mitigate bias through strong structural assumptions and priors—for example, by imposing smooth latent fields, leveraging restrictive IV conditions, or specifying parametric Bayesian models. In parallel, the "deconfounder" framework demonstrates that multiplicity of causes can be exploited to infer substitutes for unobserved confounders, thereby restoring point identification; however, these methods are designed for i.i.d. settings with many simultaneous treatments (e.g., recommender systems, panels), and do not naturally extend to spatial domains where interference and locality are intrinsic.

The *Spatial Deconfounder* closes this gap. We treat interference itself as the source of multi-cause information: treatment vectors on a unit and its neighbors contain precisely the dependence needed to reveal the hidden confounding field. By training a C-VAE with a spatial prior, we nonparametrically reconstruct a smooth latent confounder from these local treatment vectors. This substitute confounder can then be used to adjust for bias, enabling identification and estimation of both direct and spillover effects. Crucially, our method achieves this without committing to a fully specified latent-field model or relying on IV-style exclusion restrictions, thereby combining the flexibility of nonparametric deconfounding with the structural realities of spatial interference.

## B PROOFS

We first provide background by stating supporting definitions and lemmas. Then we prove our main theorem on the identifiability of the treatment effects.

### B.1 SUPPORTING LEMMAS AND DEFINITIONS

**Definition 1** (Ignorability). *The grid treatment $(a_s, \mathbf{a}_{\mathcal{N}_s})$ is* ignorable *given $Z_s, \mathbf{X}_s, \mathbf{X}_{\mathcal{N}_s}$, if for all $s = 1, \ldots, n$ and for all $(a, \mathbf{a}_{\mathcal{N}}) \in \mathcal{A}^{|\mathcal{S}|}$*

$$(A_s, \mathbf{A}_{\mathcal{N}_s}) \perp\!\!\!\perp Y_s(a, \mathbf{a}_{\mathcal{N}}) \mid Z_s, \mathbf{X}_s, \mathbf{X}_{\mathcal{N}_s}. \tag{21}$$

**Definition 2** (Factor models). *A factor model of the assigned spatial treatments is a latent-variable model*

$$p_\phi(z_{1:|\mathcal{S}|}, \boldsymbol{x}_{1:|\mathcal{S}|}, \boldsymbol{x}_{\mathcal{N}_{1:|\mathcal{S}|}}, a_{1:|\mathcal{S}|}, \mathbf{a}_{\mathcal{N}_{1:|\mathcal{S}|}}) \tag{22}$$

$$= p(z_{1:|\mathcal{S}|}, \boldsymbol{x}_{1:|\mathcal{S}|}, \boldsymbol{x}_{\mathcal{N}_{1:|\mathcal{S}|}}) \prod_{s=1}^{|\mathcal{S}|} p_\phi(a_s \mid z_s, \boldsymbol{x}_s, \boldsymbol{x}_{\mathcal{N}_s}) \prod_{k \in \mathcal{N}_s} p_\phi(a_k \mid z_s, \boldsymbol{x}_s, \boldsymbol{x}_{\mathcal{N}_s}) \tag{23}$$

*rendering the assigned treatments conditionally independent.*

**Lemma 1.** *For the relation between the substitute confounder and factor models, it holds under weak regularity conditions*

1. *Assume the true distributions of the treatments $p(a_{1:|\mathcal{S}|}, \mathbf{a}_{\mathcal{N}_{1:|\mathcal{S}|}})$ can be represented by a factor model employing the substitute confounder $Z$, i.e., $p_\phi(z_{1:|\mathcal{S}|}, \boldsymbol{x}_{1:|\mathcal{S}|}, \boldsymbol{x}_{\mathcal{N}_{1:|\mathcal{S}|}}, a_{1:|\mathcal{S}|}, \mathbf{a}_{\mathcal{N}_{1:|\mathcal{S}|}})$. With the assumption of latent field sufficiency (see Assumption 4), the assigned treatments $(a, \mathbf{a}_{\mathcal{N}})$ are ignorable given $Z_s$, $\mathbf{X}_s$, and $\mathbf{X}_{\mathcal{N}_s}$, i.e.,*

$$(A_s, \mathbf{A}_{\mathcal{N}_s}) \perp\!\!\!\perp Y_s(a, \mathbf{a}_{\mathcal{N}}) \mid Z_s, \mathbf{X}_s, \mathbf{X}_{\mathcal{N}_s}. \tag{24}$$

2. *A factor model that represents the distribution of the assigned treatments always exists.*

*Proof.* The statement follows from Proposition 5 in Wang and Blei (2019). □

### B.2 PROOF OF THE MAIN THEOREM

**Theorem 1** (Causal identifiability). *Suppose Assumptions 1–5 hold. Let $Z$ be a piecewise constant function of the assigned causes and covariates $(a, \mathbf{a}_{\mathcal{N}}, \boldsymbol{x}, \boldsymbol{x}_{\mathcal{N}})$ and let the outcome be a separable function of the observed and unobserved variables*

$$\mathbb{E}_Y\big[Y_s(a, \mathbf{a}_{\mathcal{N}}) \mid \mathbf{X}_s = \boldsymbol{x}, \mathbf{X}_{\mathcal{N}_s} = \boldsymbol{x}_{\mathcal{N}}, Z_s = z\big] = f_1(a, \mathbf{a}_{\mathcal{N}}, \boldsymbol{x}, \boldsymbol{x}_{\mathcal{N}}) + f_2(z), \tag{15}$$

$$\mathbb{E}_Y\big[Y_s \mid A_s = a, \mathbf{A}_{\mathcal{N}_s} = \mathbf{a}_{\mathcal{N}}, \mathbf{X}_s = \boldsymbol{x}, \mathbf{X}_{\mathcal{N}_s} = \boldsymbol{x}_{\mathcal{N}}, Z_s = z\big] = f_3(a, \mathbf{a}_{\mathcal{N}}, \boldsymbol{x}, \boldsymbol{x}_{\mathcal{N}}) + f_4(z), \tag{16}$$

*for continuously differentiable functions $f_1, f_2, f_3, f_4$. Consequently, the direct and spillover effects are identifiable as*

$$\tau_{\mathrm{dir}} = \mathbb{E}_{\mathbf{X}_s, \mathbf{X}_{\mathcal{N}_s}, Z}\Big[\mathbb{E}_Y\big[Y_s \mid A_s = 1, \mathbf{A}_{\mathcal{N}_s}, \mathbf{X}_s, \mathbf{X}_{\mathcal{N}_s}, Z_s\big] - \mathbb{E}_Y\big[Y_s \mid A_s = 0, \mathbf{A}_{\mathcal{N}_s}, \mathbf{X}_s, \mathbf{X}_{\mathcal{N}_s}, Z_s\big]\Big], \tag{17}$$

$$\tau_{\mathrm{spill}} = \mathbb{E}_{\mathbf{X}_s, \mathbf{X}_{\mathcal{N}_s}, Z}\Big[\mathbb{E}_Y\big[Y_s \mid a, \mathbf{A}_{\mathcal{N}_s} = \mathbf{a}_{\mathcal{N}_s}^{(1)}, \mathbf{X}_s, \mathbf{X}_{\mathcal{N}_s}, Z_s\big] - \mathbb{E}_Y\big[Y_s \mid a, \mathbf{A}_{\mathcal{N}_s} = \mathbf{a}_{\mathcal{N}_s}^{(0)}, \mathbf{X}_s, \mathbf{X}_{\mathcal{N}_s}, Z_s\big]\Big]. \tag{18}$$

*Proof.* First, observe that by the power-property and the separability of the outcome, we have

$$\mathbb{E}_Y[Y_s(a, \mathbf{a}_{\mathcal{N}})] = \mathbb{E}_{\mathbf{X}, \mathbf{X}_{\mathcal{N}}, Z}\big[\mathbb{E}_Y[Y_s(a, \mathbf{a}_{\mathcal{N}}) \mid \mathbf{X}_s, \mathbf{X}_{\mathcal{N}_s}, Z_s]\big] \tag{25}$$

$$= \mathbb{E}_{\mathbf{X}, \mathbf{X}_{\mathcal{N}}}[f_1(a, \mathbf{a}_{\mathcal{N}}, \mathbf{X}_s, \mathbf{X}_{\mathcal{N}_s})] + \mathbb{E}_Z[f_2(Z_s)]. \tag{26}$$

For the direct and indirect effects $\tau_{dir}$ and $\tau_{ind}$ follows

$$\tau_{dir} = \mathbb{E}_{\mathbf{X},\mathbf{X}_{\mathcal{N}}}[f_1(A_s = 1, \mathbf{a}_{\mathcal{N}_s}, \mathbf{X}_s, \mathbf{X}_{\mathcal{N}_s})] - \mathbb{E}_{\mathbf{X},\mathbf{X}_{\mathcal{N}}}[f_1(A_s = 0, \mathbf{a}_{\mathcal{N}_s}, \mathbf{X}_s, \mathbf{X}_{\mathcal{N}_s})] \tag{27}$$

$$= \int_{C(1,0)} \nabla_\nu \mathbb{E}_{\mathbf{X},\mathbf{X}_{\mathcal{N}}}[f_1(\nu, \mathbf{a}_{\mathcal{N}}, \mathbf{X}_s, \mathbf{X}_{\mathcal{N}_s})]d\nu, \quad \nu \in \mathbb{R} \tag{28}$$

and

$$\tau_{ind} = \mathbb{E}_{\mathbf{X},\mathbf{X}_{\mathcal{N}}}[f_1(a_s, \mathbf{A}_{\mathcal{N}_s} = \mathbf{a}_{\mathcal{N}_s}^{(1)}, \mathbf{X}_s, \mathbf{X}_{\mathcal{N}_s})] - \mathbb{E}_{\mathbf{X},\mathbf{X}_{\mathcal{N}}}[f_1(a_s, \mathbf{A}_{\mathcal{N}_s} = \mathbf{a}_{\mathcal{N}_s}^{(0)}, \mathbf{X}_s, \mathbf{X}_{\mathcal{N}_s})] \tag{29}$$

$$= \int_{C(a_{\mathcal{N}_s}^{(1)}, a_{\mathcal{N}_s}^{(0)}))} \nabla_\kappa \mathbb{E}_{\mathbf{X},\mathbf{X}_{\mathcal{N}}}[f_1(a_s, \mathbf{A}_{\mathcal{N}_s} = \kappa, \mathbf{X}_s, \mathbf{X}_{\mathcal{N}_s})]d\kappa, \quad \kappa \in \mathbb{R}^{|\mathcal{S}|-1}. \tag{30}$$

We thus need to find an expression for the gradient to rewrite the integral in terms of observable quantities.

To do so, we first consider the conditional expected outcome. By Assumption 5 there exists a function $g$ such that $Z = g(a, \mathbf{a}_{\mathcal{N}}, \mathbf{X}, \mathbf{X}_{\mathcal{N}})$. Therefore, it holds

$$\mathbb{E}_{\mathbf{X},\mathbf{X}_{\mathcal{N}},Z}\big[\mathbb{E}_Y[Y_s \mid A_s = a_s, \mathbf{A}_{\mathcal{N}_s} = \mathbf{a}_{\mathcal{N}_s}, \mathbf{X}_{\mathcal{N}_s}, Z_s]\big] \tag{31}$$

$$= \mathbb{E}_{\mathbf{X},\mathbf{X}_{\mathcal{N}}}\big[\mathbb{E}_Y[Y_s \mid A_s = a_s, \mathbf{A}_{\mathcal{N}_s} = \mathbf{a}_{\mathcal{N}_s}, \mathbf{X}_s, \mathbf{X}_{\mathcal{N}_s}, Z_s = g(a_s, \mathbf{a}_{\mathcal{N}_s}, \mathbf{X}_s, \mathbf{X}_{\mathcal{N}_s})] \tag{32}$$

$$= \mathbb{E}_{\mathbf{X},\mathbf{X}_{\mathcal{N}}}\big[\mathbb{E}_Y[Y_s(a_s, \mathbf{a}_{\mathcal{N}_s}) \mid A_s = a_s, \mathbf{A}_{\mathcal{N}_s} = \mathbf{a}_{\mathcal{N}_s}, \mathbf{X}_s, \mathbf{X}_{\mathcal{N}_s}, Z_s = g(a_s, \mathbf{a}_{\mathcal{N}_s}, \mathbf{X}_s, \mathbf{X}_{\mathcal{N}_s})]\big], \tag{33}$$

where the latter equality follows from Assumption 1.

As $Y_s(a_s, \mathbf{a}_{\mathcal{N}_s}) \perp\!\!\!\perp A_s, \mathbf{A}_{\mathcal{N}_s} \mid \mathbf{X}_s, \mathbf{X}_{\mathcal{N}_s}, Z_s$ (by Lemma 1) and the outcomes are assumed to be separable, it follows

$$\mathbb{E}_{\mathbf{X},\mathbf{X}_{\mathcal{N}},Z}\big[\mathbb{E}_Y[Y_s \mid A_s = a_s, \mathbf{A}_{\mathcal{N}_s} = \mathbf{a}_{\mathcal{N}_s}, \mathbf{X}_s, \mathbf{X}_{\mathcal{N}_s}, Z_s]\big] \tag{34}$$

$$= \mathbb{E}_{\mathbf{X},\mathbf{X}_{\mathcal{N}}}\big[\mathbb{E}_Y[Y_s(a_s, \mathbf{a}_{\mathcal{N}_s}) \mid \mathbf{X}_s, \mathbf{X}_{\mathcal{N}_s}, Z_s = g(a_s, \mathbf{a}_{\mathcal{N}_s}, \mathbf{X}_s, \mathbf{X}_{\mathcal{N}_s})]\big] \tag{35}$$

$$= \mathbb{E}_{\mathbf{X},\mathbf{X}_{\mathcal{N}}}[f_1(a_s, \mathbf{a}_{\mathcal{N}_s}, \mathbf{X}_s, \mathbf{X}_{\mathcal{N}_s})] + \mathbb{E}_Z[f_2(g(a_s, \mathbf{a}_{\mathcal{N}_s}, \mathbf{X}_s, \mathbf{X}_{\mathcal{N}_s}))]. \tag{36}$$

Recall that by the definition of the conditional expected outcome, we have

$$\mathbb{E}_{\mathbf{X},\mathbf{X}_{\mathcal{N}},Z}\big[\mathbb{E}_Y[Y_s \mid A_s = a_s, \mathbf{A}_{\mathcal{N}_s} = \mathbf{a}_{\mathcal{N}_s}, \mathbf{X}_s, \mathbf{X}_{\mathcal{N}_s}, Z_s]\big] = \tag{37}$$

$$\mathbb{E}_{\mathbf{X},\mathbf{X}_{\mathcal{N}}}[f_3(a_s, \mathbf{a}_{\mathcal{N}_s}, \mathbf{X}_s, \mathbf{X}_{\mathcal{N}_s})] + \mathbb{E}_Z[f_4(g(a_s, \mathbf{a}_{\mathcal{N}_s}, \mathbf{X}_s, \mathbf{X}_{\mathcal{N}_s}))]. \tag{38}$$

Now, we are ready to consider the gradients in 29. Observe that for the gradients of the conditional outcome, it holds

$$\nabla_{a_s}\mathbb{E}_{\mathbf{X},\mathbf{X}_{\mathcal{N}},Z}\big[\mathbb{E}_Y[Y_s \mid a_s, \mathbf{A}_{\mathcal{N}_s} = \mathbf{a}_{\mathcal{N}_s}, \mathbf{X}_s, \mathbf{X}_{\mathcal{N}_s}, Z_s]\big] \tag{39}$$

$$= \nabla_{a_s}\mathbb{E}_{\mathbf{X},\mathbf{X}_{\mathcal{N}}}[f_1(a_s, \mathbf{a}_{\mathcal{N}_s}, \mathbf{X}_s, \mathbf{X}_{\mathcal{N}_s})] + \nabla_{a_s}\mathbb{E}_Z[f_2(g(a_s, \mathbf{a}_{\mathcal{N}_s}))] \tag{40}$$

$$= \nabla_{a_s}\mathbb{E}_{\mathbf{X},\mathbf{X}_{\mathcal{N}}}[f_3(a_s, \mathbf{a}_{\mathcal{N}_s}, \mathbf{X}_s, \mathbf{X}_{\mathcal{N}_s})] + \nabla_{a_s}\mathbb{E}_Z[f_4(g(a_s, \mathbf{a}_{\mathcal{N}_s}))] \tag{41}$$

with a similar expression for $\nabla_{\mathbf{a}_{\mathcal{N}_s}}$. Note that, up to a set of Lebesgue measure zero, the gradients of $f_2$ and $f_4$ disappear, i.e.,

$$\nabla_{a_s}\mathbb{E}_Z[f_2(g(a_s, \mathbf{a}_{\mathcal{N}_s}, \mathbf{X}_s, \mathbf{X}_{\mathcal{N}_s}))] = \nabla_{g(a_s, \mathbf{a}_{\mathcal{N}_s}, \mathbf{X}_s, \mathbf{X}_{\mathcal{N}_s})}f_2 \nabla_{a_s}g(a_s, \mathbf{a}_{\mathcal{N}_s}, \mathbf{X}_s, \mathbf{X}_{\mathcal{N}_s}) = 0 \tag{42}$$

and

$$\nabla_{a_s}\mathbb{E}_Z[f_4(g(a_s, \mathbf{a}_{\mathcal{N}_s}, \mathbf{X}_s, \mathbf{X}_{\mathcal{N}_s}))] = \nabla_{g(a_s, \mathbf{a}_{\mathcal{N}_s}, \mathbf{X}_s, \mathbf{X}_{\mathcal{N}_s})}f_4 \nabla_{a_s}g(a_s, \mathbf{a}_{\mathcal{N}_s}, \mathbf{X}_s, \mathbf{X}_{\mathcal{N}_s}) = 0 \tag{43}$$

as

$$\nabla_{a_s}g(a_s, \mathbf{a}_{\mathcal{N}_s}, \mathbf{X}_s, \mathbf{X}_{\mathcal{N}_s}) = 0.$$

Similarly,

$$\nabla_{\mathbf{a}_{\mathcal{N}_s}}\mathbb{E}_Z[f_2(g(a_s, \mathbf{a}_{\mathcal{N}_s}, \mathbf{X}_s, \mathbf{X}_{\mathcal{N}_s}))] = \nabla_{\mathbf{a}_{\mathcal{N}_s}}\mathbb{E}_Z[f_4(g(a_s, \mathbf{a}_{\mathcal{N}_s}, \mathbf{X}_s, \mathbf{X}_{\mathcal{N}_s}))] = 0.$$

Overall, we receive

$$\nabla_{a_s}\mathbb{E}_{\mathbf{X},\mathbf{X}_{\mathcal{N}}}[f_1(a_s, \mathbf{a}_{\mathcal{N}_s}, \mathbf{X}_s, \mathbf{X}_{\mathcal{N}_s})] = \nabla_{a_s}\mathbb{E}_{\mathbf{X},\mathbf{X}_{\mathcal{N}}}[f_3(a_s, \mathbf{a}_{\mathcal{N}_s}, \mathbf{X}_s, \mathbf{X}_{\mathcal{N}_s})] \tag{44}$$

and

$$\nabla_{\mathbf{a}_{\mathcal{N}_s}}\mathbb{E}_{\mathbf{X},\mathbf{X}_{\mathcal{N}}}[f_1(a_s, \mathbf{a}_{\mathcal{N}_s}, \mathbf{X}_s, \mathbf{X}_{\mathcal{N}_s})] = \nabla_{\mathbf{a}_{\mathcal{N}_s}}\mathbb{E}_{\mathbf{X},\mathbf{X}_{\mathcal{N}}}[f_3(a_s, \mathbf{a}_{\mathcal{N}_s}, \mathbf{X}_s, \mathbf{X}_{\mathcal{N}_s})]. \tag{45}$$

Finally, we can identify the direct treatment $\tau_{dir}$ effect as

$$\tau_{dir} = \int_{C(1,0)} \nabla_\nu \mathbb{E}_{\mathbf{X},\mathbf{X}_\mathcal{N}}[f_1(\nu, \mathbf{a}_\mathcal{N}, \mathbf{X}_s, \mathbf{X}_{\mathcal{N}_s})]d\nu, \quad \nu \in \mathbb{R} \tag{46}$$

$$= \int_{C(1,0)} \nabla_\nu \mathbb{E}_{\mathbf{X},\mathbf{X}_\mathcal{N}}[f_3(\nu, \mathbf{a}_\mathcal{N}, \mathbf{X}_s, \mathbf{X}_{\mathcal{N}_s})]d\nu, \quad \nu \in \mathbb{R} \tag{47}$$

$$= \mathbb{E}_{\mathbf{X},\mathbf{X}_\mathcal{N}}[f_3(A_s = 1, \mathbf{a}_\mathcal{N}, \mathbf{X}_s, \mathbf{X}_{\mathcal{N}_s})] - \mathbb{E}_{\mathbf{X},\mathbf{X}_\mathcal{N}}[f_3(A_s = 0, \mathbf{a}_\mathcal{N}, \mathbf{X}_s, \mathbf{X}_{\mathcal{N}_s})] \tag{48}$$

$$= \mathbb{E}_{\mathbf{X},\mathbf{X}_\mathcal{N}}[f_3(A_s = 1, \mathbf{a}_\mathcal{N}, \mathbf{X}_s, \mathbf{X}_{\mathcal{N}_s})] + \mathbb{E}_Z[f_4(Z_s)] \tag{49}$$

$$\quad - \mathbb{E}_{\mathbf{X},\mathbf{X}_\mathcal{N}}[f_3(A_s = 0, \mathbf{a}_\mathcal{N}, \mathbf{X}_s, \mathbf{X}_{\mathcal{N}_s})] - \mathbb{E}_Z[f_4(Z_s)] \tag{50}$$

$$= \mathbb{E}_{Z,\mathbf{X},\mathbf{X}_\mathcal{N}}\Big[\mathbb{E}_Y\big[Y_s \mid a_s{=}1, \mathbf{a}_{\mathcal{N}_s}, \mathbf{X}_s, \mathbf{X}_{\mathcal{N}_s}, Z_s\big] - \mathbb{E}_Y\big[Y_s \mid a_s{=}0, \mathbf{a}_{\mathcal{N}_s}, \mathbf{X}_s, \mathbf{X}_{\mathcal{N}_s}, Z_s\big]\Big] \tag{51}$$

and similarly the indirect treatment effect $\tau_{ind}$ as

$$\tau_{ind} = \int_{C(a_{\mathcal{N}_s}^{(1)}, a_{\mathcal{N}_s}^{(0)})} \nabla_\kappa \mathbb{E}_{\mathbf{X},\mathbf{X}_\mathcal{N}}[f_1(a_s, \mathbf{A}_{\mathcal{N}_s} = \kappa, \mathbf{X}_s, \mathbf{X}_{\mathcal{N}_s})]d\kappa \tag{52}$$

$$= \int_{C(a_{\mathcal{N}_s}^{(1)}, a_{\mathcal{N}_s}^{(0)})} \nabla_\kappa \mathbb{E}_{\mathbf{X},\mathbf{X}_\mathcal{N}}[f_3(a_s, \mathbf{A}_{\mathcal{N}_s} = \kappa, \mathbf{X}_s, \mathbf{X}_{\mathcal{N}_s})]d\kappa \tag{53}$$

$$= \mathbb{E}_{\mathbf{X},\mathbf{X}_\mathcal{N}}[f_3(a_s, \mathbf{a}_{\mathcal{N}_s}^{(1)}, \mathbf{X}_s, \mathbf{X}_{\mathcal{N}_s})] - \mathbb{E}_{\mathbf{X},\mathbf{X}_\mathcal{N}}[f_3(a_s, a_{\mathcal{N}_s}^{(0)}, \mathbf{X}_s, \mathbf{X}_{\mathcal{N}_s})] \tag{54}$$

$$= \mathbb{E}_{\mathbf{X},\mathbf{X}_\mathcal{N}}[f_3(a_s, \mathbf{a}_{\mathcal{N}_s}^{(1)}, \mathbf{X}_s, \mathbf{X}_{\mathcal{N}_s})] + \mathbb{E}_Z[f_4(Z_s)] \tag{55}$$

$$\quad - \mathbb{E}_{\mathbf{X},\mathbf{X}_\mathcal{N}}[f_3(a_s, \mathbf{a}_{\mathcal{N}_s}^{(0)}, \mathbf{X}_s, \mathbf{X}_{\mathcal{N}_s})] - \mathbb{E}_Z[f_4(Z_s)] \tag{56}$$

$$= \mathbb{E}_{Z,\mathbf{X},\mathbf{X}_\mathcal{N}}\Big[\mathbb{E}_Y\big[Y_s \mid a_s, \mathbf{a}_{\mathcal{N}_s}^{(1)}, \mathbf{X}_s, \mathbf{X}_{\mathcal{N}_s}, Z_s\big] - \mathbb{E}_Y\big[Y_s \mid a_s, \mathbf{a}_{\mathcal{N}_s}^{(0)}, \mathbf{X}_s, \mathbf{X}_{\mathcal{N}_s}, Z_s\big]\Big] \tag{57}$$

Overall, we proved that the substitute confounder generated by our spatial deconfounder renders the treatment effects identifiable. $\quad\square$

# C    IMPLEMENTATION DETAILS

This section provides implementation details for our experimental setup. We cover four aspects:

1. **Semi-synthetic data generation:** construction of counterfactual outcomes under interference and spatial confounding using the `SpaCE` benchmark framework, with hidden confounders simulated by masking key covariates.
2. **Predictive model:** how the outcome model $f$ is estimated with ensembles of machine-learning models, including convolutional networks for spatial structure.
3. **Software and hyperparameters:** the AutoML framework used for training and tuning, along with default settings.
4. **Benchmarks:** implementation details for baseline methods.

**Semi-synthetic outcomes.** Recall from Section 6 that we construct counterfactual outcomes via

$$\hat{Y}_s = f(A_s, \mathbf{A}_{\mathcal{N}_s}, \mathbf{X}_s) + R_s \quad \text{or} \quad \hat{Y}_s = f(A_s, \mathbf{A}_{\mathcal{N}_s}, \mathbf{X}_s, \mathbf{X}_{\mathcal{N}_s}) + R_s,$$

where $f$ is a predictive model learned from real-world environmental data and $R_s$ are exogenous, spatially correlated residuals with the same distribution as the endogenous residuals.

**Predictive model with interference.** We estimate $f$ using ensembles of machine-learning models, with ensemble weights determined by predictive accuracy on held-out validation data. Following Tec et al. (2024) and the benchmarking guidelines of Curth et al. (2021), this avoids bias toward causal estimators tied to a single model class. To capture spatial structure, we include ResNet-18 (He et al., 2016) as one of the base learners. Training and hyperparameter tuning are automated with the `AutoGluon` Python package (Erickson et al., 2020), which performs model selection, hyperparameter search, and overfitting control with minimal human intervention. Default settings for `AutoGluon` are summarized in Table 3.

Table 3: Hyperparameters used in AutoML

| Parameter | Value |
|---|---|
| package | AutoGluon v1.4.0 |
| fit.presets | good_quality |
| fit.tuning_data | custom with algorithm 2 |
| fit.use_bag_holdout | true |
| fit.time_limit | null |
| feature_importance.time_limit | 900 |
| hyperparameters | get_hyperparameter_config('multimodal') |
| hyperparameters.AG_AUTOMM.optim.max_epochs | 10 |
| hyperparameters.AG_AUTOMM.model.timm_image.checkpoint_name | resnet18 |

**Spatially-aware train-validation split.** We implement a *spatially-aware* train-validation data split (Roberts et al., 2017) that takes interference into account to avoid overfitting due to spatial correlations. We only consider nodes with complete neighborhoods for training and validation. This spatial splitting strategy identifies a limited number of validation nodes and applies breadth-first search to exclude their adjacent neighbors from the training dataset. For this study, we define each grid cell to have edges connecting it to its 8 surrounding cells. This algorithm is described in algorithm 2.

**Synthetic Residual Generation.** Following the approach established in Tec et al. (2024), we generate synthetic residuals using a Gaussian Markov Random Field (GMRF) from a spatial graph. Specifically, we sample the synthetic residuals according to: $\boldsymbol{R} \sim_{\text{iid}} \text{MultivariateNormal}(\mathbf{0}, \hat{\lambda}(\mathbb{D} - \hat{\rho}\mathbb{A}\mathbb{D})^{-1})$, where $\mathbb{A}$ represents the spatial graph's adjacency matrix, $\mathbb{D}$ denotes a diagonal matrix containing the degree (number of neighbors) for each spatial location, $\hat{\rho}$ parameterizes the spatial dependence between observations and their neighbors (estimated from the true residuals obtained from $f$), and $\hat{\lambda}$ is calibrated to preserve the exact variance of the observed residuals. We refer the reader to Tec et al. (2024) for additional details.

**Benchmark Training and Hyperparameter Tuning.** To ensure a fair comparison, we use the RAY TUNE (Liaw et al., 2018) framework for hyperparameter tuning. For all but DAPSM, the tuning metric is implemented as mean-squared error (MSE) from a validation set obtained with the spatially-aware splitting method in algorithm 2. We use this splitting algorithm for computing the tuning metric since random splitting would result in extreme overfitting (Roberts et al., 2017). For DAPSM we

---

**Algorithm 2** Spatially-aware validation split selection with radius and complete neighborhoods

---

**Input:** Graph as map of neighbors $s \to \mathbb{N}_s$ where $\mathbb{N}_s \subset \mathbb{S}$ is the set of neighbors of $s$.

**Params:** Fraction $\alpha$ of seed validation points (default $\alpha = 0.02$); number of BFS levels $L$ to include in the validation set (default $L = 1$); buffer size $B$ indicating the number of BFS levels to leave outside training and validation (default $B = 1$); radius $r_m$ of the model to consider when determining the split (default $r_m = 1$)

**Output:** Set of training nodes $\mathbb{T} \subset \mathbb{S}$ and validation nodes $\mathbb{V} \subset \mathbb{S}$.

1: *# Helper function to check if node has complete r-hop neighborhood*
2: **function** HASCOMPLETENEIGHBORHOOD$(s, r)$:
3:    expected_count $= (2r + 1)^2$ *# For square grid*
4:    actual_neighbors $=$ GetNeighborsWithinRadius$(s, r)$
5:    **return** |actual_neighbors| $=$ expected_count
6: *# Filter to only nodes with complete neighborhoods*
7: $\mathbb{S}_{valid} = \{s \in \mathbb{S} : \text{HASCOMPLETENEIGHBORHOOD}(s, r_m)\}$
8: *# Initialize validation set with seed nodes from valid nodes only*
9: $\mathbb{V} =$ SampleWithoutReplacement$(\mathbb{S}_{valid}, \alpha)$
10: *# Expand validation set with neighbors*
11: **for** $\ell \in \{0, \ldots, L - 1\}$ **do**
12:    tmp $= \mathbb{V}$
13:    **for** $s \in$ tmp **do**
14:       $\mathbb{V} = \mathbb{V} \cup \mathbb{N}_s$
15:    **end for**
16: **end for**
17: *# Compute buffer*
18: $\mathbb{B} = \mathbb{V}$
19: **for** $b \in \{0, \ldots, B - 1 + r_m\}$ **do**
20:    tmp $= \mathbb{B}$
21:    **for** $s \in$ tmp **do**
22:       $\mathbb{B} = \mathbb{B} \cup \mathbb{N}_s$
23:    **end for**
24: **end for**
25: *# Exclude buffer for training set (from valid nodes only)*
26: $\mathbb{T} = \mathbb{S}_{valid} \setminus \mathbb{B}$
27: **return** $\mathbb{T}, \mathbb{V}$

---

use the covariate balance criterion following Papadogeorgou et al. (2019). After selecting the best hyperparameters, the method is retrained on the full data. Table 4 summarizes our hyperparameter search space for different baseline models. For C-VAE models with radius R evaluated on a dataset of radius $r_d$, training and validation are restricted to nodes with radius $r_m = \max(r_d, \text{R})$. Each C-VAE model also specifies a latent confounder dimension $d_Z \in \{1, 2, 4, 8, 16, 32\}$. The licenses of the data sources used for training are summarized in the supplement of Tec et al. (2024), which allow sharing and reuse for non-commercial purposes.

| Model | Iterations | Tuning Metric | Value |
|---|---|---|---|
| C-VAE-SPATIAL+ | 100 | weight_decay_C-VAE | loguniform between 1e-4 and 1e-3 |
| | | beta_max $(\beta)$ $(r_d = 1, PM_{2.5})$ | loguniform between 1e-8 and 10 |
| | | beta_max $(\beta)$ $(r_d = 1, SO_4)$ | loguniform between 1e-5 and 10 |
| | | beta_max $(\beta)$ $(r_d = 2)$ | loguniform between 1e-5 and 1e-4 |
| | | lam_t | loguniform between 1e-5 and 1.0 |
| | | lam_y | loguniform between 1e-5 and 1.0 |
| C-VAE-UNET | 60 | weight_decay_C-VAE | loguniform between 1e-4 and 1e-3 |
| | | beta_max $(\beta)$ | loguniform between 1e-3 and 1 |
| | | weight_decay_head | loguniform between 1e-4 and 1e-3 |
| | | unet_base_chan | 16 or 32 |
| DAPSM | N/A | propensity_score_penalty_value | choose from [0.001, 0.01, 0.1, 1.0] |
| | | propensity_score_penalty_type | l1 or l2 |
| | | spatial_weight | uniform between 0.0 and 1.0 |
| GCNN | N/A | hidden_dim | 16 or 32 |
| | | hidden_layers | 1 or 2 |
| | | weight_decay | loguniform between 1e-6 and 1e-1 |
| | | lr | 1e-3 or 3e-4 |
| | | epochs | 1000 or 2500 |
| | | dropout | loguniform between 1e-3 to 0.5 |
| SPATIAL+ | 2,500 | lam_t | loguniform between 1e-5 and 1.0 |
| | | lam_y | loguniform between 1e-5 and 1.0 |
| SPATIAL | 2,500 | lam | loguniform between 1e-5 and 1.0 |
| UNET | 50 | unet_base_chan | choose from [8, 16, 32] |

Table 4: Hyperparameter configurations evaluated for each model using a validation set. `Iterations` denotes the number of Ray Tune trials performed per model.

# D   FURTHER EXPERIMENTAL RESULTS

Our full experimental results are available for local confounding and spatial confounding at Table 5 and Table 6, respectively. There is a general pattern that C-VAE models tend to outperform benchmarks in estimating direct effects. In particular, C-VAE are the only local confounding methods that can also estimate spillover effects. In spatial confounding datasets with $r_d = 1$, deconfounders tend to have better direct effect and spillover estimation than UNET.

Table 5: Performance under *local confounding*. Results averaged over 10 runs with 95% confidence intervals. $r_d$: neighborhood radius in data generation; R: neighborhood radius used by the deconfounder. Lower values for ATE and SPILL indicate less bias. $p$ indicates the $p$-value of the predictive check, with values near 0.5 indicating good model fit to 0.5.

| Environment | Confounder | Method | DIR | SPILL | $p$ |
|---|---|---|---|---|---|
| $PM_{2.5} \to m$ $(r_d = 1)$ | $\rho_{pop}$ | C-VAE-SPATIAL+ (R=0) | 0.15 ± 0.11 | n/a | 0.36 ± 0.07 |
| | | C-VAE-SPATIAL+ (R=1) | 0.05 ± 0.02 | 0.34 ± 0.08 | 0.35 ± 0.09 |
| | | C-VAE-SPATIAL+ (R=2) | 0.07 ± 0.02 | 0.52 ± 0.08 | 0.35 ± 0.03 |
| | | DAPSM | 0.25 ± 0.01 | n/a | n/a |
| | | GCNN | 0.36 ± 0.03 | n/a | n/a |
| | | S2SLS-LAG1 | **0.03 ± 0.00** | n/a | n/a |
| | | SPATIAL+ | 0.13 ± 0.04 | n/a | n/a |
| | | SPATIAL | 0.10 ± 0.07 | n/a | n/a |
| | $q_{summer}$ | C-VAE-SPATIAL+ (R=0) | 0.15 ± 0.07 | n/a | 0.38 ± 0.08 |
| | | C-VAE-SPATIAL+ (R=1) | **0.04 ± 0.01** | **0.42 ± 0.08** | 0.37 ± 0.07 |
| | | C-VAE-SPATIAL+ (R=2) | 0.04 ± 0.01 | 0.44 ± 0.09 | 0.36 ± 0.04 |
| | | DAPSM | 0.30 ± 0.03 | n/a | n/a |
| | | GCNN | 0.41 ± 0.03 | n/a | n/a |
| | | S2SLS-LAG1 | 0.20 ± 0.00 | n/a | n/a |
| | | SPATIAL+ | 0.13 ± 0.04 | n/a | n/a |
| | | SPATIAL | 0.10 ± 0.07 | n/a | n/a |
| $PM_{2.5} \to m$ $(r_d = 2)$ | $\rho_{pop}$ | C-VAE-SPATIAL+ (R=0) | 0.11 ± 0.02 | n/a | 0.35 ± 0.03 |
| | | C-VAE-SPATIAL+ (R=1) | 0.05 ± 0.02 | 0.15 ± 0.05 | 0.34 ± 0.04 |
| | | C-VAE-SPATIAL+ (R=2) | **0.04 ± 0.03** | 0.24 ± 0.06 | 0.35 ± 0.04 |
| | | DAPSM | 0.16 ± 0.01 | n/a | n/a |
| | | GCNN | 0.18 ± 0.03 | n/a | n/a |
| | | S2SLS-LAG1 | 0.07 ± 0.00 | n/a | n/a |
| | | SPATIAL+ | 0.10 ± 0.02 | n/a | n/a |

| Environment | Confounder | Method | ATE | SPILL | $p$ |
|---|---|---|---|---|---|
| | | SPATIAL | 0.17 ± 0.03 | n/a | n/a |
| | $q_{summer}$ | C-VAE-SPATIAL+ (R=0) | 0.13 ± 0.05 | n/a | 0.36 ± 0.04 |
| | | C-VAE-SPATIAL+ (R=1) | **0.04 ± 0.02** | **0.11 ± 0.05** | 0.36 ± 0.04 |
| | | C-VAE-SPATIAL+ (R=2) | 0.07 ± 0.02 | 0.19 ± 0.06 | 0.36 ± 0.04 |
| | | DAPSM | 0.20 ± 0.01 | n/a | n/a |
| | | GCNN | 0.16 ± 0.05 | n/a | n/a |
| | | S2SLS-LAG1 | 0.09 ± 0.00 | n/a | n/a |
| | | SPATIAL+ | 0.11 ± 0.02 | n/a | n/a |
| | | SPATIAL | 0.17 ± 0.03 | n/a | n/a |
| $SO_4 \to PM_{2.5}$ ($r_d = 1$) | $NH_4$ | C-VAE-SPATIAL+ (R=0) | 0.22 ± 0.04 | n/a | 0.40 ± 0.05 |
| | | C-VAE-SPATIAL+ (R=1) | **0.07 ± 0.03** | **0.64 ± 0.10** | 0.38 ± 0.04 |
| | | C-VAE-SPATIAL+ (R=2) | **0.07 ± 0.03** | 0.16 ± 0.06 | 0.39 ± 0.06 |
| | | DAPSM | 1.44 ± 0.00 | n/a | n/a |
| | | GCNN | 0.52 ± 0.16 | n/a | n/a |
| | | S2SLS-LAG1 | 0.09 ± 0.00 | n/a | n/a |
| | | SPATIAL+ | 0.11 ± 0.03 | n/a | n/a |
| | | SPATIAL | 0.08 ± 0.02 | n/a | n/a |
| | $OC$ | C-VAE-SPATIAL+ (R=0) | 0.07 ± 0.03 | n/a | 0.41 ± 0.02 |
| | | C-VAE-SPATIAL+ (R=1) | 0.08 ± 0.03 | **0.69 ± 0.10** | 0.41 ± 0.03 |
| | | C-VAE-SPATIAL+ (R=2) | 0.11 ± 0.04 | 0.90 ± 0.12 | 0.44 ± 0.02 |
| | | DAPSM | 1.45 ± 0.00 | n/a | n/a |
| | | GCNN | 0.77 ± 0.22 | n/a | n/a |
| | | S2SLS-LAG1 | **0.00 ± 0.00** | n/a | n/a |
| | | SPATIAL+ | 0.11 ± 0.03 | n/a | n/a |
| | | SPATIAL | 0.08 ± 0.02 | n/a | n/a |
| $SO_4 \to PM_{2.5}$ ($r_d = 2$) | $NH_4$ | C-VAE-SPATIAL+ (R=0) | **0.07 ± 0.04** | n/a | 0.48 ± 0.06 |
| | | C-VAE-SPATIAL+ (R=1) | 0.08 ± 0.03 | 0.13 ± 0.05 | 0.44 ± 0.03 |
| | | C-VAE-SPATIAL+ (R=2) | 0.12 ± 0.04 | **0.09 ± 0.04** | 0.43 ± 0.03 |
| | | DAPSM | 1.23 ± 0.00 | n/a | n/a |
| | | GCNN | 0.26 ± 0.09 | n/a | n/a |
| | | S2SLS-LAG1 | 0.10 ± 0.00 | n/a | n/a |
| | | SPATIAL+ | 0.13 ± 0.07 | n/a | n/a |
| | | SPATIAL | 0.29 ± 0.01 | n/a | n/a |
| | $OC$ | C-VAE-SPATIAL+ (R=0) | 0.10 ± 0.07 | n/a | 0.43 ± 0.04 |
| | | C-VAE-SPATIAL+ (R=1) | **0.06 ± 0.03** | **0.18 ± 0.09** | 0.43 ± 0.03 |
| | | C-VAE-SPATIAL+ (R=2) | 0.12 ± 0.06 | 0.35 ± 08 | 0.43 ± 0.04 |
| | | DAPSM | 1.24 ± 0.01 | n/a | n/a |
| | | GCNN | 0.30 ± 0.10 | n/a | n/a |
| | | S2SLS-LAG1 | 0.21 ± 0.00 | n/a | n/a |
| | | SPATIAL+ | 0.13 ± 0.07 | n/a | n/a |
| | | SPATIAL | 0.29 ± 0.01 | n/a | n/a |

Table 6: Performance under *spatial confounding*. Results averaged over 10 runs with 95% confidence intervals. $r_d$: neighborhood radius in data generation; R: neighborhood radius used by the deconfounder. Lower values for ATE and SPILL indicate less bias. $p$ indicates the $p$-value of the predictive check, with values near 0.5 indicating good model fit to 0.5.

| Environment | Confounder | Method | DIR | SPILL | $p$ |
|---|---|---|---|---|---|
| $PM_{2.5} \to m$ ($r_d = 1$) | $\rho_{pop}$ | C-VAE-UNET (R=0) | 0.11 ± 0.04 | n/a | 0.34 ± 0.04 |
| | | C-VAE-UNET (R=1) | 0.05 ± 0.01 | 0.22 ± 0.06 | 0.34 ± 0.03 |
| | | C-VAE-UNET (R=2) | **0.04 ± 0.02** | **0.12 ± 0.06** | 0.36 ± 0.06 |
| | | DAPSM | 0.20 ± 0.01 | n/a | n/a |
| | | GCNN | 0.17 ± 0.06 | n/a | n/a |
| | | S2SLS-LAG1 | 0.05 ± 0.00 | n/a | n/a |
| | | SPATIAL+ | 0.27 ± 0.18 | n/a | n/a |
| | | SPATIAL | 0.06 ± 0.06 | n/a | n/a |
| | | UNET | 0.06 ± 0.01 | 0.17 ± 0.04 | n/a |
| | $q_{summer}$ | C-VAE-UNET (R=0) | **0.04 ± 0.02** | n/a | 0.35 ± 0.02 |
| | | C-VAE-UNET (R=1) | 0.06 ± 0.02 | 0.13 ± 0.07 | 0.33 ± 0.02 |
| | | C-VAE-UNET (R=2) | **0.04 ± 0.02** | 0.10 ± 0.05 | 0.36 ± 0.05 |
| | | DAPSM | 0.28 ± 0.04 | n/a | n/a |
| | | GCNN | 0.23 ± 0.03 | n/a | n/a |
| | | S2SLS-LAG1 | 0.16 ± 0.00 | n/a | n/a |
| | | SPATIAL+ | 0.27 ± 0.18 | n/a | n/a |
| | | SPATIAL | 0.07 ± 0.06 | n/a | n/a |
| | | UNET | **0.04 ± 0.01** | **0.10 ± 0.05** | n/a |
| $PM_{2.5} \to m$ ($r_d = 2$) | $\rho_{pop}$ | C-VAE-UNET (R=0) | 0.09 ± 0.03 | n/a | 0.32 ± 0.04 |
| | | C-VAE-UNET (R=1) | 0.15 ± 0.01 | **0.09 ± 0.03** | 0.31 ± 0.04 |
| | | C-VAE-UNET (R=2) | 0.15 ± 0.01 | 0.13 ± 0.05 | 0.29 ± 0.06 |
| | | DAPSM | 0.15 ± 0.02 | n/a | n/a |
| | | GCNN | 0.15 ± 0.04 | n/a | n/a |
| | | S2SLS-LAG1 | 0.06 ± 0.00 | n/a | n/a |
| | | SPATIAL+ | 0.08 ± 0.04 | n/a | n/a |
| | | SPATIAL | **0.05 ± 0.02** | n/a | n/a |

| | | | | | |
|---|---|---|---|---|---|
| | | UNET | $0.15 \pm 0.01$ | $0.15 \pm 0.03$ | n/a |
| | $q_{\text{summer}}$ | C-VAE-UNET (R=0) | **0.05 ± 0.01** | n/a | 0.30 ± 0.05 |
| | | C-VAE-UNET (R=1) | $0.14 \pm 0.01$ | $0.07 \pm 0.03$ | 0.30 ± 0.05 |
| | | C-VAE-UNET (R=2) | $0.15 \pm 0.01$ | **0.06 ± 0.03** | 0.33 ± 0.04 |
| | | DAPSM | $0.21 \pm 0.01$ | n/a | n/a |
| | | GCNN | $0.23 \pm 0.03$ | n/a | n/a |
| | | S2SLS-LAG1 | $0.10 \pm 0.00$ | n/a | n/a |
| | | SPATIAL+ | $0.07 \pm 0.03$ | n/a | n/a |
| | | SPATIAL | **0.05 ± 0.02** | n/a | n/a |
| | | UNET | $0.15 \pm 0.00$ | $0.08 \pm 0.04$ | n/a |
| $SO_4 \rightarrow PM_{2.5}$ $(r_d = 1)$ | $NH\_4$ | C-VAE-UNET (R=0) | $0.18 \pm 0.03$ | n/a | 0.44 ± 0.03 |
| | | C-VAE-UNET (R=1) | $0.05 \pm 0.02$ | $0.22 \pm 0.03$ | 0.45 ± 0.03 |
| | | C-VAE-UNET (R=2) | **0.04 ± 0.02** | $0.37 \pm 0.06$ | 0.43 ± 0.03 |
| | | DAPSM | $1.56 \pm 0.00$ | n/a | n/a |
| | | GCNN | $0.55 \pm 0.09$ | n/a | n/a |
| | | S2SLS-LAG1 | $0.22 \pm 0.00$ | n/a | n/a |
| | | SPATIAL+ | $0.06 \pm 0.05$ | n/a | n/a |
| | | SPATIAL | **0.04 ± 0.01** | n/a | n/a |
| | | UNET | **0.04 ± 0.01** | **0.19 ± 0.04** | n/a |
| | $OC$ | C-VAE-UNET (R=0) | **0.04 ± 0.02** | n/a | 0.46 ± 0.02 |
| | | C-VAE-UNET (R=1) | $0.06 \pm 0.02$ | $0.09 \pm 0.04$ | 0.44 ± 0.03 |
| | | C-VAE-UNET (R=2) | $0.06 \pm 0.02$ | $0.18 \pm 0.06$ | 0.45 ± 0.03 |
| | | DAPSM | $1.57 \pm 0.00$ | n/a | n/a |
| | | GCNN | $0.42 \pm 0.15$ | n/a | n/a |
| | | S2SLS-LAG1 | $0.13 \pm 0.00$ | n/a | n/a |
| | | SPATIAL+ | $0.06 \pm 0.05$ | n/a | n/a |
| | | SPATIAL | **0.04 ± 0.01** | n/a | n/a |
| | | UNET | $0.07 \pm 0.02$ | **0.05 ± 0.02** | n/a |
| $SO_4 \rightarrow PM_{2.5}$ $(r_d = 2)$ | $NH\_4$ | C-VAE-UNET (R=0) | **0.04 ± 0.02** | n/a | 0.43 ± 0.04 |
| | | C-VAE-UNET (R=1) | $0.13 \pm 0.02$ | **0.05 ± 0.02** | 0.45 ± 0.03 |
| | | C-VAE-UNET (R=2) | $0.15 \pm 0.01$ | $0.07 \pm 0.03$ | 0.45 ± 0.03 |
| | | DAPSM | $1.47 \pm 0.00$ | n/a | n/a |
| | | GCNN | $0.66 \pm 0.21$ | n/a | n/a |
| | | S2SLS-LAG1 | $0.16 \pm 0.00$ | n/a | n/a |
| | | SPATIAL+ | $0.06 \pm 0.02$ | n/a | n/a |
| | | SPATIAL | $0.06 \pm 0.05$ | n/a | n/a |
| | | UNET | $0.15 \pm 0.01$ | $0.11 \pm 0.04$ | n/a |
| | $OC$ | C-VAE-UNET (R=0) | **0.04 ± 0.02** | n/a | 0.43 ± 0.02 |
| | | C-VAE-UNET (R=1) | $0.12 \pm 0.02$ | **0.06 ± 0.03** | 0.43 ± 0.03 |
| | | C-VAE-UNET (R=2) | $0.13 \pm 0.03$ | $0.07 \pm 0.02$ | 0.44 ± 0.03 |
| | | DAPSM | $1.49 \pm 0.01$ | n/a | n/a |
| | | GCNN | $0.67 \pm 0.12$ | n/a | n/a |
| | | S2SLS-LAG1 | $0.09 \pm 0.00$ | n/a | n/a |
| | | SPATIAL+ | $0.05 \pm 0.02$ | n/a | n/a |
| | | SPATIAL | $0.06 \pm 0.05$ | n/a | n/a |
| | | UNET | $0.15 \pm 0.01$ | $0.08 \pm 0.04$ | n/a |

# E ADDITIONAL ROBUSTNESS TESTS

## E.1 TREATMENT SPARSITY

The results in Table 7 examine our method under sparse treatment conditions with 30% and 10% of grid cells receiving treatment. Despite similar performance under moderate treatment sparsity (30%), C-VAE-SPATIAL+ considerably outperforms SPATIAL+ when sparsity is extreme (10%), underscoring the value of our framework for direct effect estimation in highly sparse conditions. In addition, the predictive $p$-value is lower as treatment sparsity increases, showing worse model calibration in sparse settings.

Table 7: Performance under **sparse** *local confounding*. Results averaged over 10 runs with 95% confidence intervals. $r_d$: neighborhood radius in data generation; R: neighborhood radius used by the deconfounder. Lower values for ATE and SPILL indicate less bias. $p$ indicates the predictive $p$-value, with values near 0.5 indicating good model fit to 0.5. Percentage in environment denotes the fraction of observations receiving treatment.

| Environment | Confounder | Method | DIR | SPILL | $p$ |
|---|---|---|---|---|---|
| $SO_4 \rightarrow PM_{2.5}$ ($r_d = 1$) (10%) | $NH\_4$ | C-VAE-SPATIAL+ (R=0) | $0.07 \pm 0.04$ | n/a | $0.28 \pm 0.02$ |
| | | C-VAE-SPATIAL+ (R=1) | $0.19 \pm 0.08$ | $\mathbf{0.80 \pm 0.23}$ | $0.28 \pm 0.01$ |
| | | C-VAE-SPATIAL+ (R=2) | $0.14 \pm 0.08$ | $1.20 \pm 0.12$ | $0.29 \pm 0.01$ |
| | | DAPSM | $\mathbf{0.02 \pm 0.00}$ | n/a | n/a |
| | | GCNN | $0.42 \pm 0.07$ | n/a | n/a |
| | | S2SLS-LAG1 | $0.04 \pm 0.00$ | n/a | n/a |
| | | SPATIAL+ | $0.68 \pm 0.21$ | n/a | n/a |
| | | SPATIAL | $0.17 \pm 0.11$ | n/a | n/a |
| | $OC$ | C-VAE-SPATIAL+ (R=0) | $\mathbf{0.05 \pm 0.03}$ | n/a | $0.27 \pm 0.01$ |
| | | C-VAE-SPATIAL+ (R=1) | $0.14 \pm 0.05$ | $\mathbf{0.71 \pm 0.17}$ | $0.29 \pm 0.02$ |
| | | C-VAE-SPATIAL+ (R=2) | $0.08 \pm 0.03$ | $1.09 \pm 0.18$ | $0.30 \pm 0.02$ |
| | | DAPSM | $\mathbf{0.05 \pm 0.02}$ | n/a | n/a |
| | | GCNN | $0.69 \pm 0.20$ | n/a | n/a |
| | | S2SLS-LAG1 | $0.26 \pm 0.00$ | n/a | n/a |
| | | SPATIAL+ | $0.55 \pm 0.19$ | n/a | n/a |
| | | SPATIAL | $0.17 \pm 0.11$ | n/a | n/a |
| $SO_4 \rightarrow PM_{2.5}$ ($r_d = 1$) (30%) | $NH\_4$ | C-VAE-SPATIAL+ (R=0) | $0.14 \pm 0.03$ | n/a | $0.33 \pm 0.02$ |
| | | C-VAE-SPATIAL+ (R=1) | $0.18 \pm 0.06$ | $0.42 \pm 0.11$ | $0.35 \pm 0.03$ |
| | | C-VAE-SPATIAL+ (R=2) | $0.12 \pm 0.07$ | $\mathbf{0.25 \pm 0.11}$ | $0.34 \pm 0.02$ |
| | | DAPSM | $1.00 \pm 0.00$ | n/a | n/a |
| | | GCNN | $0.34 \pm 0.12$ | n/a | n/a |
| | | S2SLS-LAG1 | $\mathbf{0.03 \pm 0.00}$ | n/a | n/a |
| | | SPATIAL+ | $0.12 \pm 0.05$ | n/a | n/a |
| | | SPATIAL | $0.16 \pm 0.03$ | n/a | n/a |
| | $OC$ | C-VAE-SPATIAL+ (R=0) | $0.13 \pm 0.03$ | n/a | $0.31 \pm 0.03$ |
| | | C-VAE-SPATIAL+ (R=1) | $0.15 \pm 0.06$ | $0.35 \pm 0.09$ | $0.35 \pm 0.02$ |
| | | C-VAE-SPATIAL+ (R=2) | $0.11 \pm 0.05$ | $\mathbf{0.27 \pm 0.10}$ | $0.36 \pm 0.03$ |
| | | DAPSM | $1.00 \pm 0.00$ | n/a | n/a |
| | | GCNN | $0.35 \pm 0.14$ | n/a | n/a |
| | | S2SLS-LAG1 | $\mathbf{0.07 \pm 0.00}$ | n/a | n/a |
| | | SPATIAL+ | $0.12 \pm 0.05$ | n/a | n/a |
| | | SPATIAL | $0.15 \pm 0.03$ | n/a | n/a |

## E.2 PERFORMANCE UNDER SINGLE-CAUSE CONFOUNDERS

We evaluate our method under violation of Assumption 4 by introducing a localized single-cause unobserved confounder named $SC$. We select $\mathcal{C} = \{c_1, \ldots, c_n\}$ as cluster centers, drawn uniformly from the set of spatial sites, where $n = \lceil s|\mathcal{S}| \rceil$ and $s$ denotes the sparsity. Each cluster center is assigned a peak intensity $\alpha_c \sim U(0.5, 1.0)$. for any site $s$, the resulting single-cause confounder is

$$SC_s = \max_{c \in \mathcal{C}} \alpha_c \exp\left(-\frac{d(s,c)}{2}\right)$$

where $d(s, c)$ is the shortest distance path between $s$ and $c$. We then inject $SC$ into both the treatment and outcome by adding $0.8 \times \text{std}(X) \times SC$ to each variable where $X$ denotes the respective treatment or outcome variable. The treatments are binarized by applying a threshold. Table 8 presents the performance of our methods when Assumption 4 is violated. When the unobserved

confounder exhibits greater localization (10%), C-VAE-SPATIAL+ shows larger bias in the direct effect estimate compared to SPATIAL+. However, with a moderately sparse unobserved confounder, C-VAE-SPATIAL+ achives comparable performance to SPATIAL+.

Table 8: Performance under *local confounding* with **single-cause unobserved confounder** $SC$. Results averaged over 10 runs with 95% confidence intervals. $r_d$: neighborhood radius in data generation; R: neighborhood radius used by the deconfounder. Lower values for ATE and SPILL indicate less bias. $p$ indicates the predictive $p$-value, with values near 0.5 indicating good model fit to 0.5. Percentage in environment denotes the fraction of observations receiving treatment.

| Environment | Confounder | Method | DIR | SPILL | $p$ |
|---|---|---|---|---|---|
| $PM_{2.5} \rightarrow m \ (r_d = 1) \ (10\%)$ | $SC$ | C-VAE-SPATIAL+ (R=0) | $0.11 \pm 0.08$ | n/a | $0.40 \pm 0.02$ |
| | | C-VAE-SPATIAL+ (R=1) | $0.11 \pm 0.06$ | $\mathbf{0.44 \pm 0.14}$ | $0.40 \pm 0.02$ |
| | | C-VAE-SPATIAL+ (R=2) | $0.08 \pm 0.02$ | $0.62 \pm 0.07$ | $0.40 \pm 0.03$ |
| | | DAPSM | $0.52 \pm 0.01$ | n/a | n/a |
| | | GCNN | $0.13 \pm 0.03$ | n/a | n/a |
| | | S2SLS-LAG1 | $0.20 \pm 0.00$ | n/a | n/a |
| | | SPATIAL+ | $\mathbf{0.04 \pm 0.01}$ | n/a | n/a |
| | | SPATIAL | $0.06 \pm 0.07$ | n/a | n/a |
| $PM_{2.5} \rightarrow m \ (r_d = 1) \ (30\%)$ | $SC$ | C-VAE-SPATIAL+ (R=0) | $\mathbf{0.07 \pm 0.02}$ | n/a | $0.38 \pm 0.02$ |
| | | C-VAE-SPATIAL+ (R=1) | $0.08 \pm 0.02$ | $\mathbf{0.26 \pm 0.07}$ | $0.39 \pm 0.03$ |
| | | C-VAE-SPATIAL+ (R=2) | $0.10 \pm 0.04$ | $1.14 \pm 1.37$ | $0.42 \pm 0.05$ |
| | | DAPSM | $0.58 \pm 0.00$ | n/a | n/a |
| | | GCNN | $0.16 \pm 0.05$ | n/a | n/a |
| | | S2SLS-LAG1 | $0.23 \pm 0.00$ | n/a | n/a |
| | | SPATIAL+ | $0.09 \pm 0.01$ | n/a | n/a |
| | | SPATIAL | $0.08 \pm 0.02$ | n/a | n/a |

## E.3 SENSITIVITY TO HYPERPARAMETERS AND SPILLOVER RADIUS

**Hyperparameters:** To assess the robustness of our spatial deconfounder across different hyperparameter sets, we conduct a sensitivity analysis. Below, we provide figures that display how the hyperparameters of C-VAE-SPATIAL+ and C-VAE-UNET affect the estimation performance. Specifically, we assess the hyperparameters $\beta$ (KL term), the latent dimension $d_Z$, the learning rate, and weight decay. We observe the change in one parameter at a time, while optimizing the other hyperparameters conditional on the assessed parameter.

For C-VAE-SPATIAL+, we do not observe a consistent pattern in the error for the direct effect DIR. The estimation performance remains robust when changing a single hyperparameter while optimizing all others. For the spillover effect SPILL estimation, we generally observe that the error increases with $\beta$ but decreases as $d_Z$ grows. In our models, the optimal $\beta$ and $d_Z$ are determined through hyperparameter tuning on the MSE Loss. Datasets with large $r_d$ typically need low $\beta$ because the smoothness is lower. On the other hand, datasets with small $r_d$ need a higher $\beta$ to enforce smoothness constraints. Furthermore, the optimal value of $\beta$ depends on the nature of the unobserved confounder. For instance, models with a smooth confounder such as humidity $q_{summer}$ favor a larger $\beta$, whereas models with an anisotropic confounder like the population density $\rho_{pop}$ require a relatively smaller $\beta$. For C-VAE-UNET, the direct effect DIR and spillover effect SPILL remain consistent across varying degrees of hyperparameters, highlighting the important consistency with deep learning spatially-aware architectures.

**Neighborhood radius:** Furthermore, we assess the robustness of our spatial deconfounder with respect to different interference radii in Figures 5 to 20 for C-VAE-SPATIAL+ and Figures 21 to 36 for C-VAE-UNET. We observe that our spatial deconfounder is generally robust to misspecification of the interference radius. Note that we do not include $r = 0$ models in SPILL plots, as these models cannot include neighboring treatments, i.e., spillover effect, by design.

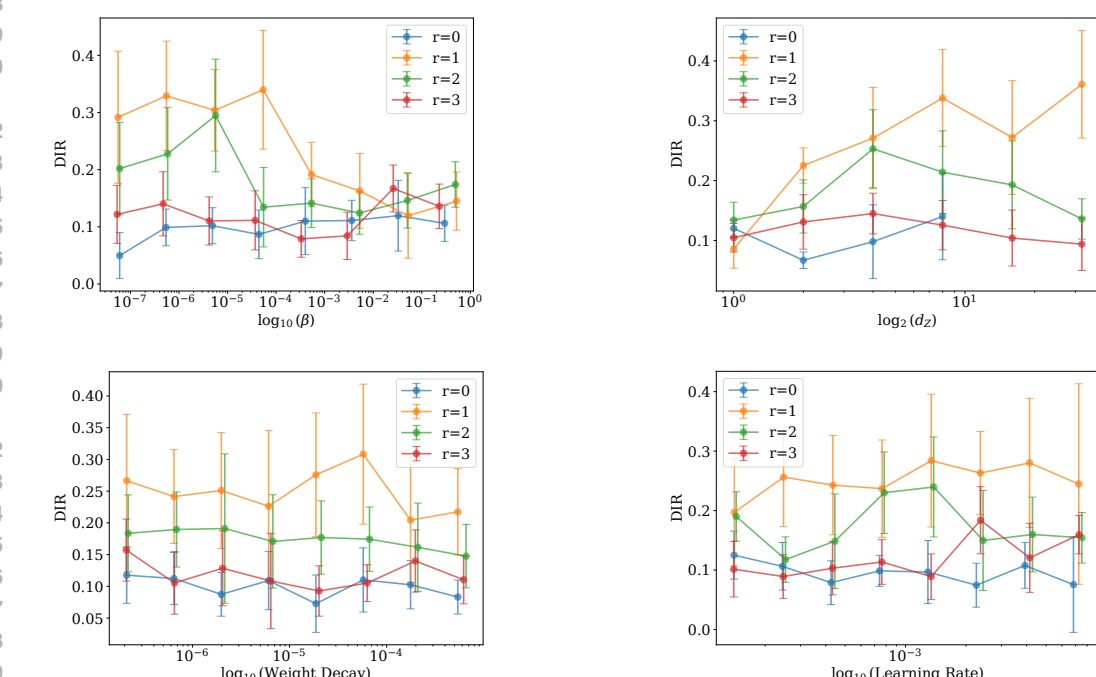

Figure 5: Sensitivity analysis for C-VAE-SPATIAL+ models trained on local confounding environment $SO_4 \rightarrow PM_{2.5}$ ($r_d = 1$) with unobserved confounder $OC$. Each subplot shows DIR as a function of a hyperparameter across different neighborhood radii $r$. The error bounds represent the 95% confidence interval. The y-axis represents the error on the direct effect.

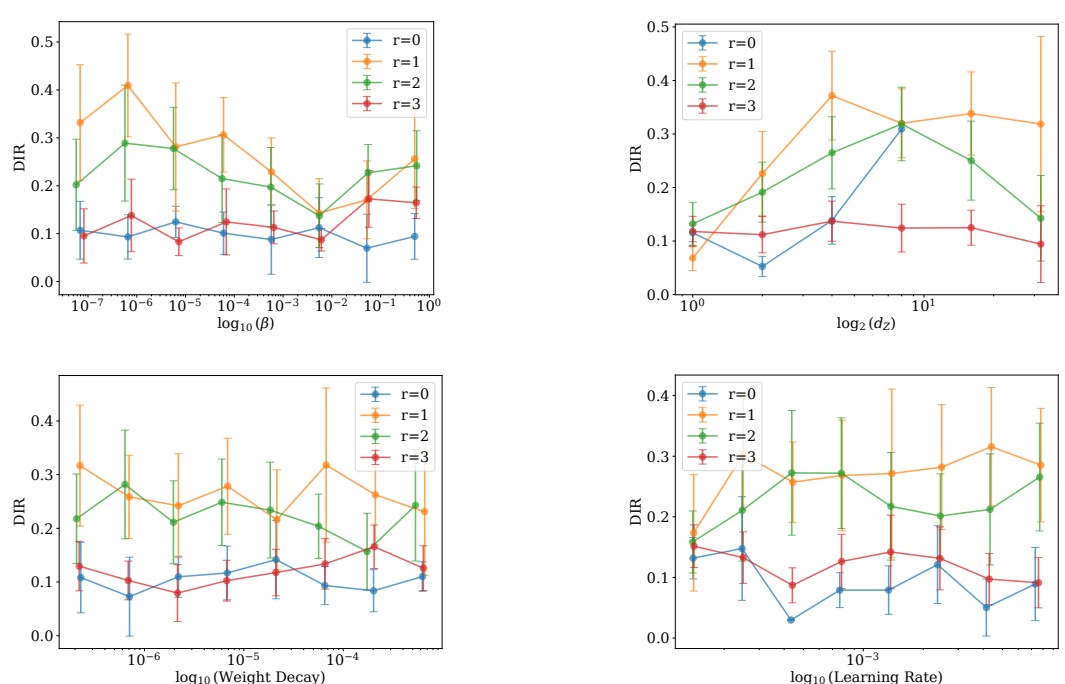

Figure 6: Sensitivity analysis for C-VAE-SPATIAL+ models trained on local confounding environment $SO_4 \rightarrow PM_{2.5}$ ($r_d = 1$) with unobserved confounder $NH_4$. Each subplot shows DIR as a function of a hyperparameter across different neighborhood radii $r$. The error bounds represent the 95% confidence interval. The y-axis represents the error on the direct effect.

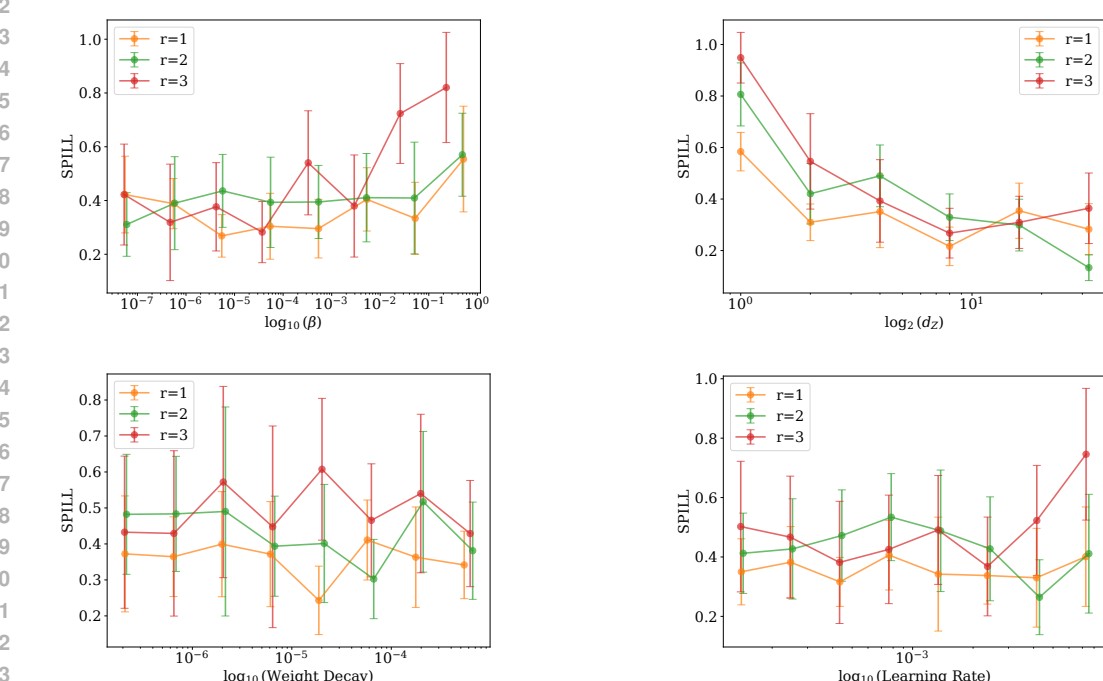

Figure 7: Sensitivity analysis for C-VAE-SPATIAL+ models trained on local confounding environment $SO_4 \rightarrow PM_{2.5}$ ($r_d = 1$) with unobserved confounder $OC$. Each subplot shows SPILL as a function of a hyperparameter across different neighborhood radii $r$. The error bounds represent the 95% confidence interval. The y-axis represents the error on the spillover effect.

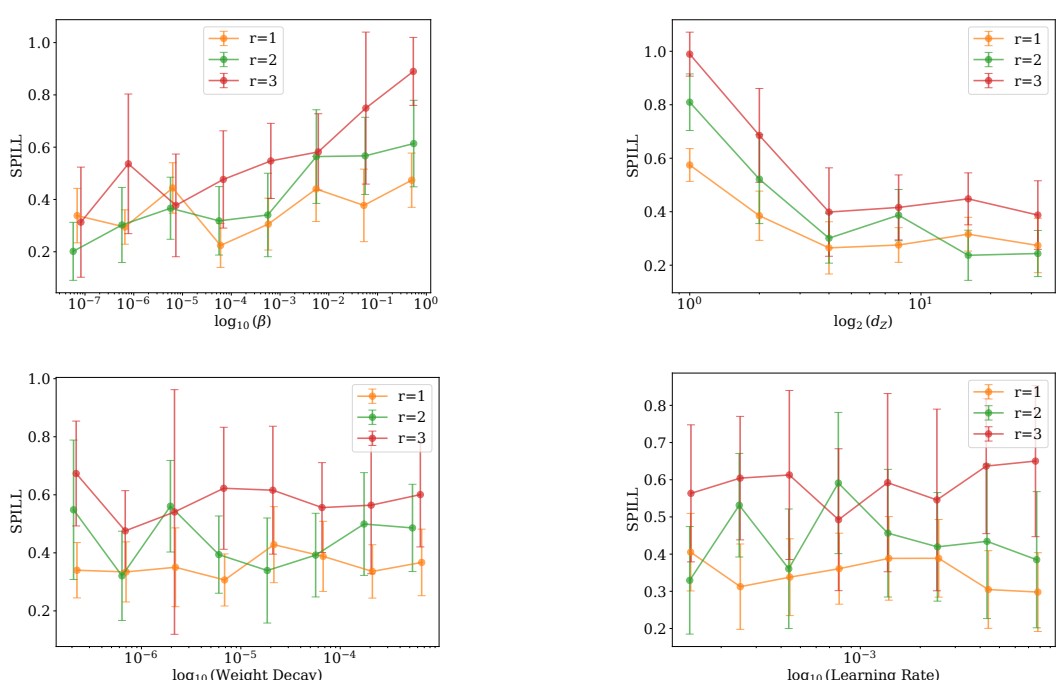

Figure 8: Sensitivity analysis for C-VAE-SPATIAL+ models trained on local confounding environment $SO_4 \rightarrow PM_{2.5}$ ($r_d = 1$) with unobserved confounder $NH_4$. Each subplot shows SPILL as a function of a hyperparameter across different neighborhood radii $r$. The error bounds represent the 95% confidence interval. The y-axis represents the error on the spillover effect.

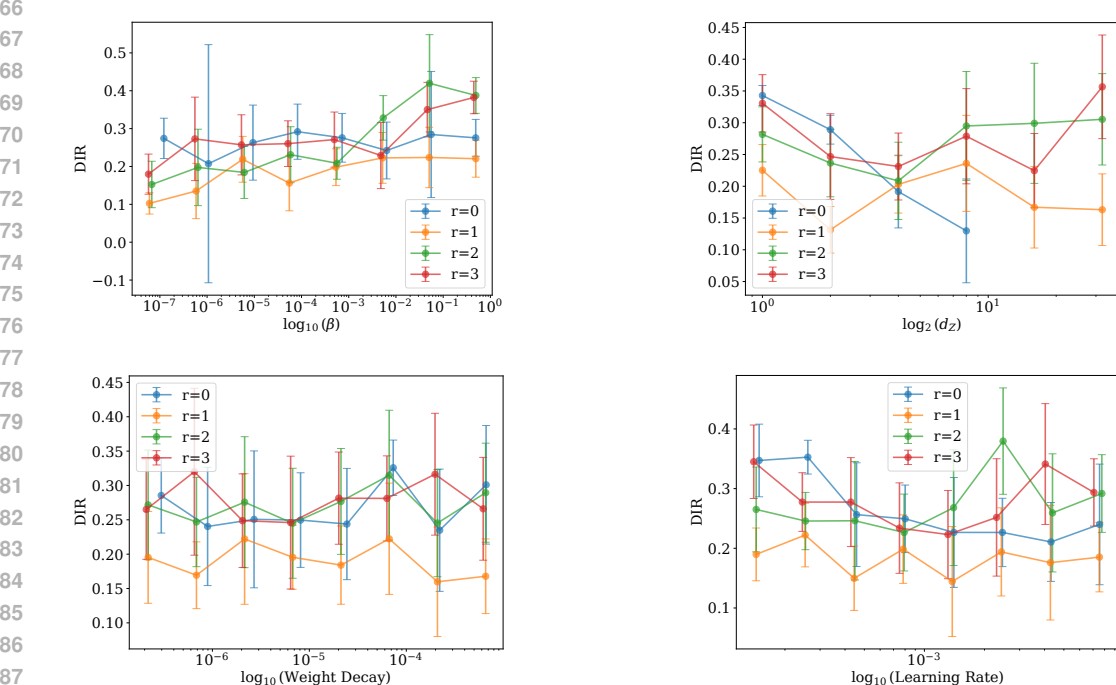

Figure 9: Sensitivity analysis for C-VAE-SPATIAL+ models trained on local confounding environment $SO_4 \rightarrow PM_{2.5}$ ($r_d = 2$) with unobserved confounder $OC$. Each subplot shows DIR as a function of a hyperparameter across different neighborhood radii $r$. The error bounds represent the 95% confidence interval. The y-axis represents the error on the direct effect.

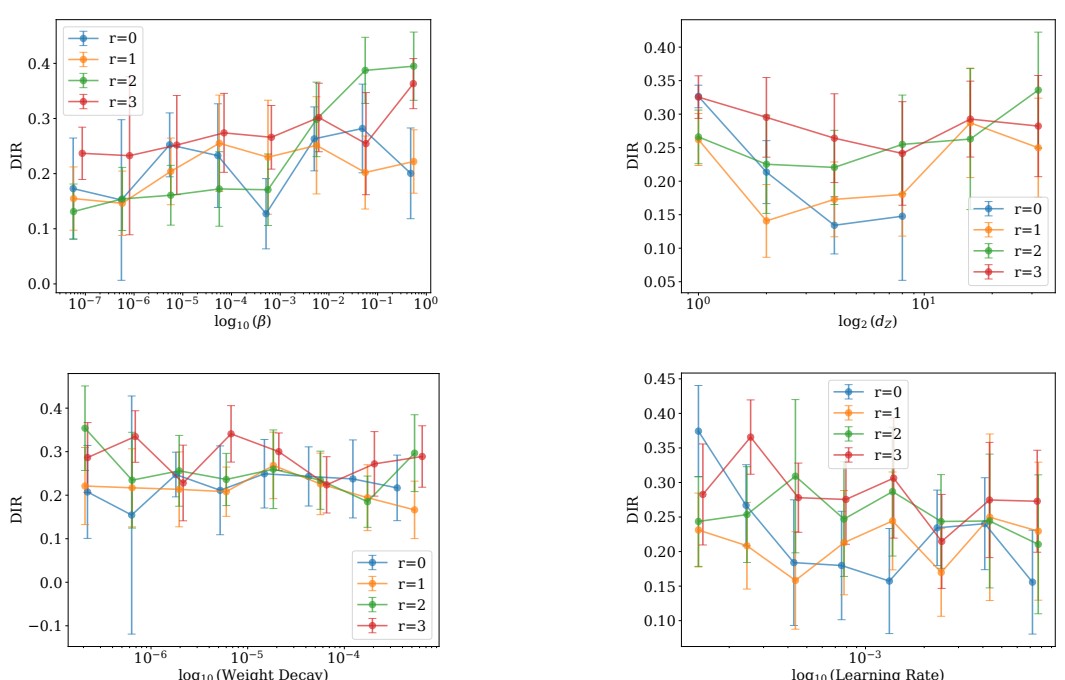

Figure 10: Sensitivity analysis for C-VAE-SPATIAL+ models trained on local confounding environment $SO_4 \rightarrow PM_{2.5}$ ($r_d = 2$) with unobserved confounder $NH_4$. Each subplot shows DIR as a function of a hyperparameter across different neighborhood radii $r$. The error bounds represent the 95% confidence interval. The y-axis represents the error on the direct effect.

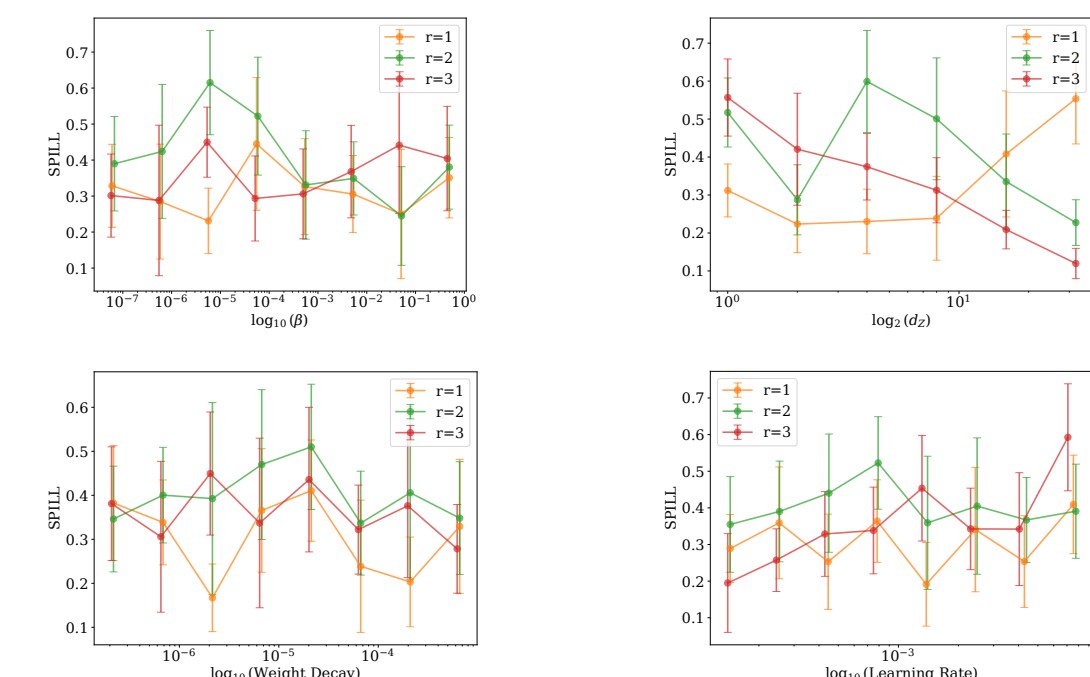

Figure 11: Sensitivity analysis for C-VAE-SPATIAL+ models trained on local confounding environment $SO_4 \rightarrow PM_{2.5}$ $(r_d = 2)$ with unobserved confounder $OC$. Each subplot shows SPILL as a function of a hyperparameter across different neighborhood radii $r$. The error bounds represent the 95% confidence interval. The y-axis represents the error on the spillover effect.

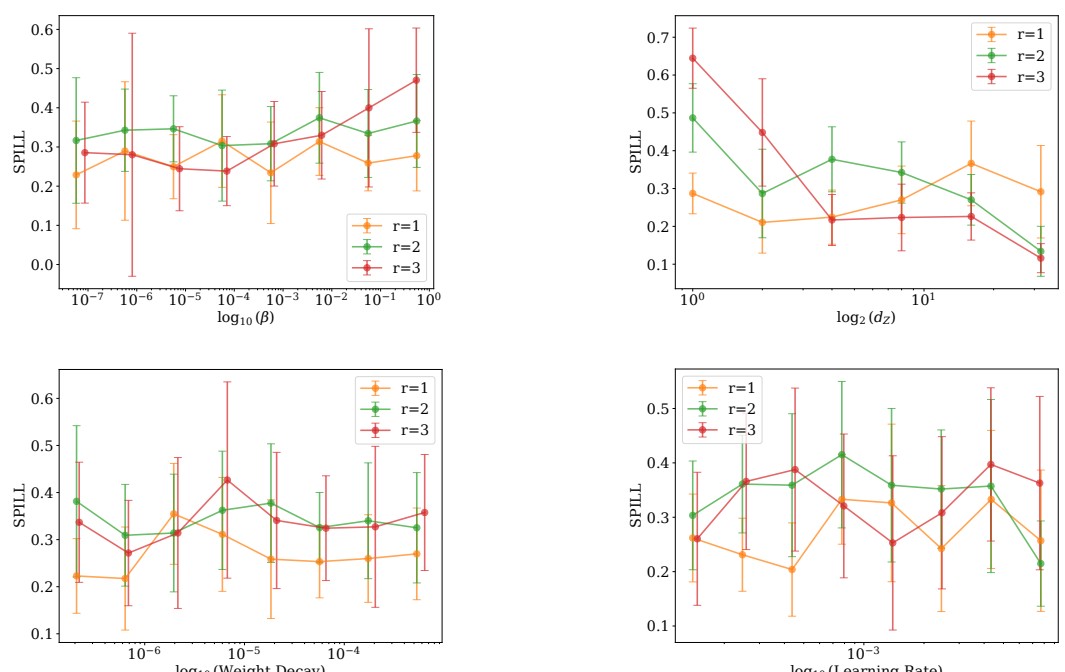

Figure 12: Sensitivity analysis for C-VAE-SPATIAL+ models trained on local confounding environment $SO_4 \rightarrow PM_{2.5}$ $(r_d = 2)$ with unobserved confounder $NH_4$. Each subplot shows SPILL as a function of a hyperparameter across different neighborhood radii $r$. The error bounds represent the 95% confidence interval. The y-axis represents the error on the spillover effect.

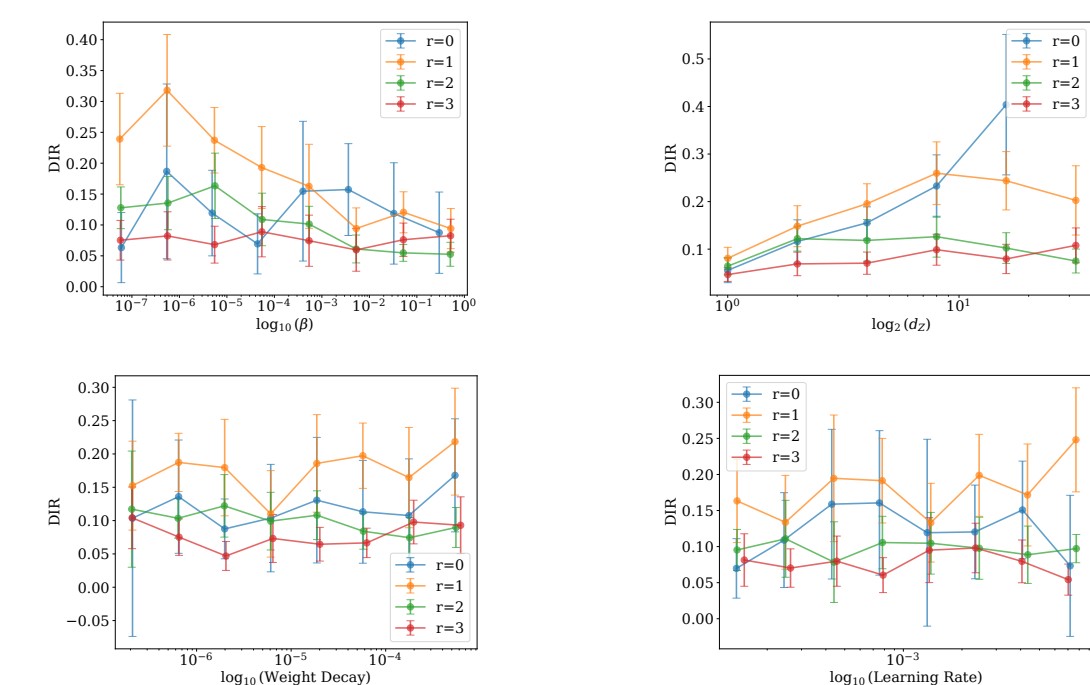

Figure 13: Sensitivity analysis for C-VAE-SPATIAL+ models trained on local confounding environment $PM_{2.5} \rightarrow m$ ($r_d = 1$) with unobserved confounder $q_{summer}$. Each subplot shows DIR as a function of a hyperparameter across different neighborhood radii $r$. The error bounds represent the 95% confidence interval. The y-axis represents the error on the direct effect.

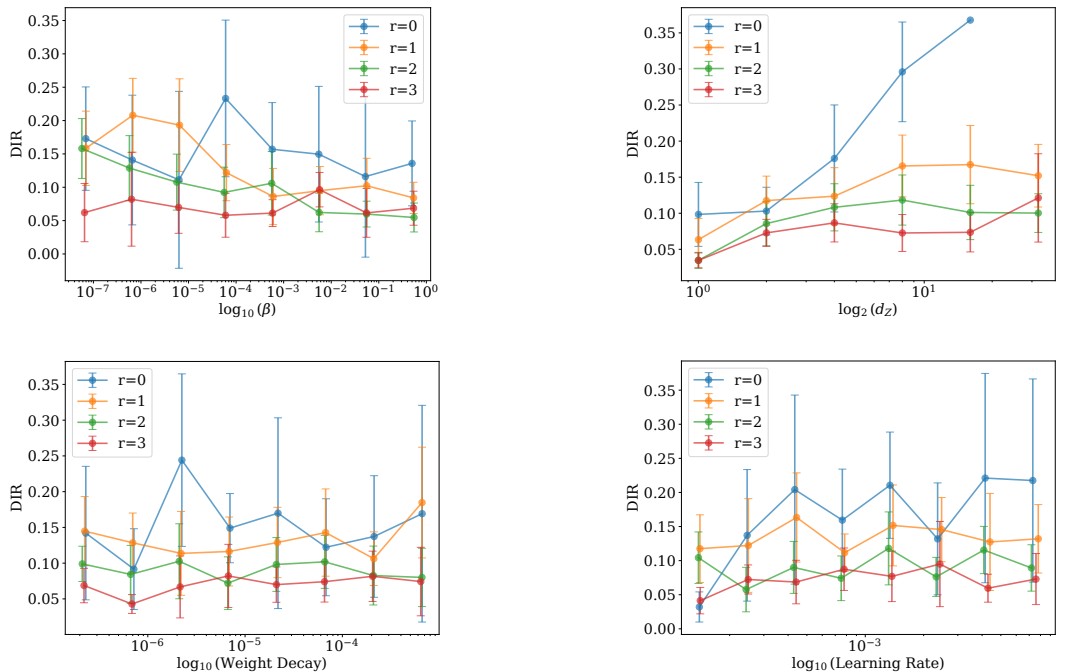

Figure 14: Sensitivity analysis for C-VAE-SPATIAL+ models trained on local confounding environment $PM_{2.5} \rightarrow m$ ($r_d = 1$) with unobserved confounder $\rho_{pop}$. Each subplot shows DIR as a function of a hyperparameter across different neighborhood radii $r$. The error bounds represent the 95% confidence interval. The y-axis represents the error on the direct effect.

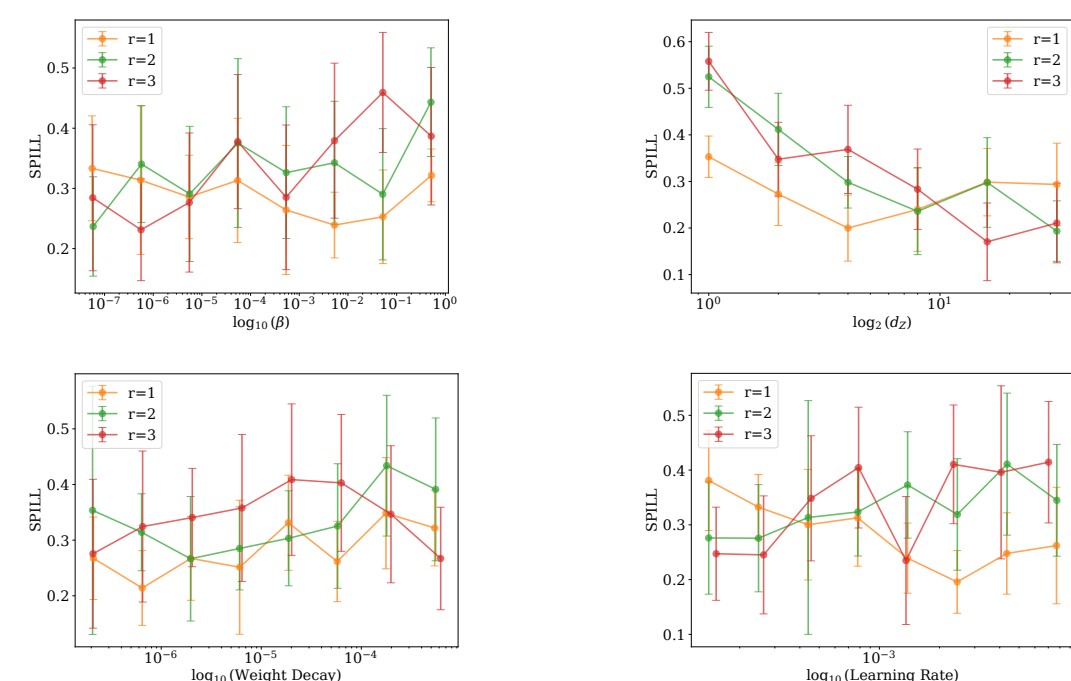

Figure 15: Sensitivity analysis for C-VAE-SPATIAL+ models trained on local confounding environment $PM_{2.5} \rightarrow m \ (r_d = 1)$ with unobserved confounder $q_{\text{summer}}$. Each subplot shows SPILL as a function of a hyperparameter across different neighborhood radii $r$. The error bounds represent the 95% confidence interval. The y-axis represents the error on the spillover effect.

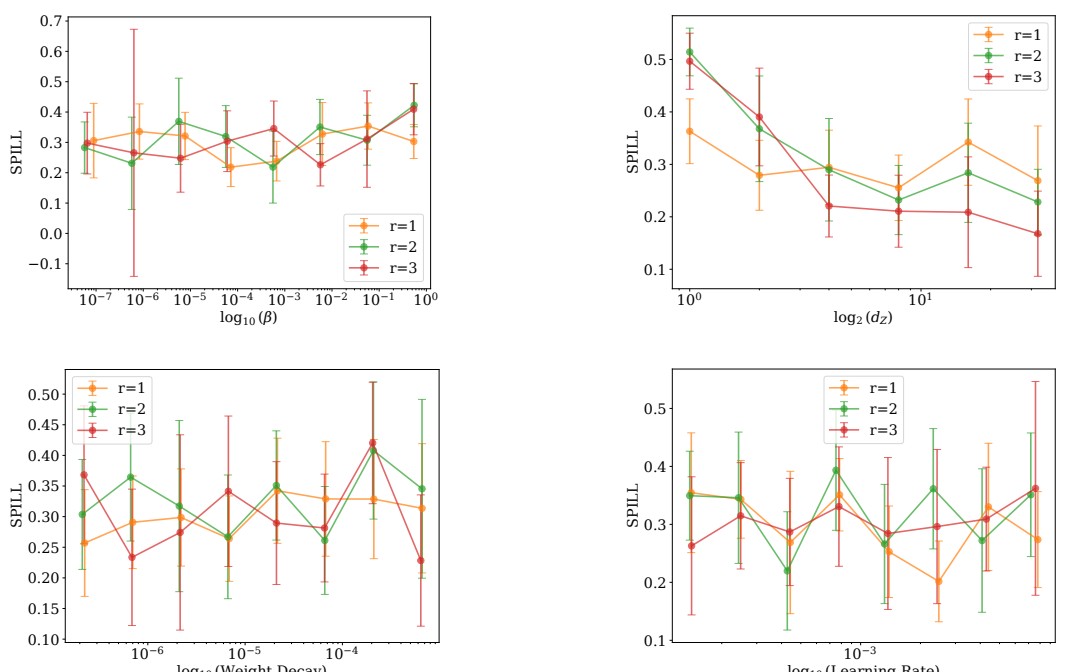

Figure 16: Sensitivity analysis for C-VAE-SPATIAL+ models trained on local confounding environment $PM_{2.5} \rightarrow m \ (r_d = 1)$ with unobserved confounder $\rho_{\text{pop}}$. Each subplot shows SPILL as a function of a hyperparameter across different neighborhood radii $r$. The error bounds represent the 95% confidence interval. The y-axis represents the error on the spillover effect.

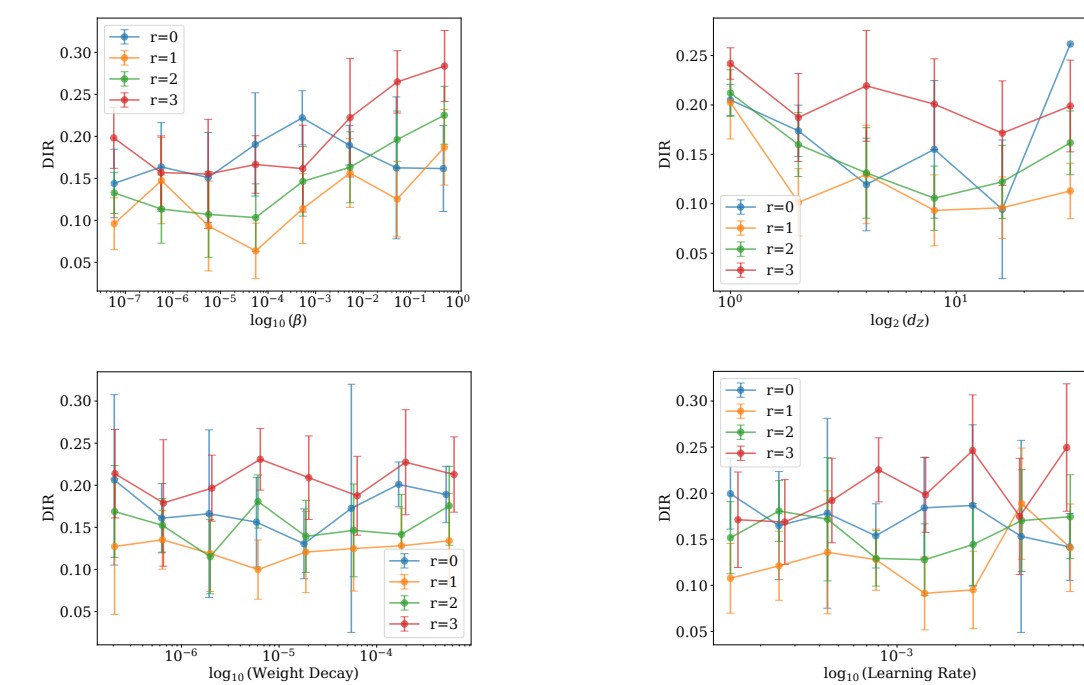

Figure 17: Sensitivity analysis for C-VAE-SPATIAL+ models trained on local confounding environment $PM_{2.5} \rightarrow m$ ($r_d = 2$) with unobserved confounder $q_{summer}$. Each subplot shows DIR as a function of a hyperparameter across different neighborhood radii $r$. The error bounds represent the 95% confidence interval. The y-axis represents the error on the direct effect.

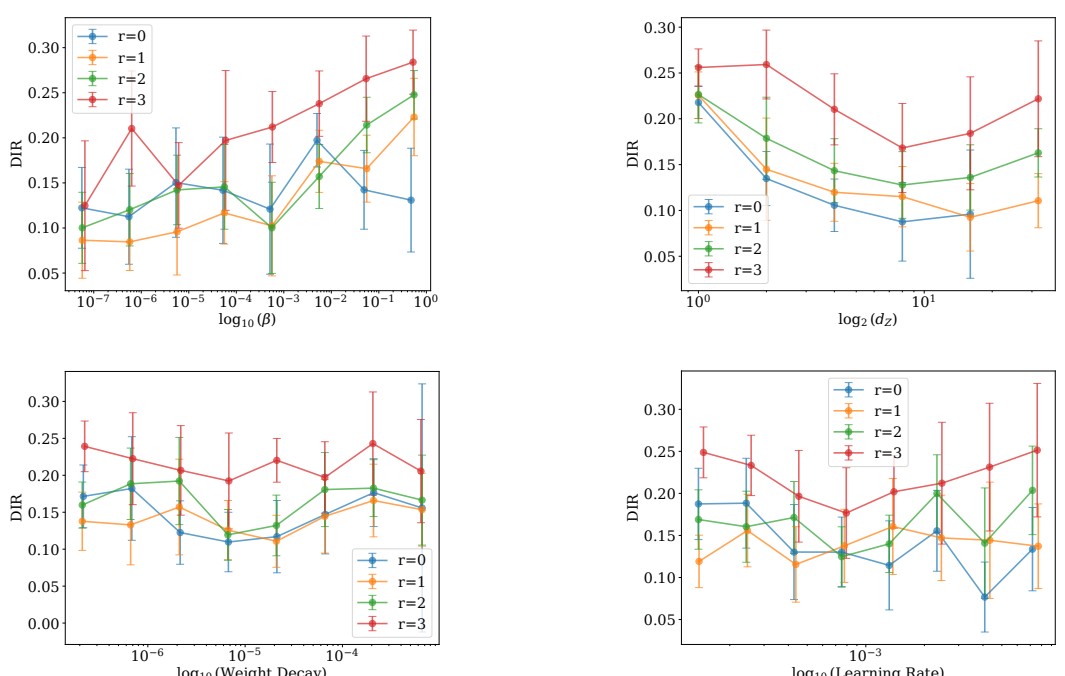

Figure 18: Sensitivity analysis for C-VAE-SPATIAL+ models trained on local confounding environment $PM_{2.5} \rightarrow m$ ($r_d = 2$) with unobserved confounder $\rho_{pop}$. Each subplot shows DIR as a function of a hyperparameter across different neighborhood radii $r$. The error bounds represent the 95% confidence interval. The y-axis represents the error on the direct effect.

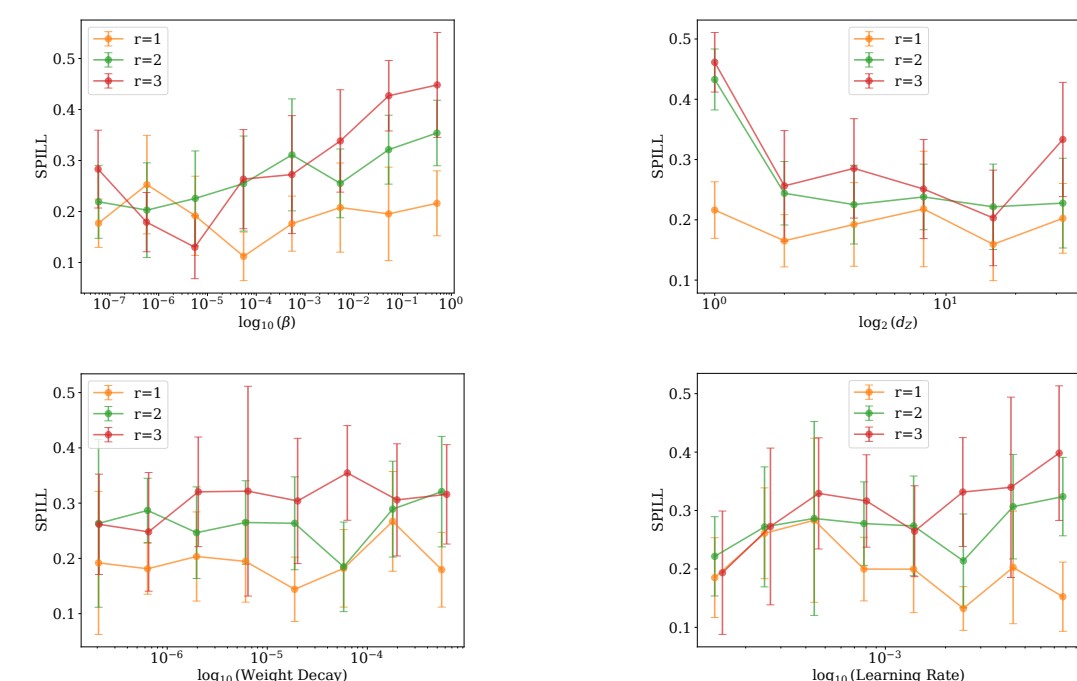

Figure 19: Sensitivity analysis for C-VAE-SPATIAL+ models trained on local confounding environment $PM_{2.5} \rightarrow m$ ($r_d = 2$) with unobserved confounder $q_{\text{summer}}$. Each subplot shows SPILL as a function of a hyperparameter across different neighborhood radii $r$. The error bounds represent the 95% confidence interval. The y-axis represents the error on the spillover effect.

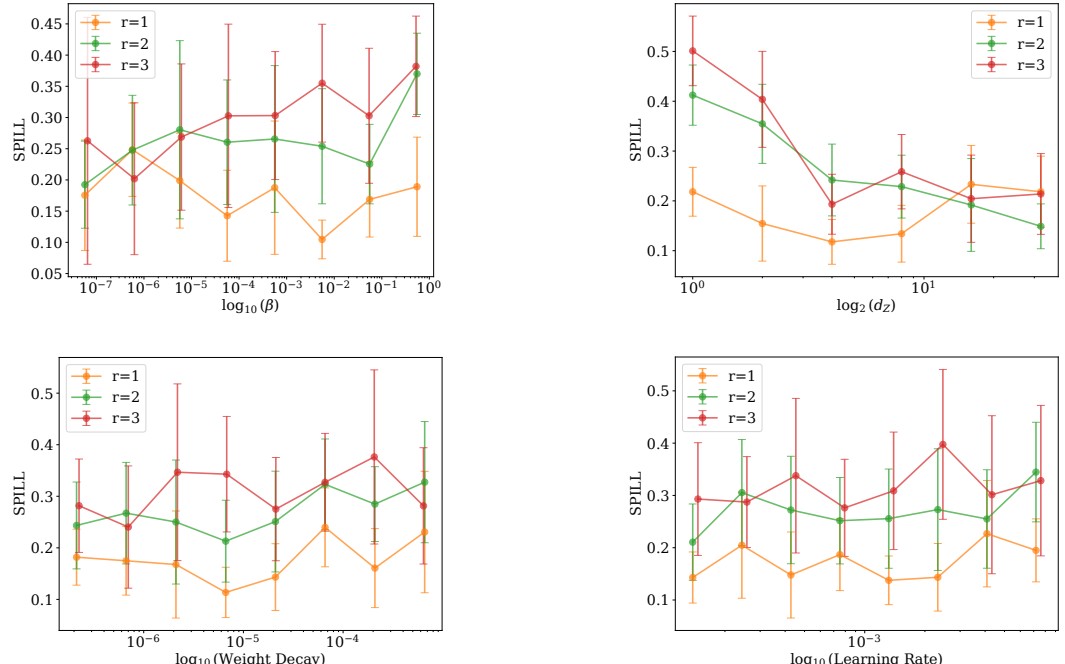

Figure 20: Sensitivity analysis for C-VAE-SPATIAL+ models trained on local confounding environment $PM_{2.5} \rightarrow m$ ($r_d = 2$) with unobserved confounder $\rho_{\text{pop}}$. Each subplot shows SPILL as a function of a hyperparameter across different neighborhood radii $r$. The error bounds represent the 95% confidence interval. The y-axis represents the error on the spillover effect.

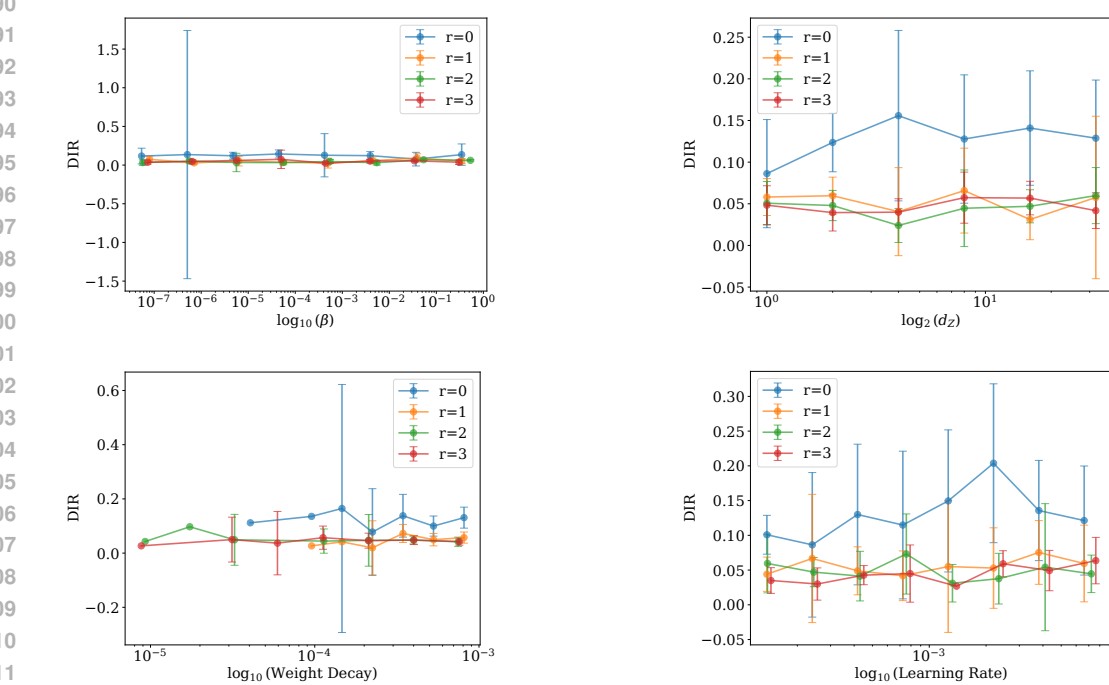

Figure 21: Sensitivity analysis for C-VAE-UNET models trained on spatial confounding environment $SO_4 \rightarrow PM_{2.5}$ ($r_d = 1$) with unobserved confounder $OC$. Each subplot shows DIR as a function of a hyperparameter across different neighborhood radii $r$. The error bounds represent the 95% confidence interval.

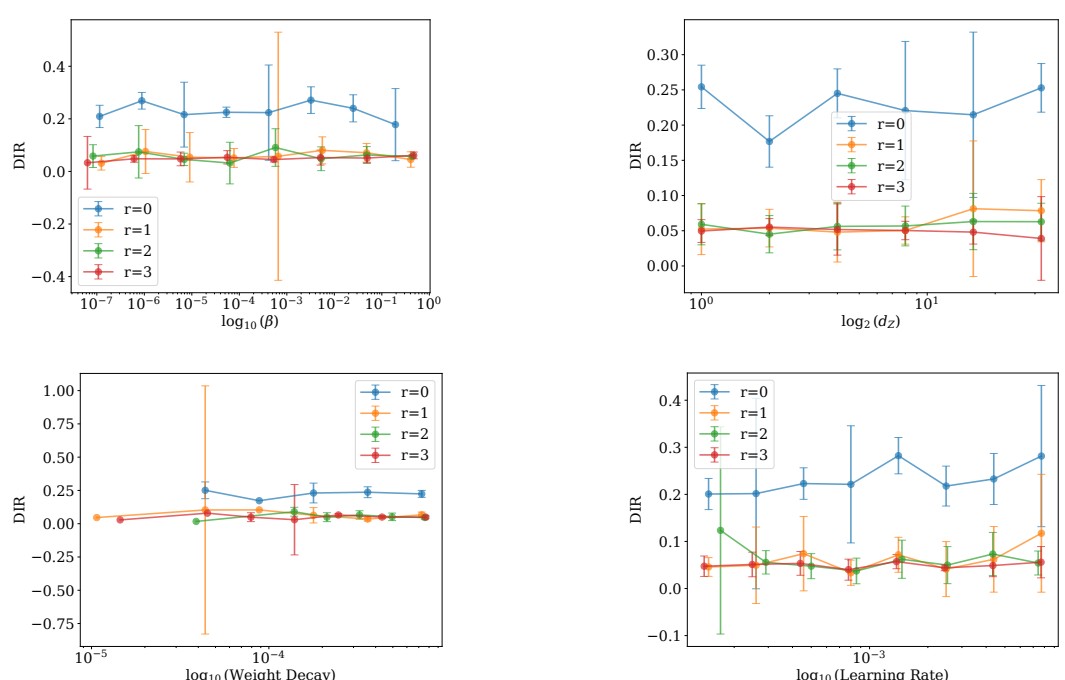

Figure 22: Sensitivity analysis for C-VAE-UNET models trained on spatial confounding environment $SO_4 \rightarrow PM_{2.5}$ ($r_d = 1$) with unobserved confounder $NH_4$. Each subplot shows DIR as a function of a hyperparameter across different neighborhood radii $r$. The error bounds represent the 95% confidence interval.

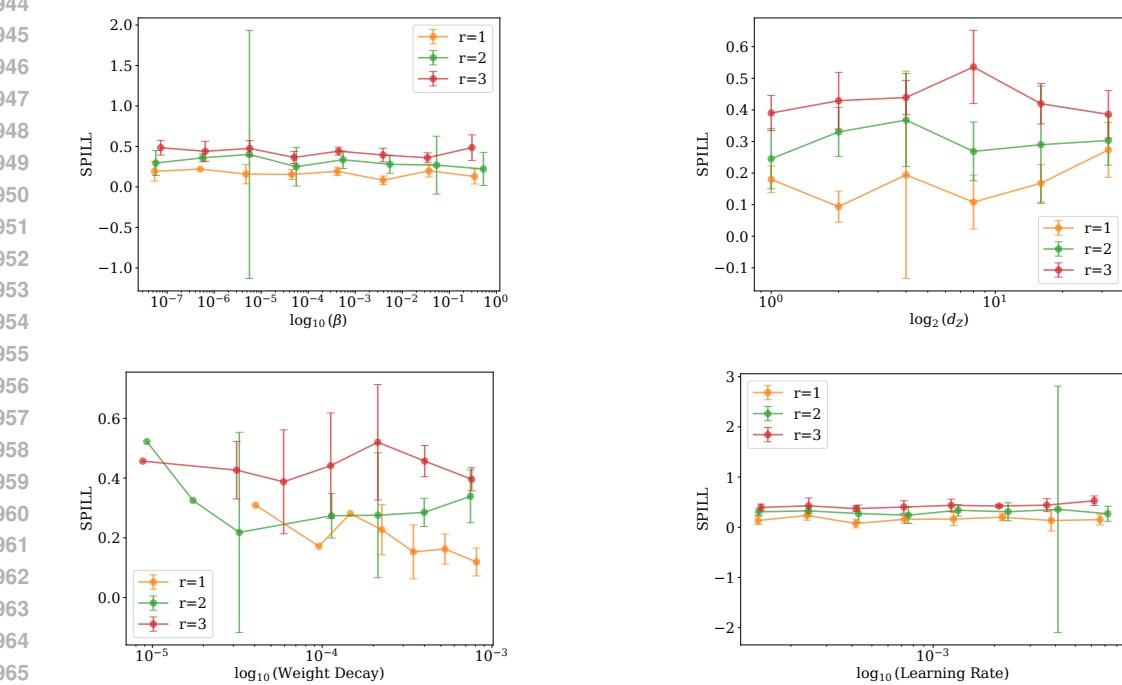

Figure 23: Sensitivity analysis for C-VAE-UNET models trained on spatial confounding environment $SO_4 \rightarrow PM_{2.5}$ ($r_d = 1$) with unobserved confounder $OC$. Each subplot shows SPILL as a function of a hyperparameter across different neighborhood radii $r$. The error bounds represent the 95% confidence interval.

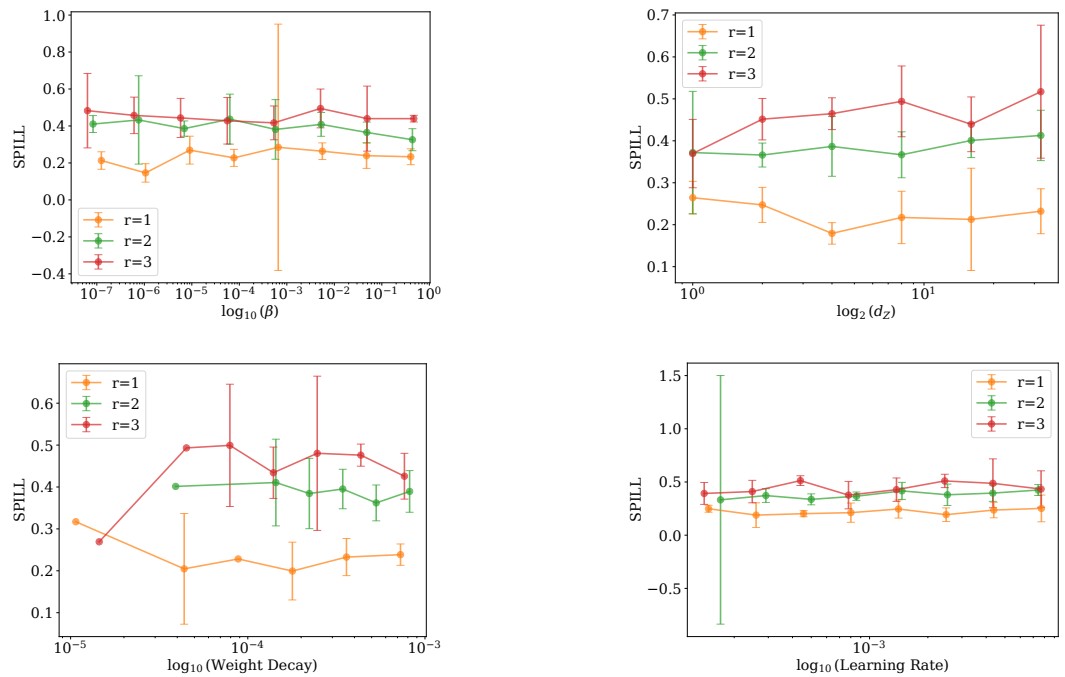

Figure 24: Sensitivity analysis for C-VAE-UNET models trained on spatial confounding environment $SO_4 \rightarrow PM_{2.5}$ ($r_d = 1$) with unobserved confounder $NH_4$. Each subplot shows SPILL as a function of a hyperparameter across different neighborhood radii $r$. The error bounds represent the 95% confidence interval.

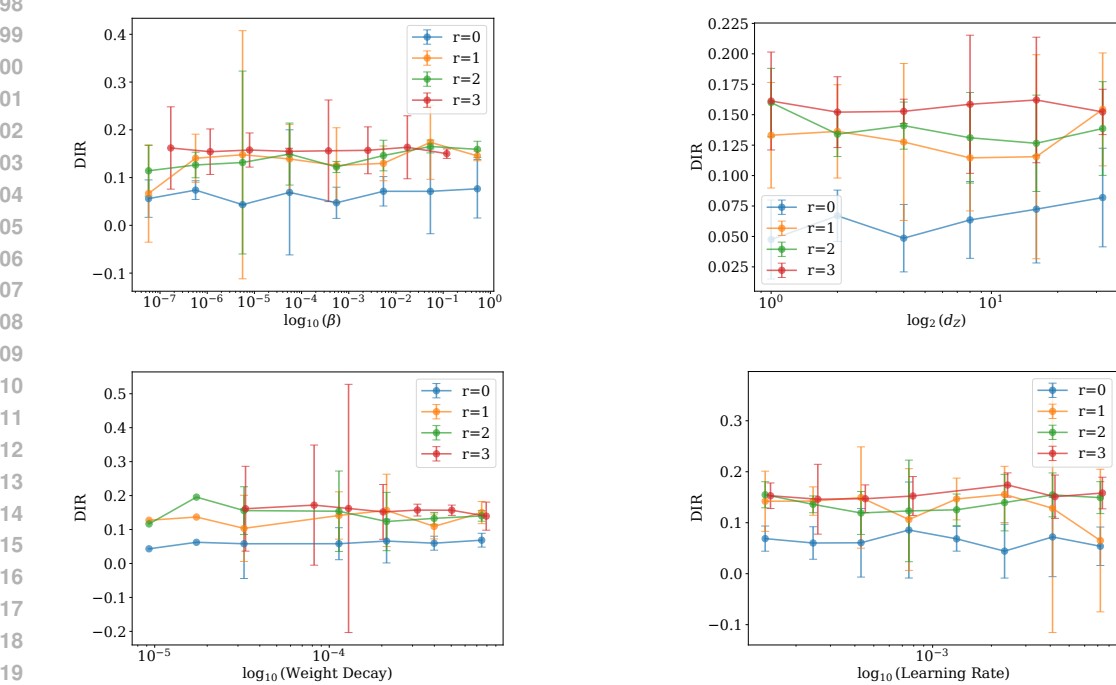

Figure 25: Sensitivity analysis for C-VAE-UNET models trained on spatial confounding environment $SO_4 \rightarrow PM_{2.5}$ ($r_d = 2$) with unobserved confounder $OC$. Each subplot shows DIR as a function of a hyperparameter across different neighborhood radii $r$. The error bounds represent the 95% confidence interval.

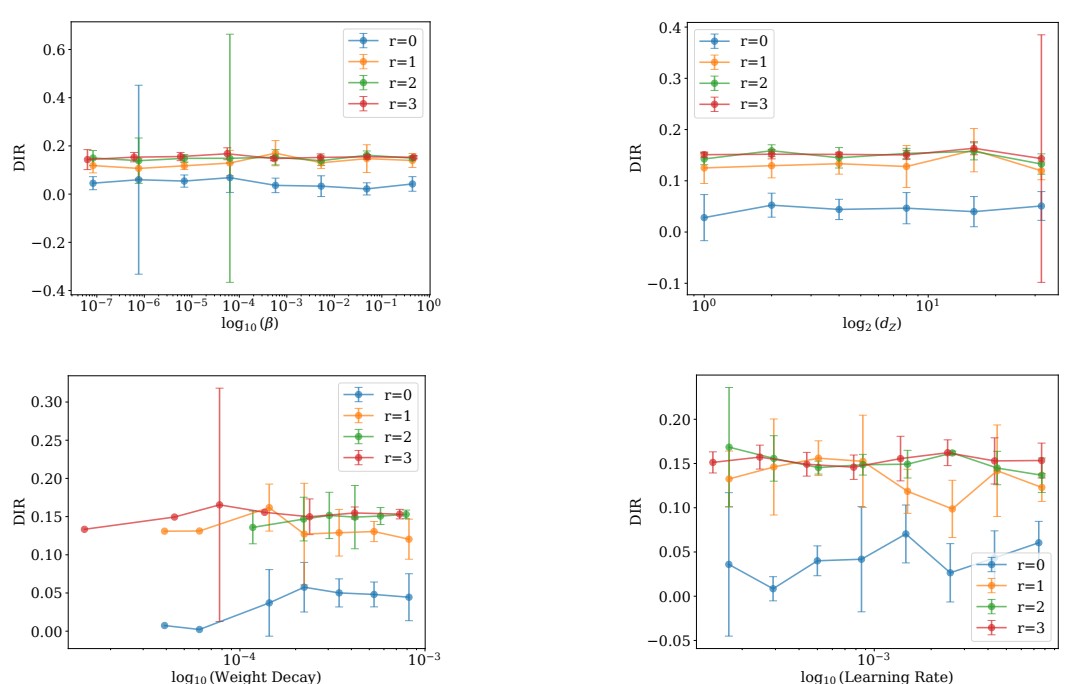

Figure 26: Sensitivity analysis for C-VAE-UNET models trained on spatial confounding environment $SO_4 \rightarrow PM_{2.5}$ ($r_d = 2$) with unobserved confounder $NH_4$. Each subplot shows DIR as a function of a hyperparameter across different neighborhood radii $r$. The error bounds represent the 95% confidence interval.

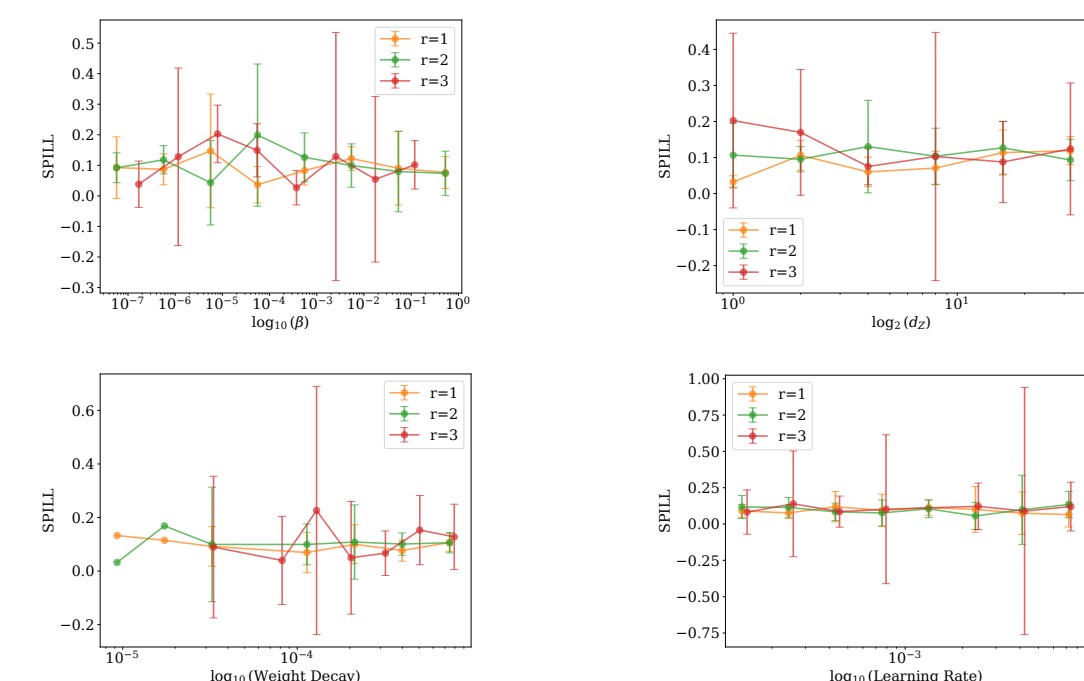

Figure 27: Sensitivity analysis for C-VAE-UNET models trained on spatial confounding environment $SO_4 \rightarrow PM_{2.5}$ ($r_d = 2$) with unobserved confounder $OC$. Each subplot shows SPILL as a function of a hyperparameter across different neighborhood radii $r$. The error bounds represent the 95% confidence interval.

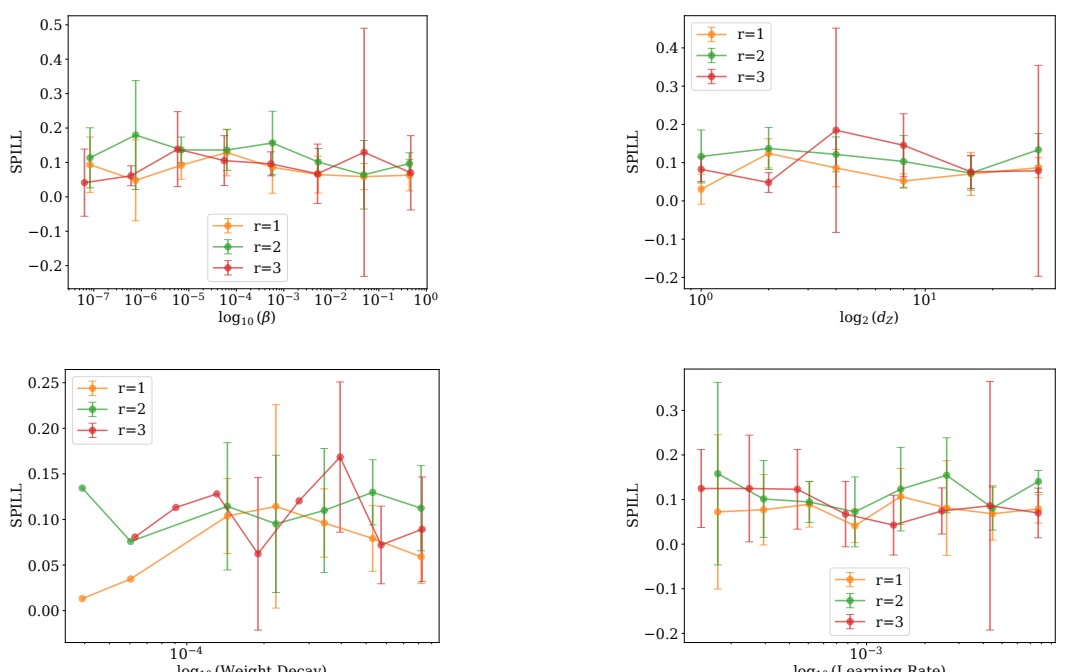

Figure 28: Sensitivity analysis for C-VAE-UNET models trained on spatial confounding environment $SO_4 \rightarrow PM_{2.5}$ ($r_d = 2$) with unobserved confounder $NH_4$. Each subplot shows SPILL as a function of a hyperparameter across different neighborhood radii $r$. The error bounds represent the 95% confidence interval.

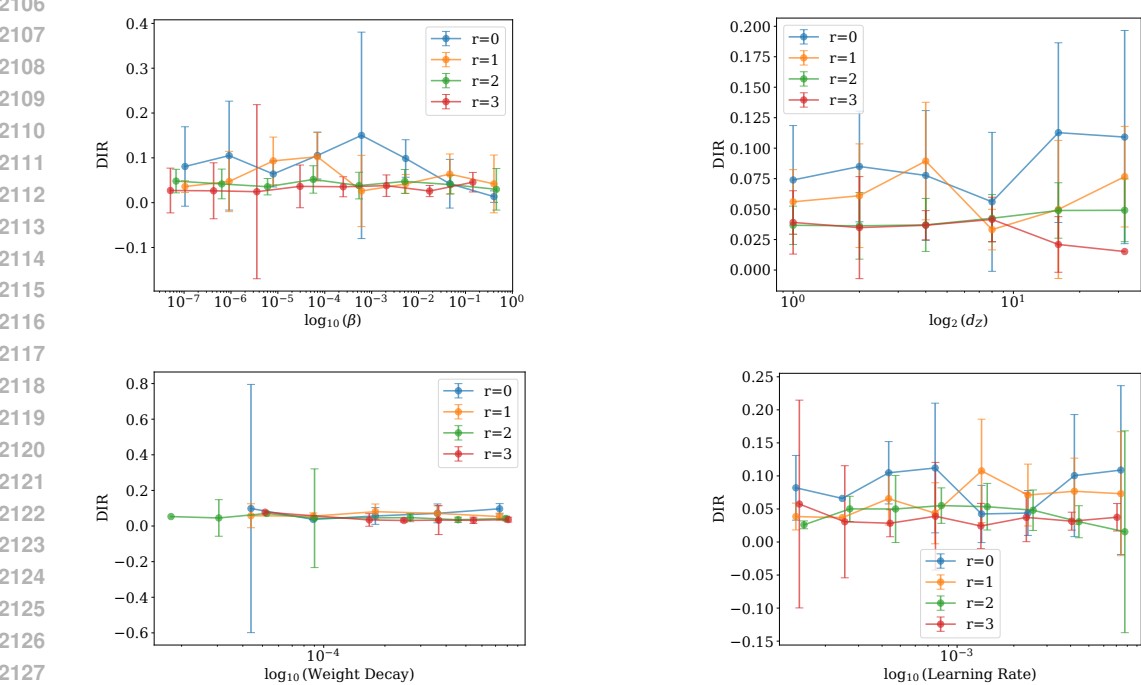

Figure 29: Sensitivity analysis for C-VAE-UNET models trained on spatial confounding environment $PM_{2.5} \rightarrow m$ $(r_d = 1)$ with unobserved confounder $q_{\text{summer}}$. Each subplot shows DIR as a function of a hyperparameter across different neighborhood radii $r$. The error bounds represent the 95% confidence interval.

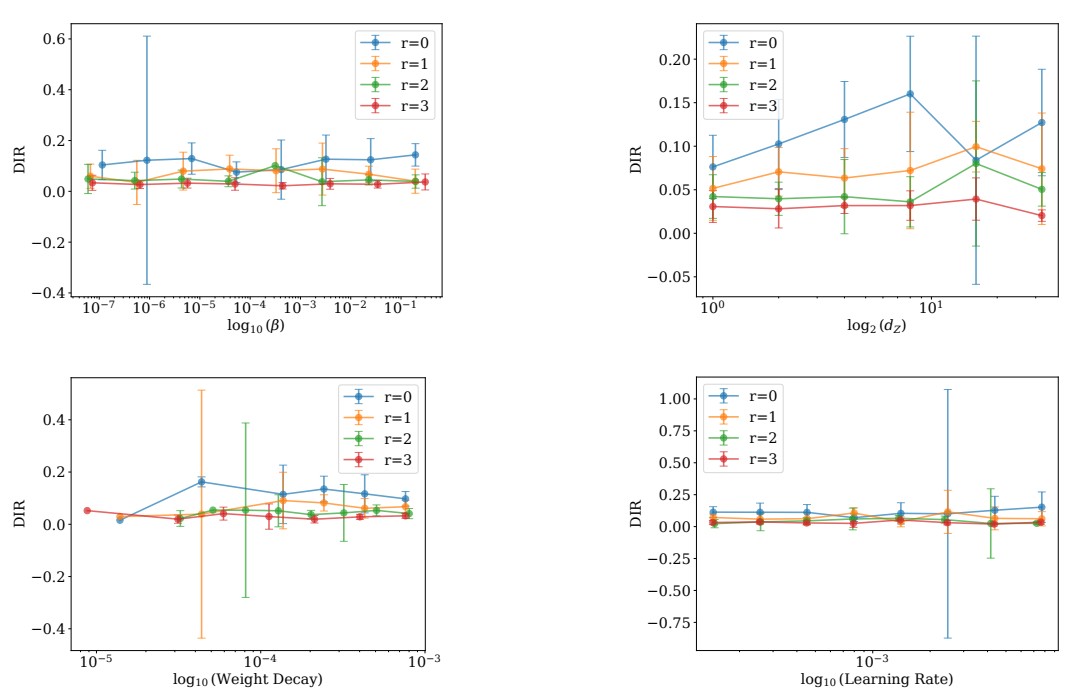

Figure 30: Sensitivity analysis for C-VAE-UNET models trained on spatial confounding environment $PM_{2.5} \rightarrow m$ $(r_d = 1)$ with unobserved confounder $\rho_{\text{pop}}$. Each subplot shows DIR as a function of a hyperparameter across different neighborhood radii $r$. The error bounds represent the 95% confidence interval.

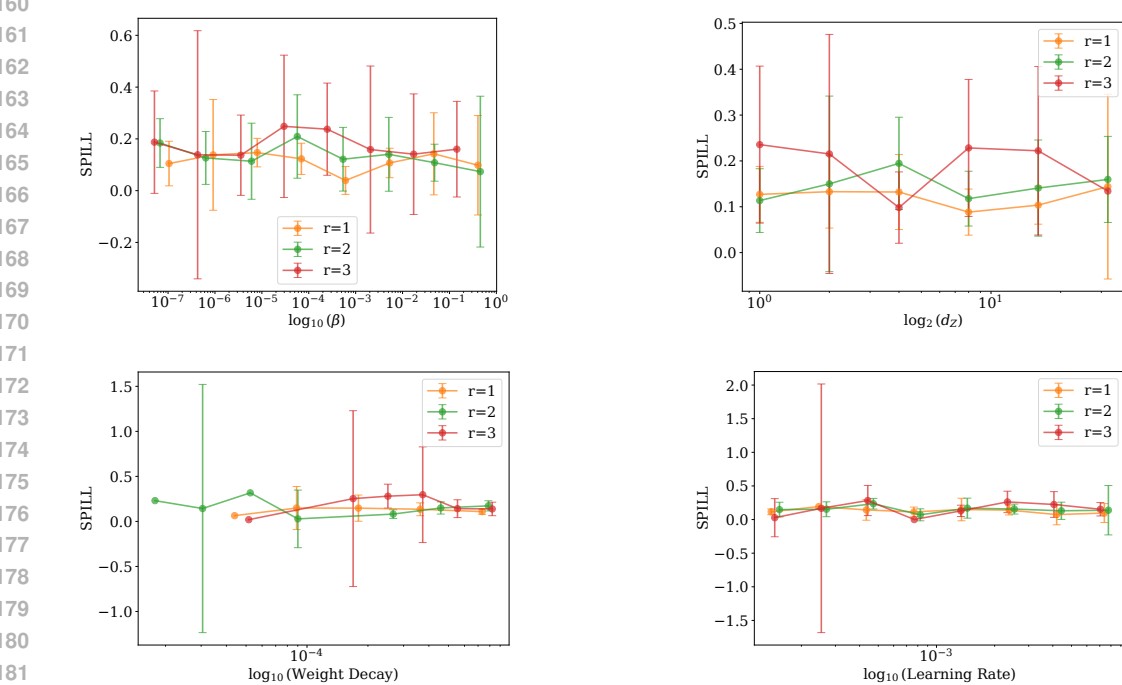

Figure 31: Sensitivity analysis for C-VAE-UNET models trained on spatial confounding environment $PM_{2.5} \rightarrow m\ (r_d = 1)$ with unobserved confounder $q_{\text{summer}}$. Each subplot shows SPILL as a function of a hyperparameter across different neighborhood radii $r$. The error bounds represent the 95% confidence interval.

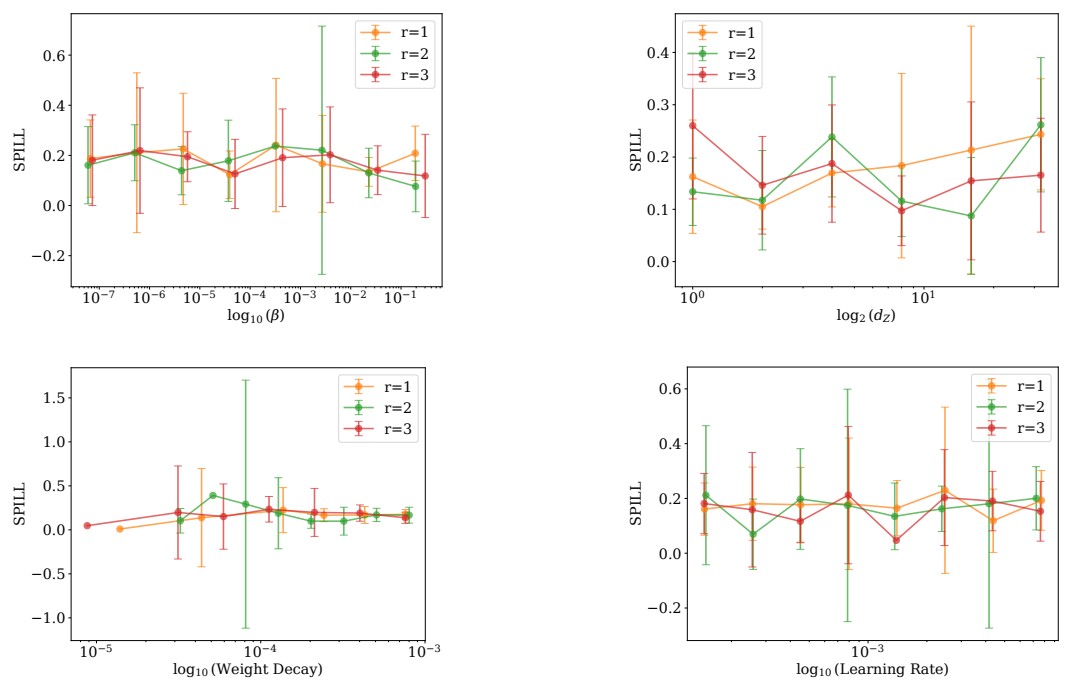

Figure 32: Sensitivity analysis for C-VAE-UNET models trained on spatial confounding environment $PM_{2.5} \rightarrow m\ (r_d = 1)$ with unobserved confounder $\rho_{\text{pop}}$. Each subplot shows SPILL as a function of a hyperparameter across different neighborhood radii $r$. The error bounds represent the 95% confidence interval.

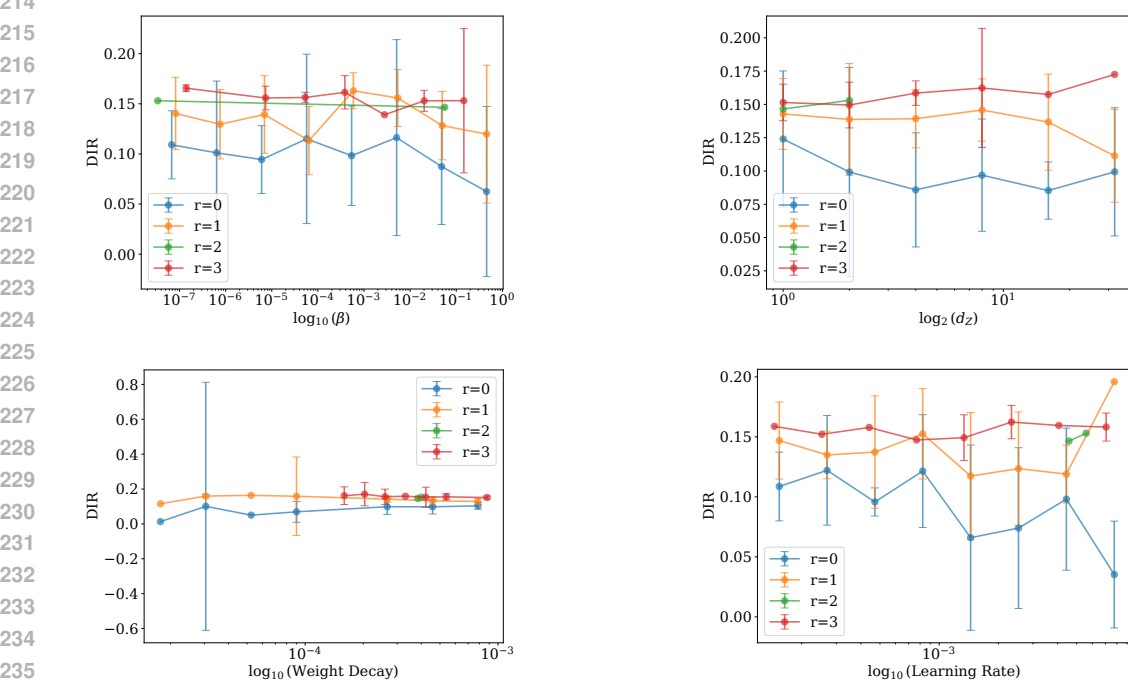

Figure 33: Sensitivity analysis for C-VAE-UNET models trained on spatial confounding environment $PM_{2.5} \rightarrow m$ ($r_d = 2$) with unobserved confounder $q_{\text{summer}}$. Each subplot shows DIR as a function of a hyperparameter across different neighborhood radii $r$. The error bounds represent the 95% confidence interval.

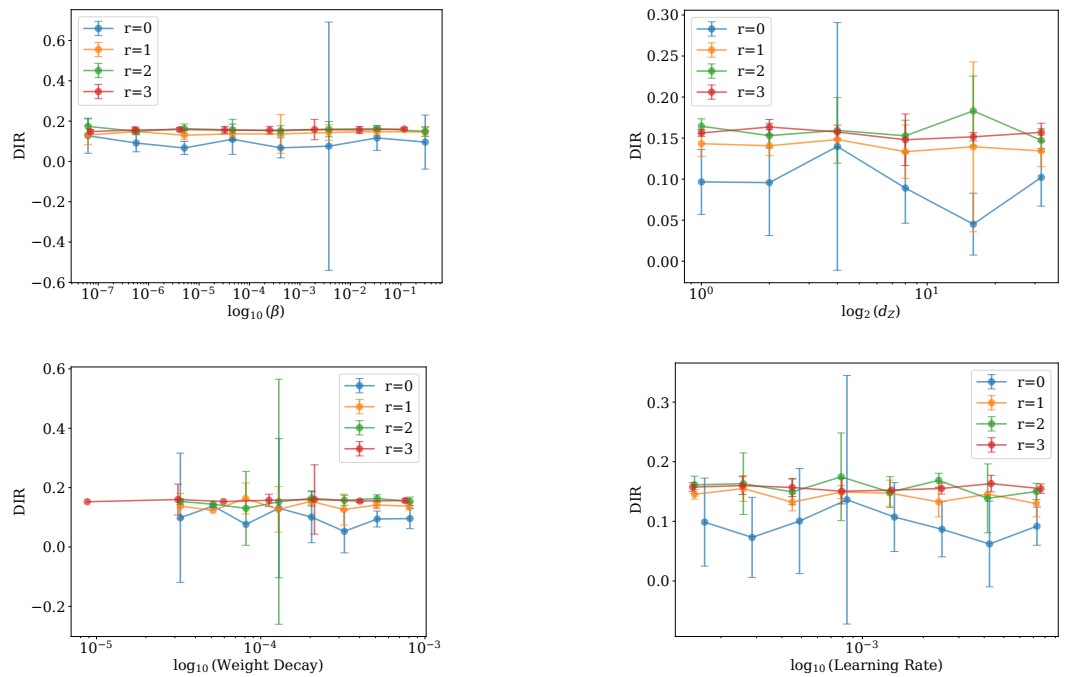

Figure 34: Sensitivity analysis for C-VAE-UNET models trained on spatial confounding environment $PM_{2.5} \rightarrow m$ ($r_d = 2$) with unobserved confounder $\rho_{\text{pop}}$. Each subplot shows DIR as a function of a hyperparameter across different neighborhood radii $r$. The error bounds represent the 95% confidence interval.

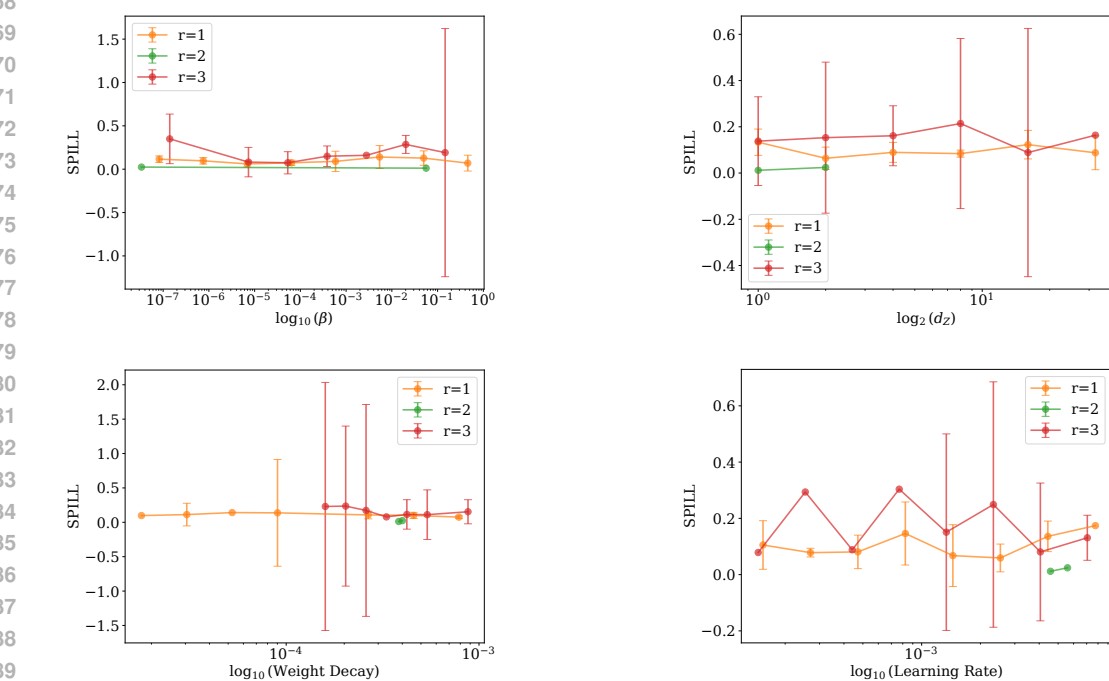

Figure 35: Sensitivity analysis for C-VAE-UNET models trained on spatial confounding environment $PM_{2.5} \rightarrow m$ $(r_d = 2)$ with unobserved confounder $q_{summer}$. Each subplot shows SPILL as a function of a hyperparameter across different neighborhood radii $r$. The error bounds represent the 95% confidence interval.

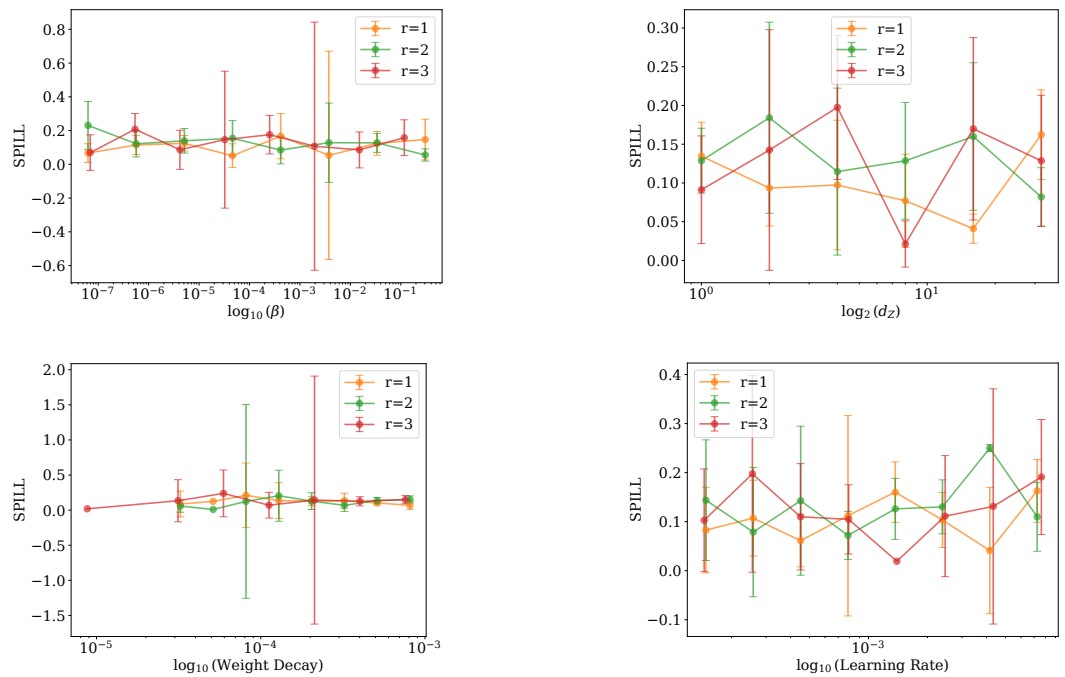

Figure 36: Sensitivity analysis for C-VAE-UNET models trained on spatial confounding environment $PM_{2.5} \rightarrow m$ $(r_d = 2)$ with unobserved confounder $\rho_{pop}$. Each subplot shows SPILL as a function of a hyperparameter across different neighborhood radii $r$. The error bounds represent the 95% confidence interval.

## F  BROADER IMPACTS AND LLM DISCLOSURE

**Limitations**  While the Spatial Deconfounder advances identification and estimation under interference and unobserved spatial confounding, several limitations remain. First, our theoretical guarantees rely on assumptions such as localized interference and smooth latent confounders; these are useful but idealized and may not hold in domains with global spillovers or irregular hidden processes. Second, the framework is designed for gridded spatial data and assumes a regular lattice; extending to irregular spatial structures (e.g., graphs or administrative units) is an important direction for future work. Finally, although the C-VAE prior aids in recovering latent structure, it may not fully capture unobserved confounders in extremely sparse or noisy data, and computational demands grow with grid size.

**Broader impacts**  This work contributes to machine learning and causal inference by introducing a framework for more reliable effect estimation in spatial domains. Applications include environmental health, climate science, and social sciences, where accurate causal estimates can inform policy decisions. At the same time, we caution against uncritical use in high-stakes settings: violations of assumptions or biases in observational data may yield misleading conclusions. We encourage responsible deployment—especially in contexts affecting vulnerable populations—and recommend pairing our method with domain expertise, sensitivity analyses, and uncertainty quantification.

**LLM usage disclosure.**  We used ChatGPT-5 and Claude Sonnet 4 to assist with editing, restructuring, and polishing the paper text. The authors carefully reviewed, revised, and validated all outputs to ensure alignment with the intended scientific content. All substantive contributions—conceptual framing, methodology, theoretical results, and experiments—are the work of the authors. Consistent with ICLR policy, the authors remain fully responsible for the accuracy and integrity of the paper's content.

