# OpenReview forum: "Spatial Deconfounder: Interference-Aware Deconfounding for Spatial Causal Inference"
_ICLR.cc/2026/Conference — Submitted to ICLR 2026_

### Official Review · Reviewer_yeAS · 2025-10-27

**Soundness:** 3
**Presentation:** 3
**Contribution:** 2
**Rating:** 4
**Confidence:** 4

**Summary:**

This paper proposed a new method, CVAE-SPATIAL+ for spatial causal inference with two challenges in spatial settings: localized interference (spillovers from neighbors) and unobserved spatial confounding (latent fields like meteorology or socioeconomic context). This is a two-stage method. The key idea is to treat interference as a multi‑cause signal: by looking at a unit’s treatment/covariates together with its neighbors’ treatments/covariates, a CVAE is trained for recovering a smooth substitute confounder, then estimate direct and spillover effects with a flexible outcome regression model (e.g., U‑Net/GNN architecture). An important “latent‑field sufficiency” assumption is proposed, which shows that the Z representation of the observed assignment/covariates that is equivalent to the latent field for the purpose of adjustment. The paper provides results for both effects under localized confounding and spatial confounding and evaluates the approach on an extended SpaCE semi‑synthetic benchmark. Across air‑quality/health and PM₂.₅‑components tasks, the proposed method reduces standardized absolute bias relative to spatial econometric, spline/RSR, matching, GCNN, and U‑Net baselines.

**Strengths:**

1.	The methodology is pretty clear and straightforward. Easy to understand and follow.
2.	Using a Gaussian–Markov random‑field prior over the grid Laplacian to enforce smoothness is very reasonable
3.	Methods comparison is comprehensive, the proposed method shows clear advantage over existing methods versus S2SLS‑LAG1, SPATIAL/SPATIAL+, GCNN, DAPSM

**Weaknesses:**

1.	Especially what additional information has CVAE learned in Z. Since all inputs are also used in the second stage.
2.	Theoretical analysis is relatively weak, the intuition is not clear why adding a CVAE is better than just use SPATIAL+.
3.	The radius is set to only 1 or 2 in experimental settings, which may be quite impractical in real applications. The real data typically reply on irregular graphs.

**Questions:**

1.	Assumption 4 is very strong and even impractical given unmeasured confounder. Could the authors provide intuition and failure modes and show robustness when this is violated?
2.	In Theorem 1, How necessary is the additivity for Z? This also seems very strong assumption since Z is essentially a representation of A,As,X,Xs.
3.	Some deeper analysis is needed to show intuition of what type of additional information contained in the Z that is learned and contribute to the performance gain.
4.	A systematic sensitivity analysis over neighborhood radius r is very necessary.
5.	How data size and dimensionality of X affect the performance of the new method?
6.	How the dimension of Z affect the final results?

---

> ### Author Response · Authors · 2025-11-25
>
> Thank you for your constructive feedback on our manuscript. Below, we provide answers to all the questions and concerns you raised. We **updated our PDF** accordingly and highlighted all key changes in **blue color**.
>
> ### **Answers to weaknesses:**
>
> **1. Additional information in Z:**
>
> By learning the substitute confounder $Z$, we aim to capture the **information contained in the unobserved confounder $U$**. Without this information, the final treatment effect estimates will be biased because the propensity and outcome models will not be able to learn the correct relationship due to the missing confounding information. In the **first stage**, we aim to **construct the substitute confounder $Z$** based on the spatial treatments $A$ and the observed confounders $X$. Note that in stage 1, we do not include the outcome $Y$ in the training process to prevent learning mediators instead of the substitute confounder, which would break the identifiability of the causal effect. In the **second stage**, we train the outcome model $h$ to predict $Y$ (see Eq. 9 and Fig. 3) through modeling the **conditional expectation of  $Y$ given all observed variables and the substitute confounder.** Therefore, the input to both stages is **not** the same.
>
> **Action:** We added more **explanation to Section 4** in the updated version of our manuscript. Furthermore, we now provide a **visualization of the latent confounder** showing that across latent dimensions we retrieve the information of the unobserved confounder $U$ in **new Figure 4**.
>
>
> **2. Benefits compared to Spatial+:**
>
> Thank you for allowing us to elaborate on the differences between the baseline method, Spatial+, and our approach. The Spatial+ method makes **restrictive parametric assumptions** in terms of a partially linear model, which are **unlikely to hold in practice**. If the underlying data does not follow the partially linear model, the causal effect **cannot** be identified. Our spatial deconfounder approach, however, is completely **non-parametric and model-agnostic** and does **not** make restrictive parametric assumptions. Furthermore, Spatial+ **ignores spatial interference**, i.e., it assumes that outcomes at location $i$ only depend on variables at location $i$ and the unobserved confounder. This is **unrealistic**. In spatial settings, interference is omnipresent, as, e.g., wind can transport air pollution from one location to another. Overall, we provide a much more **flexible method**, which **incorporates interference** relations and thus can lead to unbiased treatment effect estimates in spatial settings under unobserved confounding. We also demonstrate that our spatial deconfounder outperforms existing methods in our case study on the effect of air pollution on mortality and the components of PM2.5 in the US.
>
> **Action:** We highlighted the differences more clearly and provided more intuition for our spatial deconfounder in the updated version of our manuscript.
>
>
> **3. Neighborhood radius:**
>
> Thank you for raising this question. It is correct that **network data** often relies on irregular graphs in practice. However, in our work, we focus on **grid data**. In this type of data, every cell (besides the cells on the border) has the **same number of neighboring cells**. Therefore, specifying the neighborhood through a radius is reasonable.
>
> We agree with the reviewer that a **sensitivity analysis** over the neighborhood radius can be of interest to the reader and now include such analysis in the **updated version of our manuscript**. We observe that our spatial deconfounder is generally robust to misspecification of the neighborhood radius.
>
> **Action:** We include a sensitivity analysis over the neighborhood radius in **new Supplement E**.

---

> ### Author Response · Authors · 2025-11-25
>
> ### **Answers to questions:**
>
>
> **1. Assumption 4:**
>
> Thank you for giving us the chance to elaborate on the latent field sufficiency assumption. Our intent with Assumption 4 is to **adapt the “no single-cause confounders” condition** from Wang & Blei (2019) to the **spatial setting**. Standard spatial causal analyses would assume:
> $(A_s, A_{N_s}) \perp Y_s(a,a_{N_s}) \mid X_s, X_{N_s} \quad \text{for all } s$,
> which is often unrealistic when there are latent spatial fields (weather, socioeconomic context, etc.). We instead **weaken this by allowing an unobserved field $U(s)$** that acts *globally* across sites, but rule out additional idiosyncratic, site-specific confounders $\tilde U$ that affect only a single site’s exposures and outcome.
>
> To overcome the challenge of single-cause confounders, partial identification through causal sensitivity analysis (as common in standard causal settings) could offer a solution. We leave this to future work.
> We assess the robustness of our spatial deconfounder to violation of assumption 4 in currently running **new experiments**. We will upload the results and inform the reviewer as soon as possible.
>
> **Action:** We will provide new experiments to assess the robustness of our method to the presence of single-cause confounders in the next version of our manuscript.
>
>
>
> **2. Theorem 1:**
>
> The reviewer is correct that Theorem 1 requires separability of the unobserved confounder. This assumption is necessary for guaranteeing identifiability of the direct and indirect effects (see proof of Theorem 1 in Supplement B). We agree that the assumption is rather strong. However, we would like to note that the assumption is **standard in the literature** (e.g., [1],[2],[3]). Furthermore, many often unobserved or unavailable variables in spatial settings can be assumed to fulfill the equations in practice. In our air quality example, such variables could be persistent differences in baseline respiratory risk driven by unmeasured long-run pollution and chronic disease burden or regional differences in care-seeking and reporting intensity. Additionally, systematic measurement errors in the recorded outcome, i.e., due to the general difficulty in detecting, assessing, and correctly identifying respiratory diseases in a unified manner, can represent such latent confounders.
>
> **Action:** We included a discussion of the assumption and examples in the **updated version of our manuscript.**
>
>
> **3. Additional information in Z:**
>
> Please see our answer above to weakness 1.
>
>
> **4. Additional information in Z:**
> Please see our answer above to weakness 3.
>
>
> **5. Dependence on data size and covariate dimension:**
>
> Thank you for raising this question. As with any other causal treatment effect estimation method, the performance of our method **increases with the sample size**. The relation of the performance with regard to the dimension of $X$ depends on the informativeness of $X$. If $X$ is of low dimension but includes almost all confounders, the performance of our model is high. This is in line with other CATE estimators, as in this case, the propensity and outcome model estimation are comparatively simple. The same holds for the encoder and decoder training in the deconfounder. However, if $X$ is only of low dimension because we are missing a lot of (unobserved) confounders, the performance of our method will drop. When the dimension of $X$ is moderate and the relevant signal is concentrated in a low-dimensional subspace of $X$, our method performs well for the same reasons stated above. A very high-dimensional $X$ complicates the training of the pipeline as standard in ML estimation. Note, however, that a high-dimensional $X$ might be necessary for causal inference tasks to capture as many confounders as possible. Overall, the performance of our method does **not directly depend on the dimension of $X$**, but rather on the **confounding strength of $X$** in comparison to the confounding strength of the unobserved confounder $U$. For our method to achieve (nearly) unbiased estimation, the information in $X$ must be enough to learn an informative surrogate confounder $Z$. This can be assessed through the **predictive checks**.
>
>
> **6. Dependence on dimension of Z:**
>
> This is an interesting question. Observe that $Z$ is a latent surrogate confounder, i.e., not the original unobserved confounder. The dimension of $Z$ is **tunable** as standard in training VAEs. The final model performance will again **not depend on the dimension of $Z$**, but rather on the **information captured in $Z$**. The sufficiency of the information captured in $Z$ can be assessed through the **predictive checks**.

---

> > ### Author Response · Authors · 2025-11-25
> >
> > [1] Y. Wang and D. M. Blei. The blessings of multiple causes. Journal of the American Statistical
> > Association, 114(528):1574–1596, 2019.
> >
> > [2] G. Papadogeorgou and S. Samanta. Spatial causal inference in the presence of unmeasured confounding and interference. arXiv preprint arXiv:2303.08218 , 2023
> >
> > [3] Khan, Kori, and Candace Berrett. "Re-thinking spatial confounding in spatial linear mixed models." arXiv preprint arXiv:2301.05743 (2023).

---

### Official Review · Reviewer_2eqm · 2025-10-28

**Soundness:** 2
**Presentation:** 3
**Contribution:** 1
**Rating:** 2
**Confidence:** 4

**Summary:**

The paper addresses the problem of spatial confounding by proposing a method to recover an unobserved spatial confounder from observed neighbor treatments and covariates. The core methodology employs a Conditional Variational Autoencoder (CVAE), where the latent variable is intended to represent this hidden confounder.

For identification, the paper relies on a key assumption (Assumption 4) that extends a single-site ignorability condition to hold uniformly across all sites if it holds for some. The proposed estimator's validity is assumed directly (Assumption 5), with a reference made to a multi-cause confounding model by Wang and Blei (2019) as a conceptual analogy. The paper's setting is ambitious, aiming to handle spatial confounding in a broad context.

**Strengths:**

- Relatively well written.
- Transparent about its assumptions.
- Ambitious general problem setting and goals.

**Weaknesses:**

### 1: Estimation
The paper assumes away the core difficulties of estimation. It directly posits that the CVAE estimator is valid (Assumption 5), implying the hidden confounder can be fully recovered. However, a valid estimator must be tied to a specific identification strategy. Even with identification in hand, designing a valid estimator is highly nontrivial—especially for VAEs, where identifiability remains an active research area [1, 2]. The current approach mirrors the CEVAE paper [3], which has faced similar critiques [4, 5].

The reference to Wang and Blei (2019) does not substantiate the estimator's validity. The proposed estimator diverges from the identification and modeling in Wang and Blei: the multi-cause structure it alludes to is merely a vague analogy, and the VAE does not leverage their factor model.

The estimator offers limited novelty and relies on arbitrary architecture choices. The broad idea of a VAE-based causal effect estimator appears in numerous works [2, 3] (see references in [2, 5] for more examples), and this paper contributes nothing new. Moreover, the loss in eq. (8) leaves unclear what observational distribution the variational inference targets and how it factorizes. It is also unclear why the encoder takes its current form rather than other viable options. These details matter, as they encode the independence relationships assumed by the inference procedure.

### 2: Identification
Assumption 4, crucial for identification, is unclear and appears unrealistic. It states that if eq. (14) (ignorability?) “holds for some sites, then it holds uniformly across sites.” This is unclear compared to standard ignorability of joint exposure given observed covariates. First, it should be explicitly stated as an additional assumption that single-site ignorability (eq. 14) must hold. The paper seems to imply that ignorability is typically assumed across all sites, but it fails to explain why a "some sites, then uniformly across sites" condition would hold in practice—an example (perhaps based on Figure 2) would help.

The confounding/spatial structure is underexplained. This seems to tie back to Assumption 4, but it reads more like "global sharing" than true structure. Readers would expect a graphical model, or some algebraic/analytical structure in the functional equations, to substantiate it.

### 3: Missing Related Work
Key omissions include [2, 5], which develop general ideas for identifiable deep models, particularly *conditional VAEs*, that could bolster the current work. Specifically, they include an application to network deconfounding that exploits spatial information from neighbor cities—relevant to spatial deconfounding here. They also analyze and demonstrate the role of the $\beta$ parameter, which appears in this paper. See further applications to real-world health data in [6].


### Minor Points
- "Exposure" and "treatment" appear to be used interchangeably, which is confusing. Clarify or standardize to one term.

### References
[1] Khemakhem, Ilyes, et al. "Variational autoencoders and nonlinear ICA: A unifying framework." *International Conference on Artificial Intelligence and Statistics*. PMLR, 2020.

[2] Wu, Pengzhou Abel, and Kenji Fukumizu. "\beta-Intact-VAE: Identifying and Estimating Causal Effects under Limited Overlap." *International Conference on Learning Representations* (2022).

[3] Louizos, Christos, et al. "Causal effect inference with deep latent-variable models." *Advances in Neural Information Processing Systems* 30 (2017). [Early work, *not* recommended]

[4] Rissanen, Severi, and Pekka Marttinen. "A critical look at the consistency of causal estimation with deep latent variable models." *Advances in Neural Information Processing Systems* 34 (2021): 4207-4217.

[5] Wu, Pengzhou, and Kenji Fukumizu. "Towards principled causal effect estimation by deep identifiable models." *arXiv preprint arXiv:2109.15062* (2021).

[6] Ma, Wenao, et al. "Treatment outcome prediction for intracerebral hemorrhage via generative prognostic model with imaging and tabular data." *International Conference on Medical Image Computing and Computer-Assisted Intervention*. 2023.

**Questions:**

Please refer to the points in Weaknesses.

---

> ### Author Response · Authors · 2025-11-25
>
> We thank the reviewer for the careful and detailed comments. Below, we provide answers to all the questions and concerns you raised. We **updated our PDF** accordingly and highlighted all key changes in **blue color**.
>
> ### **1. Estimation**
>
> (a) What Assumption 5 actually requires
> Assumption 5 does **not** claim that the CVAE uniquely recovers the true latent confounder, nor that the VAE is generically identifiable in the sense of [1,2]. Instead, it is the **spatial analogue of the “consistency of substitute confounders” condition** in Wang & Blei’s Blessings of Multiple Causes (their Definition 4), where the posterior $p(Z_i \mid A_i,\theta)$ collapses to a point mass at a deterministic function of the multiple causes. In our setting, the multi-cause vector is the collection of own and neighbor treatments $(A_s, A_{N_s})$, and we allow the encoder to also use pre-treatment covariates $X_s, X_{N_s}$ as side information. Concretely, Assumption 5 states that the **encoder converges to a degenerate distribution**
> $q_\phi(Z_s \mid A_s, A_{N_s}, X_s, X_{N_s}) \Rightarrow \delta_{Z_s}, \text{with } Z_s = f_\phi(A_s, A_{N_s}, X_s, X_{N_s})$
>
> so that $Z_s$​ is a deterministic function of the observed inputs and, together with $(X_s, X_{N_s})$, suffices to render the joint treatment vector $(A_s, A_{N_s})$ ignorable.
> The **identification theorem** (Theorem 1) only assumes that some such $Z_s$​ exists and is a function of the observed causes and covariates; the proof does **not** rely on a particular neural architecture or on generic identifiability results for deep latent-variable models. In other words, identification is separated from estimation: if a learning procedure yields a $Z_s$​ that satisfies ignorability, then the plug-in estimator is consistent.
> We **clarified this explicitly in the paper** by:
> Rephrasing Assumption 5 as “existence of a substitute confounder $Z_s$​ that is a deterministic function of $(A_s, A_{N_s}, X_s, X_{N_s})$ and renders the joint exposure ignorable, in the sense of Definition 1” and
>
>
> Making clear that Assumption 5 is an identification assumption rather than a generic claim about VAE identifiability.
> (b) Relation to deep identifiability works [1,2,5]
> We agree that VAE identifiability is a nontrivial topic, and we appreciate the pointers. Our **goal in this paper is different**: we do **not** try to identify the latent factors up to a unique transformation in the sense of nonlinear ICA or $\beta$-Intact-VAE. Instead, we only need a latent representation $Z_s$​ that:
> is a (possibly non-unique) function of the spatial latent field $U(s)$, and
>
>
> suffices to restore ignorability when conditioning on $(X_s, X_{N_s}, Z_s)$.
> The recent works [1,2,5] provide sufficient conditions for identifiability of deep latent-variable models under additional side information (e.g., auxiliary variables, limited overlap), and we view them as complementary. In the revision, we:
> **Added these works** to the extended related-work section,
>
>
> Explicitly **discuss how their identifiability conditions could be leveraged** to further justify or strengthen Assumption 5 in future work, and
>
>
> **Clarify that we do not claim to solve generic VAE identifiability**, but rather leverage a substitute-confounder condition analogous to Wang & Blei (2019) in a new spatial-interference setting.
> (c) Loss in Eq. (8) and factorization / independence structure
> We apologize that the factorization was not explicit enough. The intended generative model for treatments is:
> $p(Z) = \mathcal{N}(0, \tau^{-1}(L + \varepsilon I)^{-1}),\quad p(A \mid X, X_N, Z) = \prod_{s \in S} p_\psi(A_s \mid X_s, X_{N_s}, Z_s)$
> with $p_\psi(A_s \mid X_s, X_{N_s}, Z_s) = \text{Bernoulli}(\sigma(f_\psi(\cdot)))$ as in Eq. (7). Thus, **conditional independence of treatments holds across sites given $(Z,X)$**, and **spatial dependence is encoded entirely via the GMRF prior** on $Z$. The ELBO in Eq. (8) is the standard sum over sites of $-\log p_\psi(A_s \mid X_s, X_{N_s}, Z_s)$ plus a KL term between the variational posterior and this prior. The **“multi-cause” structure** of $(A_s, A_{N_s})$ enters on the inference side through the **encoder** $q_\phi(Z_s \mid A_s, A_{N_s}, X_s, X_{N_s})$, which uses local treatment patterns (plus covariates) to infer a **substitute confounder for the local value of the spatial latent field.**
> We **added the explicit factorization** to the main text.
> (d) “Arbitrary” architecture choices and novelty of the estimator
> First, we note that we **do not use the CVAE for causal inference**, but merely to construct the substitute confounder. Second, our contribution on the estimation side is not just “use a VAE for causal inference.” The key design choices are tied to the spatial/interference setting:
> We condition the encoder on $(A_s, A_{N_s}, X_s, X_{N_s})$, i.e., we explicitly use **neighbor treatments as the multi-cause signal** that reveals the spatial latent field.

---

> ### Author Response · Authors · 2025-11-25
>
> We impose a GMRF (Gaussian–Markov random-field) prior with grid Laplacian $L$ on $Z$ to **encode the spatial smoothness and dependence structure** of the latent confounder, in line with our latent-field sufficiency assumption.
>
>
> We then **separate confounder reconstruction from outcome modeling**, plugging $U$ into a spatial outcome head (SPATIAL+ or U-Net) that captures interference in outcomes.
> To make this clearer, we **tightened the text around Algorithm 1** and **highlighted these design choices** as part of the method’s novelty, beyond simply adopting a VAE.
>
> ### **2. Identification**
>
> (a) Clarifying Assumption 4 and its relation to standard ignorability
> Our intent with Assumption 4 is to **adapt the “no single-cause confounders” condition** from Wang & Blei (2019) to the **spatial setting**. Standard spatial causal analyses would assume:
> $(A_s, A_{N_s}) \perp Y_s(a,a_{N_s}) \mid X_s, X_{N_s} \quad \text{for all } s$,
> which is often unrealistic when there are latent spatial fields (weather, socioeconomic context, etc.). We instead **weaken this by allowing an unobserved field $U(s)$** that acts *globally* across sites, but rule out additional idiosyncratic, site-specific confounders $\tilde U$ that affect only a single site’s exposures and outcome.
> The “if (14) holds for some sites, then uniformly across sites” wording was clearly confusing. What we mean is:
> If, after conditioning on $(X_s, X_{N_s}$, there is *any* unobserved confounding that only affects a particular site (rather than being mediated through a shared spatial field), it cannot be recovered from neighboring treatment vectors and hence cannot be removed by our deconfounder.
>
>
> Therefore, for the method to be valid, we assume (as in the original deconfounder work) that all such purely local, non-shared confounders are already included in $X$. Any remaining unobserved confounding must come from a latent spatial field $U$ that influences multiple sites.
>
>
> We **rephrased Assumption 4** to say this directly and explicitly connect this to the factor-model representation and to Proposition 5 of Wang & Blei (2019).
>
> “All confounders that act only on a single site are observed in $(X_s, X_{N_s})$. Any unobserved confounding is mediated through a shared spatial latent field $U$ which affects treatments across sites.”
>
> (b) Example illustrating plausibility
> We agree that an example will help. In the revision, we **added a new example** (tied to Figure 2):
> $U(s)$: a smooth meteorological field (humidity) that affects PM$_{2.5}$2​ levels and respiratory mortality across many nearby counties;
>
>
> $X_s$​: local demographic and healthcare covariates (observed);
>
>
> No additional “one-off” unmeasured confounder that only affects a single county independently of its neighbors.
>
>
> In such settings, it is plausible that the **only unmeasured confounding is spatially shared $U(s)$**, which is precisely what our method is designed to recover from the joint treatment patterns.

---

> > ### Author Response · Authors · 2025-11-25
> >
> > ### **3. Related work**
> >
> > We thank the reviewer for pointing out these works; they are indeed relevant, and we incorporated them in the updated version of our manuscript.
> > (a) Relationship to $\beta$-Intact-VAE and deep identifiable causal models
> > Wu & Fukumizu [2,5] propose identifiable deep models for causal effect estimation under multiple causes, including applications that use network structure. Our setting differs in two key ways:
> > Interference as the source of multiplicity. We are in a spatial-interference setting where each unit has a single treatment variable, but **interference provides multi-cause vectors** $(A_s, A_{N_s})$. Our contribution is to show that this interference structure can be used to construct a factor model suitable for deconfounding in spatial lattices and to **identify both direct and spillover effects**.
> >
> >
> > Spatial prior on the latent field. We introduce a GMRF prior tailored to spatial grids and prove an identification theorem for direct/spillover effects under localized interference and latent-field sufficiency, without specifying a parametric model for the hidden field.
> > We **expanded the related-work section** to (i) discuss $\beta$-Intact-VAE and related identifiable models, (ii) contrast their settings and assumptions with ours, and (iii) emphasize that our main novelty lies in the spatial causal formulation—using interference-generated multi-cause vectors plus a spatial prior, and proving identification of spillover effects.
> > (b) Network deconfounding and spatial deconfounding
> > We agree that network deconfounding is closely related and complementary. Our contribution is to instantiate and analyze this idea specifically for **spatial lattices with localized interference**, in a way that yields nonparametric identification of direct and spillover effects and that connects to existing spatial methods (SPATIAL+, S2SLS, etc.). We also extend the SpaCE benchmark to include interference and show that leveraging interference as a multi-cause signal yields systematic gains over both classical spatial models and purely predictive deep spatial models.
> >
> >
> > We made this positioning more explicit in the **introduction and related-work sections**.
> > Conclusion. We acknowledge that our identification and estimation strategy relies on structural assumptions (latent-field sufficiency, substitute confounders, and a particular CVAE parameterization), and we have clarified their role and limits above. That said, our goal is **not only conceptual but also practical**: in a highly realistic environmental-health setting—where we only minimally modify observed data to construct semi-synthetic ground truth—the spatial deconfounder consistently reduces bias and outperforms both classical spatial methods and strong deep learning baselines. This suggests that leveraging interference as a multi-cause signal, together with a spatial latent-field representation, is a promising and practically useful approach for mitigating spatial confounding, even if some underlying assumptions remain idealizations.
> >
> >
> > ### **Minor points**
> >
> > Thank you for noting the inconsistency.
> >
> > **Action:** We streamline the naming and now refer as *treatment exposure* to the spatial grid treatment, i.e., the tuple $(a_s, a_{\mathcal{N}_s)$ in the **revision of our manuscript**. When mentioning exposure mappings, we keep the naming, as it is a standard concept in causal inference in the presence of interference, e.g., [1],[2].
> >
> >
> >
> >
> > [1]  Peter M. Aronow. Cyrus Samii. "Estimating average causal effects under general interference, with application to a social network experiment." Ann. Appl. Stat. 11 (4) 1912 - 1947, December 2017. https://doi.org/10.1214/16-AOAS1005
> >
> > [2] L. Forastiere, E. M. Airoldi, and F. Mealli. Identification and estimation of treatment and interference effects in observational studies on networks. Journal of the American Statistical Association, 116(534):901–918, 2021.

---

### Official Review · Reviewer_jtsH · 2025-10-31

**Soundness:** 2
**Presentation:** 3
**Contribution:** 2
**Rating:** 4
**Confidence:** 3

**Summary:**

This paper proposes the Spatial Deconfounder, a two-stage framework for spatial causal inference that handles both unmeasured spatial confounders and interference (spillovers) between neighboring units. It treats the vector of a site’s own and neighbors’ treatments as a multi-cause signal and uses a CVAE with a spatial prior to reconstruct a smooth substitute confounder; this proxy is then fed into a flexible outcome model (e.g., U-Net/GNN) to estimate direct and spillover effects via plug-in contrasts. Under localized interference, positivity, and a weak “latent-field sufficiency” assumption, the authors show nonparametric identification of both effects without specifying a parametric latent-field model. Empirically, they extend the SpaCE benchmark to include interference and demonstrate, on environmental health and social-science datasets, that Spatial Deconfounder variants consistently reduce bias versus spatial econometric, spline/RSR, matching, and GNN baselines—while uniquely recovering spillover effects. Conceptually, the work reframes interference from a nuisance into a source of information for uncovering hidden spatial structure.

**Strengths:**

- The paper tackles unmeasured spatial confounding and localized interference together and establishes identifiability of both direct and spillover effects under spatial consistency, spatial positivity, localized interference, and a latent-field sufficiency assumption.

- The paper relaxes full ignorability by allowing an unobserved latent spatial field while only requiring that purely local confounders are observed, formalizing this with a latent-field sufficiency condition that is plausible for lattice-structured data.

- The paper provides explicit, nonparametric identification expressions for the direct and spillover effects, which anchor the subsequent estimation strategy.

- The estimation procedure cleanly separates confounder reconstruction from outcome modeling by using an interference-aware CVAE with a Gaussian-Markov random-field prior to recover a smooth substitute confounder, and then plugging this proxy into flexible outcome models such as U-Nets or GNNs.

**Weaknesses:**

1/ The approach relies critically on accurate reconstruction of the latent confounder via a CVAE; if the substitute confounder is poorly recovered, both direct and spillover effect estimates can be biased. can the authors quantify how estimation error in the latent proxy translates into bias or variance of the effect estimators?

2/ As a VAE-based method, the assignment model is vulnerable to posterior collapse and latent non-identifiability, which could yield an uninformative latent and ineffective adjustment. What concrete safeguards (e.g., KL warm-up schedules, mutual-information terms, decoder constraints) are used, and how often do collapse diagnostics fail in practice?

3/ The identifiability story ultimately assumes a "sufficient" latent field and that the learned proxy recovers a valid transformation of it; this is a strong modeling assumption at the learning stage. Can the authors provide formal or empirical evidence that their training procedure consistently recovers an informative proxy under realistic misspecification?

4/ Performance may be sensitive to hyperparameters that directly affect identifiability and collapse (e.g., \beta in the KL term, temperature or strength of the spatial prior). Do the results remain stable under a systematic sensitivity analysis over these hyperparameters?

5/ The spatial smoothness prior (e.g., GMRF) imposes a particular structure on the latent field; if the true confounding varies non-smoothly or anisotropically, the proxy may be biased. Have the authors evaluated robustness to misspecified spatial priors (non-stationary/anisotropic fields, sharp boundaries)?

6/ The localized-interference assumption is central; if interference extends beyond the specified neighborhood, the multi-cause signal may be mis-modeled and the recovered latent may absorb spillovers rather than confounders. How sensitive are estimates to enlarging or shrinking the interference neighborhood, and do the authors provide misspecification experiments?

7/ The method’s benefits rely on interference providing a strong multi-cause signal; in sparse treatments or weak neighbor correlations, the signal may be too weak to recover the latent reliably. Please present stress tests varying treatment sparsity and network density to map the regime where the approach is reliable.

**Questions:**

Please refer to the questions in Weaknesses.

---

> ### Author Response · Authors · 2025-11-25
>
> Thank you for your constructive feedback on our manuscript. Below, we provide answers to all the questions and concerns you raised. We **updated our PDF** accordingly and highlighted all key changes in **blue color**.
>
> **1) Estimation error of the latent confounder**
>
> Thank you for this interesting question. Note that we do **not** require the learned $Z_s$ to equal the true latent field, but only to recover a sufficient proxy of it. To assess whether the substitute confounder is adequately recovered, we perform **posterior predictive checks**. Posterior predictive checks (PPCs) are a straightforward way to assess whether a Bayesian model can generate data that resemble what you actually observed. We now **include the predictive check results in all our result tables**. We observe that all p-values are considerably close to the optimum value of 0.5 (average p-value across all experiments and models > 0.4).
>
> If the substitute confounder is poorly estimated, i.e., we do not recover sufficient information, the final treatment effect will be biased. The remaining, unidentified confounding information will act as a standard unobserved confounder. The size of the bias will depend on the confounding strength of the remaining unidentified information.
>
> **Action:** We added the results of the predictive checks in a **separate column to each results table** in our main paper and the Supplements.
>
>
>
> **2) Posterior collapse diagnostics:**
>
> As the reviewer stated, VAE-based methods can be vulnerable to posterior collapse and latent non-identifiability. To overcome this issue, we took the following safeguards: (i) KL warm-up schedule, (ii) weight decay to prevent overfitting, (iii) we only used models achieving p-values between 0.25 and 0.75 for estimating $\hat Z$ for the final outcome model training.
>
> In our experiments, the collapse diagnostics failed only rarely. It occurred only for very large models subject to overfitting, very large latent dimension $d_z$, or for settings in which we neglected the effect of the neighboring treatments (r=0) for C-VAE combined with Spatial+ .
>
>
> **3) Informativeness of proxy latent confounder:**
>
> We are happy to elaborate on the latent field sufficiency assumption. Our intent with Assumption 4 is to **adapt the “no single-cause confounders” condition** from Wang & Blei (2019) to the **spatial setting**. Standard spatial causal analyses would assume:
>
> $(A_s, A_{N_s}) \perp Y_s(a,a_{N_s}) \mid X_s, X_{N_s} \quad \text{for all } s$,
>
> which is often unrealistic when there are latent spatial fields (weather, socioeconomic context, etc.). We instead **weaken this by allowing an unobserved field $U(s)$** that acts *globally* across sites, but rule out additional idiosyncratic, site-specific confounders $\tilde U$ that affect only a single site’s exposures and outcome.
>
> To create insights into the learned proxy confounder, we included a **visualization of the reconstructed latent confounder** compared to the true (unobserved) spatial field in **Figure 4**. We observe that the leading principal component of the proxy confounder recovers the large-scale spatial structure of the true confounder.
> Furthermore, we will assess the robustness of our spatial deconfounder to a misspecified latent confounder in the form of an additional unobserved confounder. The experiments are currently running. We will upload the results and inform the reviewer as soon as possible.
>
> **Action:** We added a visualization of the reconstructed latent confounder in **new Figure 4** in the **updated version of our manuscript**. We will also provide new experiments to assess the robustness of our method to the presence of additional confounders in the next revision of our manuscript.

---

> > ### Author Response · Authors · 2025-11-25
> >
> > **4) Sensitivity analysis over the hyperparameters:**
> >
> > We agree with the reviewer that a **sensitivity analysis** over the hyperparameters can be of interest to the reader and now include such analysis in **Supplement E** in the **updated version of our manuscript**. We provide figures that display how the hyperparameters $\beta$ (KL term), the latent dimension $d_Z$, the learning rate, and weight decay affect the estimation performance. We observe the change in one parameter at a time, while optimizing the other hyperparameters conditional on the assessed parameter.
> >
> > Overall, we do not observe a consistent pattern in the error for the direct effect. The estimation performance **remains robust** when changing a single hyperparameter while optimizing all others. For the spillover effect estimation, we generally observe that the error increases with $\beta$ but decreases as $d_Z$ grows. Datasets with a large neighborhood radius $r_d$ typically need a low $\beta$ because the smoothness is lower. On the other hand, datasets with small $r_d$ need a higher $\beta$ to enforce smoothness constraints. Furthermore, the optimal value of $\beta$ depends on the nature of the unobserved confounder. For instance, models with a smooth confounder such as humidity $q_{summer}$ favor a larger $\beta$, whereas models with an anisotropic confounder like the population density $\rho_{pop}$ require a relatively smaller $\beta$.
> >
> > **Action:** We include a sensitivity analysis over the hyperparameters for various experiments in **new Supplement E**.
> >
> >
> >
> > **5) Robustness to spatial priors:**
> >
> > Thank you for raising this interesting question. Several of our experiments include the unobserved confounder of population density, which is high near urban centers and highly non-smooth. The results in Tables 1 and 2 for the $\rho_{\text{pop}}$ setting show that our spatial deconfounder framework reduces direct-effect bias relative to the benchmarks, demonstrating **robustness to anisotropic spatial priors**.
> >
> >
> > **6) Sensitivity to neighborhood radius:**
> >
> > We agree with the reviewer that a **sensitivity analysis** over the neighborhood radius can be of interest to the reader and now include such analysis in **Supplement E** in the **updated version of our manuscript**. We observe that our spatial deconfounder is generally robust to misspecification of the neighborhood radius.
> >
> > Furthermore, we will include the numerical results for r=3 in the tables in the next update of our manuscript.
> >
> > **Action:** We include a sensitivity analysis over the neighborhood radius in **new Supplement E**.
> >
> >
> >
> > **7) Interference strength:**
> >
> > The review raised an interesting point. Assessing the effect of treatment sparsity and network density can help to evaluate the robustness of our method. We are currently running experiments on this matter and will report them in the next revision of our manuscript. We will inform the reviewer when we upload the next revision.

---

### Official Review · Reviewer_UF6e · 2025-11-02

**Soundness:** 3
**Presentation:** 3
**Contribution:** 3
**Rating:** 4
**Confidence:** 4

**Summary:**

This work focuses on learning a model for which to perform causal inference in settings where covariates, treatments and outcomes live withing a spatial grid (such that neighboring variables can cause each other), and where the goal is to do so under: 1) interference, where treatments can affect outcomes in neighboring cells; and 2) spatially structured hidden confounding, which affects both the treatment and the outcome in that grid cell. The key idea of this work is that interference can be interpreted as a multiple-cause setting, for which they can leverage the work on the deconfounder framework to enable causal inference. Then, the authors propose a model based on conditional VAEs to build up a substitute of the hidden confounder, which is then fed to a potential outcome model that uses that substitute to perform causal inference. Moreover, the authors show causal identifiability results and empirically test their proposed method.

**Strengths:**

- **S1.** The core idea behind this work is sound and interesting: Interference as observed in climate science applications can be interpreted as a multi-cause framework.
- **S2.** The paper is well-motivated and well-written, with clear text, explanations, and walking the reader through all the process.
- **S3.** The proposed model is sound, and the authors provide some causal identifiability results to back it up.
- **S4.** All the assumptions are clearly stated and they are reasonable and sound.
- **S5.** The experimental results show promise on the proposed model.

**Weaknesses:**

- **W1.** I have a few concerns regarding assumptions:
  - **W1.1.** Assumption 5 sounds a bit unrealistic, given that the model uses a Gaussian with positive variance to model the hidden confounder. I understand it can be interpreted in the "nearly-deterministic" setting if the variance is extremely small, but it needs to be ensured. ([This paper](http://arxiv.org/abs/2206.02416) could be of interest to the authors regarding this topic.)
  - **W1.2.** Causal identifiability in theorem 1 relies on the hidden confounding being separable from the rest of variables in the structural equation.
  - **W1.3.** Assumption 2 regard the learned hidden confounder as far as I understand, which depends on the model itself.
- **W2.** Related work does not include any causal generative models, which are relevant and related with the current work. This includes  [NCMs](https://arxiv.org/abs/2107.00793), [CNFs](https://arxiv.org/abs/2306.05415), or [Diff-SCM](http://arxiv.org/abs/2202.10166), among others (see references therein). [Follow-up work](https://proceedings.mlr.press/v139/wang21c.html) of the deconfounder framework by the same lab should also be relevant.
   - **W2.1.** Particularly related to this work is [DeCaFlow](https://arxiv.org/pdf/2503.15114), which combines the Deconfounder framework with Causal Normalizing Flows to perform causal inference under hidden confounding with a given yet general causal graph. DeCaFlow shares quite some similarities with the proposed model, where it also uses a encoder-decoder architecture to build a substitute of the hidden confounder, and trains with the ELBO. Indeed, DeCaFlow should be applicable in the experiments as a baseline. I'd suggest to relax the statements regarding being the "first framework" (line 70).
- **W3.** Following up on the experiments, I have three main concerns.
  - **W3.1.** I am not sure to understand what it means to "mask" some elements. Does it mean to zero-them out? Or to completely removed them from the model's input?
  - **W3.2.** Similarly, I am not sure why the exogenous noise was introduced in equations 19 and 20 as additive, rather than as another input to the function $f$.
  - **W3.3.** Finally, I am concerned about the selection of baselines, most of them looking rather weak as they are not causally-aware and, despite that, the differences in performance with the proposed model are not statistically significant, which concerns me the most. (Indeed, I find the bold numbering misleading.)

**Questions:**

- **Q1.** What do the authors mean by "Posterior draws of Zs yield uncertainty bands." in line 263?
- **Q2.** Are the samples of the treatment from the decoder in line 269 sampled using samples from the prior of z? or from the posterior?
- **Q3.** Where are the predictive checks used?
- **Q4.** What is $L_Y$ in remark 1?

---
Other feedback:
- I'd be careful with using CVAE as name during the manuscript, as it closely resembles another causal inference work: [CEVAE](https://arxiv.org/abs/1705.08821).

I'll happily increase my score after my concerns are addressed.

---

> ### Author Response · Authors · 2025-11-24
>
> Thank you for your feedback and the positive evaluation of our manuscript. Below, we provide answers to all the questions and concerns you raised. We **updated our PDF** accordingly and highlighted all key changes in **blue color**.
>
> ## **Answers to weaknesses:**
>
> ### **W1: Assumptions**
>
> **Assumption 5:**
> Assumption 5 presents the **spatial analogue of the “consistency of substitute confounders” condition** in Wang & Blei’s Blessings of Multiple Causes (their Definition 4), where the posterior $p(Z_i \mid A_i,\theta)$ collapses to a point mass at a deterministic function of the multiple causes. In our setting, the multi-cause vector is the collection of own and neighbor treatments $(A_s, A_{N_s})$, and we allow the encoder to also use pre-treatment covariates $X_s, X_{N_s}$ as side information. Concretely, Assumption 5 states that the **encoder converges to a degenerate distribution**. This assumption is common in the literature, e.g., [1],[3].
>
> $q_\phi(Z_s \mid A_s, A_{N_s}, X_s, X_{N_s}) \Rightarrow \delta_{Z_s}, \text{with } Z_s = f_\phi(A_s, A_{N_s}, X_s, X_{N_s})$
>
> so that $Z_s$​ is a deterministic function of the observed inputs and, together with $(X_s, X_{N_s})$, suffices to render the joint treatment vector $(A_s, A_{N_s})$ ignorable.
>
> We would like to thank the reviewer for highlighting the useful reference in the review. Indeed, replacing the standard ELBO by the  IMA-regularized objective can help invert the data-generating process and thus fulfill the assumption.
>
> **Action:** We included an **explanation on how IMA can help** to satisfy assumption 5 in the **updated version of our manuscript**.
>
>
>
> **Identifiability:**
> The reviewer is correct that Theorem 1 requires separability of the unobserved confounder. We agree that the identifiability of our method only applies to settings in which the specific structural equations hold. However, we would like to note that the assumption is **standard in the literature** (e.g., [1],[2],[4]). Furthermore, many often unobserved or unavailable variables in spatial settings can be assumed to fulfill the equations in practice. In our air quality example, such variables could be persistent differences in baseline respiratory risk driven by unmeasured long-run pollution and chronic disease burden or regional differences in care-seeking and reporting intensity. Additionally, systematic measurement errors in the recorded outcome, i.e., due to the general difficulty in detecting, assessing, and correctly identifying respiratory diseases in a unified manner, can represent such latent confounders.
>
> **Action:** We included a discussion of the assumption and examples in the updated version of our manuscript.
>
> **Assumption 2:**
> Thank you for raising this question. Assumption 2 postulates the standard causal positivity assumption as well as a *latent positivity* requirement. The latter involves the (true) latent confounder we aim to estimate; *not* the model or the estimate itself. The assumption is **equivalent to the standard positivity assumption** in that we require a positive probability of treatment and neighborhood treatment assignment conditional on both the observed variables and the latent confounder, i.e., all propensities must be bounded away from 0 and 1.
>
>
>
>
> ### **W2: Related work**
>
> We agree with the reviewer that a discussion of the related work on causal generative models and the difference between our method and this research field can be of interest for the reader, and therefore **include a discussion** including all references stated by the reviewer in the **updated version of our manuscript**. Regarding the method DeCaFlow as a baseline, we would like to point out that the method only applies to continuous variables (especially continuous treatments). In our experiments (and in general real-world settings), variables are both discrete and continuous. Therefore, the DeCaFlow is **not applicable** as a baseline. We discuss the differences to the DeCaFlow in the new extended related work section.
>
> **Action:** We provide a discussion of the related work on causal generative models in **new Supplement A.4**.

---

> ### Author Response · Authors · 2025-11-24
>
> ### **W3: Experiments**
>
> **Masking of elements:**
> Thank you for noting the ambiguity of the word “mask”. We adopted the wording from the SpaCE paper upon which we base our dataset construction. However, we agree that clarification is necessary. To simulate unobserved confounding, we mask key covariates after data generation, meaning we completely **remove** them from the dataset.
>
> **Action:** We added the clarification to the updated version of our manuscript.
>
>
>
> **Exogenous noise:**
> In our experiments, we aim to simulate **realistic scenarios** for causal inference. This is why we include interference and unobserved confounders. In reality, we only have access to the outcome and covariate measurements, so it is important for the model not to be aware of the exogenous noise.
>
>
> **Baselines:**
> We are unsure if we fully understand your question. **3 of our 5 baselines are causally aware** methods for spatial data. For **additional comparison**, we included the 2SLS-Lag1 method, as this is a commonly used method in spatial econometrics, and a GNN-based approach, as GNNs provided a different but powerful approach to analysing spatial data. In our experiments on spatial confounding, i.e., when not only the neighboring treatments but also the neighboring confounders affect the outcome at a certain cell, we combine our spatial deconfounder method with a UNet. To highlight the superiority of our spatial deconfounder method over the simple UNet, we also include it as an additional baseline for this setting.
>
> The performance of our spatial deconfounder is superior to the baselines or on par with the best baseline for our experiment settings. On average across all experiments, our spatial deconfounder **significantly outperforms the baselines** (see Tables 1+2, 5+6). The best-performing model is highlighted in bold font. Additionally, note that most of the **baselines cannot estimate the spillover effects**. Therefore, our spatial deconfounder provides **both** superior performance and additional estimation capabilities.
>
>
>
>
> ## **Answers to questions:**
>
> ### **Q1: Posterior draws of Z**
>
> Thank you for pointing out that the formulation might not be clear to the reader. By drawing multiple $\hat Z_s$ from the full posterior $q_{\phi}$ instead of the mean, we can obtain uncertainty bands on $\hat Z_s$. We can then obtain uncertainty bands (with respect to the substitute confounder) by evaluating Eq.11 on the different draws of $\hat Z_s$.
>
> **Action:** We rephrased the sentence and added more explanation in the updated version of our manuscript.
>
>
> ### **Q2: Predictive check sampling**
>
> To assess whether the substitute confounder adequately explains the treatment assignment, we perform **posterior predictive checks**. Posterior predictive checks (PPCs) are a straightforward way to assess whether a Bayesian model can generate data that resemble what you actually observed. Specifically, we **sample $Z$ from the posterior $q_{\phi}$** to obtain the test statistic $T(a)$ over the posterior of $A$ according to Eq. 13 (former Eq. 12). Then, we calculate p-values according to Eq. 12 (former Eq. 11).
>
>
> ### **Q3: Predictive check results**
>
> Thank you for noting that the results of the predictive checks can be important to the reader. We now **include the predictive check results in all our result tables**. We observe that all **p-values are considerably close to the optimum value** of 0.5 (average p-value across all experiments and models > 0.4). Of note, the PM2.5 components typically have p-values very close to 0.5, likely due to a smoother distribution of covariates and less sparsity in the treatments, making the confounder easier to reconstruct than the other tested scenarios.
>
> **Action:** We added the results of the predictive checks in a **separate column to each results table** in our main paper and the Supplements.
>
>
> ### **Q4: Remark 1**
>
> Thank you for noting that we had only introduced the outcome model loss $\mathcal{L}_Y$ in Algorithm 1.
>
> **Action:** We added the definition of the loss as well to the main text (see Eq. 10).
>
>
>
> [1] Y. Wang and D. M. Blei. The blessings of multiple causes. Journal of the American Statistical
> Association, 114(528):1574–1596, 2019.
>
> [2] G. Papadogeorgou and S. Samanta. Spatial causal inference in the presence of unmeasured confounding and interference. arXiv preprint arXiv:2303.08218 , 2023
>
> [3] T. Hatt and S. Feuerriegel. Sequential deconfounding for causal inference with unobserved confounders. In Conference on Causal Learning and Reasoning, 2024.
>
> [4] Khan, Kori, and Candace Berrett. "Re-thinking spatial confounding in spatial linear mixed models." arXiv preprint arXiv:2301.05743 (2023).

---

### Author Response · Authors · 2025-12-02
**Closing Remarks & Author Summary for Area Chair**

As the discussion period is nearing its end, we are grateful to all reviewers for their thoughtful and detailed feedback—your comments have already helped us refine and strengthen this work. Given the unusual situation this year (reviews frozen), we would like to briefly summarize our contribution and how we have addressed the main concerns in the updated draft (all changes are highlighted in blue).

---
### 1. Brief summary of our work
**(1) Problem.** The Spatial Deconfounder addresses two ubiquitous challenges in spatial causal inference:
*  **Localized interference** (spillover from neighbors), and
* **Unobserved spatial confounding** (latent fields such as meteorology or socio-economic context).

**(2) Key idea and method**. Interference itself creates a **multi-cause signal**: each site’s *s* treatment exposure $(A_s, A_{\mathcal N_s})$ is shaped by a shared latent field $U(s)$. Leveraging the deconfounder framework [Wang and Blei, 2019], we:
- use an interference-aware C-VAE with a spatial (GMRF) prior to reconstruct a smooth substitute confounder $Z_s$ from $(A_s, A_{\mathcal N_s}, X_s, X_{\mathcal N_s})$, and
- plug $Z_s$ into a flexible spatial outcome model (SPATIAL+ or U-Net) to estimate **direct** and **spillover** effects via plug-in contrasts.

**(3) Guarantees and practical impact.** Under localized interference and classical deconfounder-style assumptions (latent-field sufficiency and consistency of substitute confounders), we prove **nonparametric identification** of both direct and spillover effects. That said, our goal is not only conceptual but also practical: despite relying on idealized assumptions, our semi-synthetic experiments on minimally modified environmental-health and air-quality benchmark data show that the Spatial Deconfounder **reduces bias relative to strong classical and deep-learning spatial baselines and uniquely recovers spillover effects**, providing empirical support for leveraging interference-driven multi-cause vectors in practice.

---

### 2. What reviewers appreciated

Across reviews, several shared strengths of the work were highlighted:

* **Core idea and framing (UF6e, 2eqm)**. **UF6e** described the main idea as *“sound and interesting: interference as observed in climate science applications can be interpreted as a multi-cause framework,”* and noted that the paper is *“well-motivated and well-written,”* with assumptions that are *“clearly stated”* and *“reasonable and sound”*. **2eqm** acknowledged that the paper addresses an “ambitious general problem setting and goals.”.

* **Joint treatment of interference and hidden confounding (jtsH)**. **jtsH** emphasized that the paper *“tackles unmeasured spatial confounding and localized interference together”* and *“relaxes full ignorability by allowing an unobserved latent spatial field,”* while still providing *“explicit, nonparametric identification expressions for the direct and spillover effects.”*

* **Methodology and empirical advantage (yeAS)**. **yeAS** found that *“the methodology is pretty clear and straightforward. Easy to understand and follow,”* that using a GMRF prior is *“very reasonable,”* with the proposed method showing *“clear advantage over existing methods versus S2SLS-LAG1, SPATIAL/SPATIAL+, GCNN, DAPSM.”*

---
### 3. How we addressed the reviewers’ concerns

We grouped the reviewers’ concerns into a few themes and updated the paper accordingly (blue text):

**(a) Assumptions and identifiability**

We **rephrased Assumption 4** as a spatial analogue of “no single-cause confounders”: all purely local confounders are assumed observed in $(X_s, X_{\mathcal N_s})$, and any remaining unobserved confounding arises from a shared spatial latent field $U$ that affects multiple sites. We added a humidity example (Figure 2) to illustrate when this is plausible (smooth meteorology vs. idiosyncratic local shocks).

We **rewrote Assumption 5** as an existence statement: there exists a deterministic function
$Z_s = f_\phi(A_s, A_{\mathcal N_s}, X_s, X_{\mathcal N_s})$
such that conditioning on $(X_s, X_{\mathcal N_s}, Z_s)$ restores ignorability. We emphasize that this is a **causal identification assumption**, not a claim that CVAEs are generically identifiable; Theorem 1 only requires that some such $Z_s$ exist, independent of the specific architecture.

We clarified the **separable outcome structure** in Theorem 1, noted that it is standard in related work, and gave concrete examples of latent variables (baseline risk, chronic disease burden, measurement error) that reasonably fit this form in our environmental-health setting.

---

> ### Author Response · Authors · 2025-12-02
> **Closing Remarks & Author Summary for Area Chair (Cont'd)**
>
> ### 3. How we addressed the reviewers’ concerns (continued)
>
> **(b) Estimation and architecture**
>
> We now **spell out the generative model** for the treatment field: $p(Z)$ is a GMRF prior, and
> $p(A \mid X,Z) = \prod_s p_\psi(A_s \mid X_s, X_{\mathcal N_s}, Z_s)$,
> with spatial dependence encoded via the prior. The “multi-cause” structure $(A_s, A_{\mathcal N_s})$ enters only through the encoder
> $q_\phi(Z_s \mid A_s, A_{\mathcal N_s}, X_s, X_{\mathcal N_s})$.
>
> We clarify that the **C-VAE is used only to reconstruct a substitute confounder**, and that outcome modeling is a separate second stage, which avoids mediator leakage and mirrors standard deconfounding practice.
>
> To address concerns about collapse and informativeness of $Z$, we describe safeguards (KL warm-up, weight decay, discarding poor models) and now report posterior predictive $p$-values in all tables (typically near 0.5); we also add a visualization showing that principal components of $Z$ recover large-scale structure of the true confounder.
>
> **(c) Experiments, robustness, and practical performance**
>
> We clarified the **data-generation and masking** procedure: key covariates are removed after learning the outcome model from real data, so semi-synthetic outcomes remain close to the original environmental-health data while encoding realistic hidden confounding.
>
> We added **sensitivity analyses** (Appendix E) over hyperparameters $(\beta, d_Z,$ learning rate, weight decay) and over the neighborhood radius $r$. Direct-effect estimates remain stable under these perturbations; spillover-effect errors follow interpretable patterns (e.g., larger $\beta$ for smoother confounders). The new hyperparameter-tuning plots also show that the C-VAE–UNet variant is noticeably more stable than C-VAE–SPATIAL+.
>
> We use **strong causal baselines** (SPATIAL+, S2SLS, DAPSM, GCNN, UNet) and highlight that most cannot estimate spillover effects at all. Across the main experiments and additional settings with anisotropic, non-smooth confounders (e.g., population density), the Spatial Deconfounder reduces standardized bias for direct effects and provides reliable spillover estimates. New sparsity experiments further show that, in extremely sparse treatment regimes, our method substantially improves upon SPATIAL+; robustness to an additional single-cause unobserved confounder is currently being evaluated and will be incorporated in a subsequent revision.
>
> **(d) Related work and positioning**
>
> We expanded the extended related work to discuss causal generative models, identifiable VAEs, and network deconfounding (e.g., Intact-VAE, $\beta$-Intact-VAE, DeCaFlow), and clarified that our focus is complementary: we specialize to spatial lattices with localized interference and show how interference-generated multi-cause vectors, together with a spatial prior on $Z$, can be used to identify and estimate direct and spillover effects. We also emphasize that the Spatial Deconfounder is **model-agnostic** and could be paired with factor models beyond the C-VAE used here.
>
> ---
>
> ### 4. Closing remarks for the AC
> All four reviewers agree that the problem we tackle—**jointly handling localized interference and unobserved spatial confounding**—is important and that our framing of interference as a multi-cause signal is conceptually compelling. The main concerns centered on the strength and clarity of our assumptions, the connection to deconfounder-style identifiability, the role and robustness of the C-VAE, and missing connections to recent causal generative and identifiable deep models.
> We have worked hard to address these points by:
> * Making the **assumptions more precise and transparent**, and tying them explicitly to existing theory (latent-field sufficiency, substitute confounders).
> * Clarifying the **estimation procedure and generative model**, including safeguards against collapse and diagnostics for the informativeness of $Z$.
> * Providing additional **robustness analyses** (hyperparameters, neighborhood radius, anisotropic confounders) and richer empirical evidence, while acknowledging that some aspects remain idealizations.
> * Situating the Spatial Deconfounder more clearly relative to network deconfounding, identifiable VAEs, and causal generative models.
>
> In our view, the Spatial Deconfounder offers a principled, flexible, and empirically validated approach for a class of spatial causal problems that are both practically important and methodologically underserved. We hope this strengthened version will be considered favorably alongside related work in spatial causal inference, deconfounding, and deep latent-variable modeling.

---

### Meta-Review · Area_Chair_Bjr6 · 2026-01-07

**Summary:**

This submission proposes an approach to localize spatial interference as a multi-cause signal. The objective is addressing unobserved spatial confounding, combining insights and tools from spatial causal inference and deep latent-variable models. Its main contribution is unifying how interference can be exploited to construct substitute confounders. It provides identifiability guarantees for direct and spillover effects under localized interference and latent-field sufficiency assumptions. While the framing is novel in that it naturally integrates these tools, the core ideas are generally incremental, building on existing deconfounder and network/spatial methods. Additionally, while the theoretical insights are presented clearly and appear technically sound, they rely on strong assumptions that significantly narrow the work's scope (e.g., no single-cause confounders, a separable outcome structure). Finally, there are several key relevant lines of work that the manuscript had not discussed. While the authors have included them in the revised paper, I believe properly discussing them and providing adequate analysis to establish the connections to and differences from them requires more extensive effort.

**Reviewer Concerns:**

Despite careful rephrasing, the rebuttal does not address the core circularity issue of the paper, based on which identification is conditional on the existence of a substitute confounder that renders the joint treatment ignorable; specifically, no concrete conditions are given under which the proposed CVAE actually recovers such a confounder. Furthermore, the rebuttal acknowledges that poor recovery of $$Z$$ leads to bias, but does not offer a way to quantify the relationship between proxy error and bias/variance of the causal estimands. Furthermore, the missing literature and the pertinent discussion and analysis -- which are acknowledged -- are not reflected in the paper satisfactorily.

**Reviewer Scores:**

In the face of a lack of response by the reviewers, it is difficult to gauge how they have received the rebuttals. However, given the extent and scope of the criticism and the responses provided, I do not believe, collectively, the rebuttals would have changed the reviewers' position on the paper in a significant way that would rise to the level of the paper having enough support to be accepted.

---

### Decision · Program_Chairs · 2026-01-26

Reject